# A live biohybrid bacterial therapy based on engineered *Serratia marcescens*

Lihao Ji[1], Tianze Zhu[1], Tianqi Jiang[1], Li Wang[1], Zhonghui Qiu[1], Shiqi Gao[1], Yuqi Wang[1], Jing Wang[2,3], Jingyi Zhang[2], Haomiao Huang[4], Yunlong Mao[5], Chen Lin [2], Jing Zhao [1,2,3,6] ✉, Xiuxiu Wang[1,2,3] ✉ & Wei Wei [1,3,6] ✉

Bacterial therapeutics hold great promise for cancer treatment by targeting oxygen-poor tumor regions and complementing existing therapies. However, current approaches often struggle with safety concerns and complex engineering. Developing a safe, effective delivery platform relying entirely on natural bacterial biosynthesis remains a challenge. Here we show that attenuated *Serratia marcescens* serves as a powerful biohybrid platform for cancer therapy by leveraging its natural biosynthesis of prodigiosin, a photosensitive pigment. We engineer *S. marcescens* to yield high prodigiosin levels, which exhibit strong intrinsic anti-cancer activity and near-infrared photosensitivity. In female mouse models of melanoma and colorectal cancer, this platform triggers robust systemic immune responses, including enhanced T cell recruitment and long-term memory against tumor recurrence. Furthermore, the bacteria induces tumor cell death via mitophagy, while photothermal properties of prodigiosin enables rapid, light-controlled bacterial clearance post-treatment. These findings establish *S. marcescens* as a versatile, self-regulating biosynthetic platform for precise and safe cancer immunotherapy.

Bacteria-based delivery systems have garnered attention due to their unique capabilities in targeting and treating various diseases, including cancer and bacterial infections[1-4]. These systems leverage the natural properties of bacteria-such as hypoxia-targeting capability, versatile biosynthetic machinery, and immunostimulatory potential-to address drug resistance[5,6]. Various bacterial strains have been employed as delivery vehicles to date, including *Salmonella typhimurium* VNP20009[7-10], *Clostridium novyi*-NT[11-14], *Escherichia coli*[15,16], *Listeria monocytogenes*[17-19] and so on. Among these, VNP20009 and C. novyi-NT have undergone attenuation to improve safety profiles and have reached Phase I clinical trials[20-22], whereas others remain in preclinical development. Conventional bacterial therapies predominantly inhibit tumor growth indirectly by activating the immune system[23,24]. However, their direct ability to kill tumor cells remains limited and often

necessitates intricate physical or chemical modifications to facilitate drug or therapeutic media delivery—a process that is time-intensive, inefficient, with limitations in drug release efficiency and scalability.

To address this limitation, researchers have employed genetic engineering to amplify the innate anti-tumor immune function, thereby circumventing the reliance on external interventions. For instance, Andrew et al.[2] engineered *Escherichia coli* Nissle 1917 to deliver tumor-specific neoantigenic epitopes, thereby activating specific CD4+/CD8+ T cells and significantly enhancing the efficacy of antitumor immunotherapy. Similarly, Wang et al.[25] modified *Salmonella typhimurium* VNP20009 to secrete granulocyte-macrophage colony-stimulating factor (GM-CSF), which persistently induced the reprogramming of M2-type tumor-associated macrophages into M1-type anti-tumor macrophages, thereby boosting the anti-tumor

[1]State Key Laboratory of Coordination Chemistry, Chemistry and Biomedicine Innovation Center (ChemBIC), School of Life Sciences, Nanjing University, Nanjing, P. R. China. [2]School of Chemistry, Nanjing University, Nanjing, P. R. China. [3]Wuxi Xishan NJU Institute of Applied Biotechnology, Wuxi, P. R. China. [4]Nanjing Foreign Language School, Nanjing, P. R. China. [5]State Key Laboratory of Novel Software Technology, Nanjing University, Nanjing, P. R. China. [6]Nanchuang (Jiangsu) Institute of Chemistry and Health, Sino-Danish Ecolife Science Industrial Incubator, Jiangbei New Area, Nanjing, P. R. China. ✉e-mail: jingzhao@nju.edu.cn; wangxiuxiu@nju.edu.cn; weiwei@nju.edu.cn

immune response. While these strategies effectively inhibit tumor growth by heightening bacterial immune activation, their direct tumor-cell-killing capabilities remain constrained. Additionally, developing a safe, biosynthesis-based platform for anticancer drug delivery continues to pose significant challenges.

Bacteria serve as an important platform for anticancer natural products, such as polyketide/macrolide antibiotics and non-ribosomal peptides[26,27]. However, their complex synthetic pathways and difficulties in heterologous expression often constrain production yields[28–30]. By contrast, microbial pigments with antitumor abilities can typically be produced in large quantities, making them an attractive alternative[31]. Among these is *Serratia marcescens*–derived prodigiosin, a linear tripyrrole bright red natural product with demonstrated antibacterial, antioxidant, and anti-cancer properties[32–34]. Prodigiosin suppresses the Wnt/β-catenin pathway by downregulating the phosphorylation of key regulatory factors, thereby reducing β-catenin activation, lowering cyclin D1 expression, and ultimately inducing tumor cell apoptosis to inhibit tumor progression[35]. However, its application in cancer therapy has been hampered by challenges, including its low bioavailability, chemical instability in varying environments, and short systemic half-life[36]. Unlocking its potential requires a deeper understanding of its properties and innovative mechanisms for its application.

In this study, we report a biohybrid strategy based on attenuated *S. marcescens* for cancer therapy that leverages the natural biosynthesis of photosensitive prodigiosin. This biohybrid approach addresses both safety and efficacy concerns: it delivers potent antitumor effects in vitro and in vivo, inducing robust cell death, mainly through mitophagy-driven necroptosis and caspase-3-independent apoptosis in tumor cells, promotes immunological responses (dendritic cell maturation, T cell recruitment, and macrophage polarization), and rapid bacterial clearance via prodigiosin-mediated photothermal ablation. Collectively, these findings indicate that the engineered *S. marcescens* can serve as a robust chassis for delivering photosensitive natural products, offering a valuable avenue for more precise and effective bacterial cancer therapies (Fig. 1).

## Results

### Discovery of prodigiosin exhibiting near-infrared fluorescence and photothermal effects

Starting from *S. marcescens* NRRLB-1481, we developed an engineered strain, *S. marcescens* JC11 (hereafter referred to as SMM), through iterative UV irradiation and visual screening for intensified red

pigmentation, indicating enhanced prodigiosin biosynthesis (Supplementary Figs. 1 and 2). Random mutagenesis yielded SMM, capable of producing a competitive $8.3 \pm 0.5$ g/L prodigiosin (upper range for native *S. marcescens* under optimized conditions) in a 5-L bioreactor within 48 h (Supplementary Table 2: representative production studies). While yields of up to 10.25 g/L have been achieved through techniques like promoter engineering of OmpR and PsrA[37], SMM offers a scalable and cost-effective alternative, particularly valuable for downstream translational and clinical applications. Interestingly, this high-yield phenotype is tightly regulated by temperature; consistent with the known regulatory network, prodigiosin biosynthesis peaks at 30 °C because the LysR-family repressor HexS silences the pigA-N promoter at ≥37 °C[38,39]. Activators such as PigP remain expressed at higher temperature but cannot override HexS-dependent repression[40]. A serendipitous live-cell imaging experiment revealed that prodigiosin emits fluorescence under 808 nm excitation (Fig. 2a). Emission scanning identified a primary peak at approximately 940–950 nm (near-infrared I), extending to around 1300 nm (near-infrared II), with a first-order fluorescence lifetime of 1208 ns (Fig. 2c, d). At a physiological temperature of 37 degrees, the bacteria undergo physiological degradation, with a clearance efficiency of over 80% observed within seven days (Supplementary Fig. 3a, b). This characteristic of the strain also indicates its potential to develop into an excellent platform for integrated diagnosis and treatment delivery.

A 5 mM solution of purified prodigiosin in ethanol exhibited potent photothermal conversion, with temperatures rapidly exceeding 60 °C (under 4.5 W/cm² irradiation) (Fig. 2e, f and Supplementary Fig. 4a). Importantly, the bacteria with pre-synthesized prodigiosin, acting as living photothermal agents also demonstrated this capability (Supplementary Fig. 5a). Aqueous suspensions of SMM showed a rapid, concentration-dependent temperature increase under 808 nm laser irradiation (4.5 W/cm²), with a concentration of $1 \times 10^{11}$ CFU/mL reaching over 60 °C in just 3 min (Fig. 2g, h and Supplementary Fig. 5). This bacterial agent with pre-synthesized prodigiosin also exhibited excellent photostability across multiple heating cycles (Supplementary Fig. 5b). This temperature threshold (-60 °C) was validated as being sufficient for thermal ablation; a 3-min treatment at 60 °C resulted in a greater than 8-log reduction in bacterial viability, effectively sterilizing the culture (Supplementary Figs. 6 and S7). However, routine purification and direct use of prodigiosin are hampered by the co-production of proteases and surfactants, as well as prodigiosin's structural complexity and moderate instability. Building on the hypoxia-targeting capability of SMM and the intrinsic anticancer

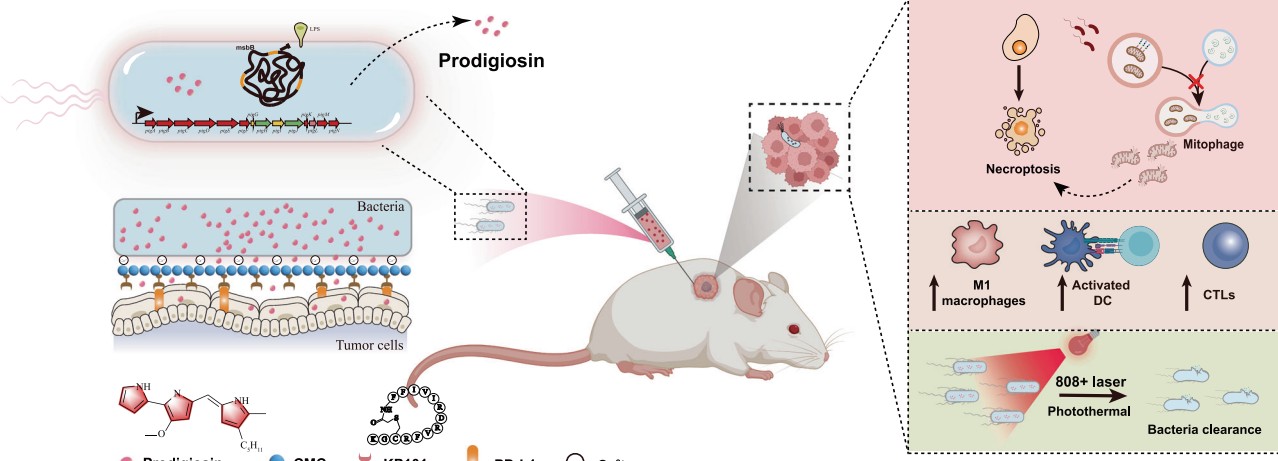

**Fig. 1 | Delivery mechanism of the attenuated *S. marcescens* system for tumor therapy.** The system induces necrosis and apoptosis in tumor cells while enhancing anti-tumor immunity and facilitating bacterial clearance via near-infrared photothermal effect. Created in BioRender. JI, L. (2026) https://BioRender.com/uma9zrk. All non-BioRender elements in this figure were independently created by the authors.

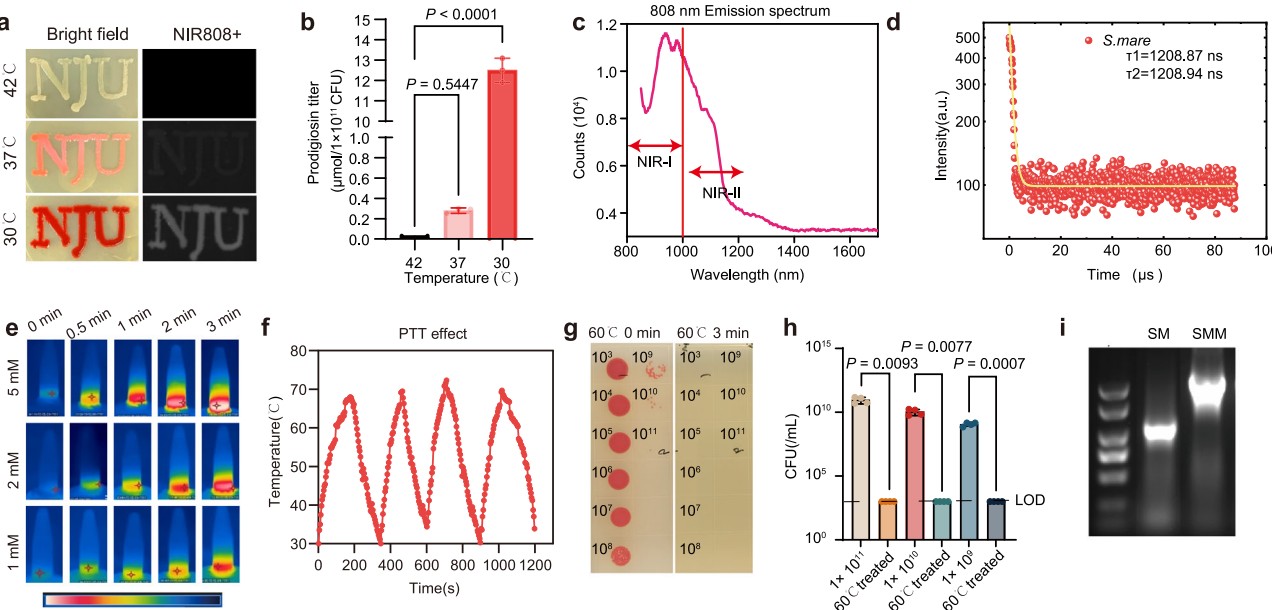

**Fig. 2 | Photophysical, photothermal, and biological characterization of prodigiosin from engineered *S. marcescens* JC11. a** Bright-field and corresponding near-infrared (NIR, 808 nm excitation) images of *S. marcescens* colonies grown at 42, 37, and 30 °C. Similar results were obtained in 3 independent experiments. **b** Prodigiosin titer at 42 °C compared to 30 °C and 37 °C in LB plate. Data represent the mean ± SD of $n = 3$ biologically independent samples. Statistical significance was determined by a two-sided one-way ANOVA followed by Dunnett's post-hoc test. Exact $P$ values are indicated within the figure. **c** NIR emission spectrum of Prodigiosin upon 808 nm excitation, revealing broad emission spanning both NIR-I and NIR-II windows. **d** Time-resolved fluorescence decay curve of Prodigiosin, indicating a long fluorescence lifetime ($\tau\_avg \approx 1.2\,\mu s$). **e** Photothermal heating of bacterial suspensions (equivalent to 1, 2, and 5 mM prodigiosin) under continuous 808 nm laser irradiation (4.50 W/cm²). The thermal images demonstrate a rapid, concentration-dependent temperature increase. Similar results were obtained in 3

independent experiments. **f** Photothermal stability of a purified prodigiosin solution during four on/off cycles of 808 nm laser irradiation, showing excellent durability and consistent heating performance. **g, h** Validation of the thermal killing threshold for *S. marcescens*. **g** Representative images of spot-plated bacteria and **h** Quantitative CFU analysis of *S. marcescens* at different dilutions ($1 \times 10^{11}$, $1 \times 10^{10}$, $1 \times 10^9$ CFU/mL) before and after a 3-min treatment at 60 °C. Data represent the mean ± SD of $n = 4$ biologically independent experiments. Statistical significance was determined by a two-sided unpaired Student's $t$-test. Exact $P$ values are indicated within the figure. **i** Agarose gel confirmation of *msbB* gene knockout. The first lane contains a DNA size marker (DL2000 Plus; bands from top to bottom: 2000, 1500, 1000, 750, 500, 250, and 100 bp). Lane SMM shows the PCR product from the parent strain (SMM), while lane SM confirms the successful gene deletion in the final theranostic strain. Source data are provided as a Source Data file.

potential of biosynthesized prodigiosin, we hypothesize that SMM can serve as a cellular drug-delivery host, delivering prodigiosin specifically to tumor sites and enabling a living theranostic strategy. To establish a robust, dual-layered safety profile for this approach, we first engineered an attenuated strain based on SMM, by deleting the *msbB* gene (hereafter referred to as SM), which is critical for the biosynthesis of endotoxic lipid A; the successful knockout was confirmed by PCR of genome (Fig. 2i). Layered on top of this genetic modification, the photothermal property of the bacteria-produced prodigiosin permits on-demand bacterial ablation via 808 nm near-infrared irradiation, substantially enhancing the overall and controllable safety of this living therapeutic.

### Biohybrid strategies to attenuate *Serratia marcescens* and achieve targeted tumor localization

Building upon the enhanced safety profile of the genetically detoxified SM strain, we introduced multi-functional microencapsulation strategy using a polymeric material (Fig. 3a). This encapsulation serves two key purposes: first, it minimizes systemic exposure and reduces potential immunogenicity by limiting bacterial surface antigen presentation. Second, the polymeric shell provides abundant surface carboxyl groups, enabling efficient amide coupling of targeting peptides and minimizing off-target delivery. Notably, even before *msbB* deletion, SMM does not cause hemolysis on blood agar plates, thus further highlighting its reduced virulence and improved biosafety (Fig. 3b). Like *Escherichia coli Nissle1917* (ECN) and *Salmonella typhimurium* VNP20009 (VNP20009), SM is a facultative anaerobe; however, under strict anaerobic conditions at 37 °C, SM growth is

significantly inhibited (Supplementary Fig. 8). Interestingly, the degree of anaerobic growth inhibition observed for SM was less pronounced than that of ECN, and roughly equivalent to VNP20009 (Supplementary Fig. 8). Deletion of *msbB* (SM) lowered lipopolysaccharide levels by 39.8% relative to the wild-type strain, as confirmed using Limulus amebocyte lysate assays (Fig. 3c). In a murine model, SMM, ECN, and JM109 all exhibited similar high mortality (>50% by day 4) following $1 \times 10^8$ CFU/mice intravenous administration (Fig. 3d). Strikingly, SM showed 100% survival, demonstrating elimination of SMM lethality upon *msbB* deletion and confirming its critical role in virulence (Fig. 3d).

A separate chemical surface engineering platform was then established to improve tumor selectivity (Fig. 3a). Polydopamine (PDA), carboxymethyl cellulose (CMC), and carboxylated chitosan (CMCS) were initially tested as coating materials (Supplementary Fig. 9a–d). PDA was excluded due to potential fluorescence interference, and CMCS posed handling challenges arising from high viscosity. Consequently, CMC was selected for encapsulation (Supplementary Fig. 9a–d). Transmission electron microscopy confirmed a successful coating process, and the 808 nm-induced fluorescence remained intact (Supplementary Fig. 9a). The CMC coating achieved 98.2% efficiency without compromising cell viability (>90% viability; Supplementary Fig. 9). In parallel, the zeta potential decreased by 59.5%, while the particle size peak increased from 615.14 to 712.38 nm (Supplementary Fig. 9b, c). Energy-dispersive X-ray spectroscopy (EDS) further validated Ca²⁺ doping and CMC functionalization (Supplementary Fig. 9d). Therefore, we selected CMC for encapsulation, naming the resulting composite SM@CMC, where the

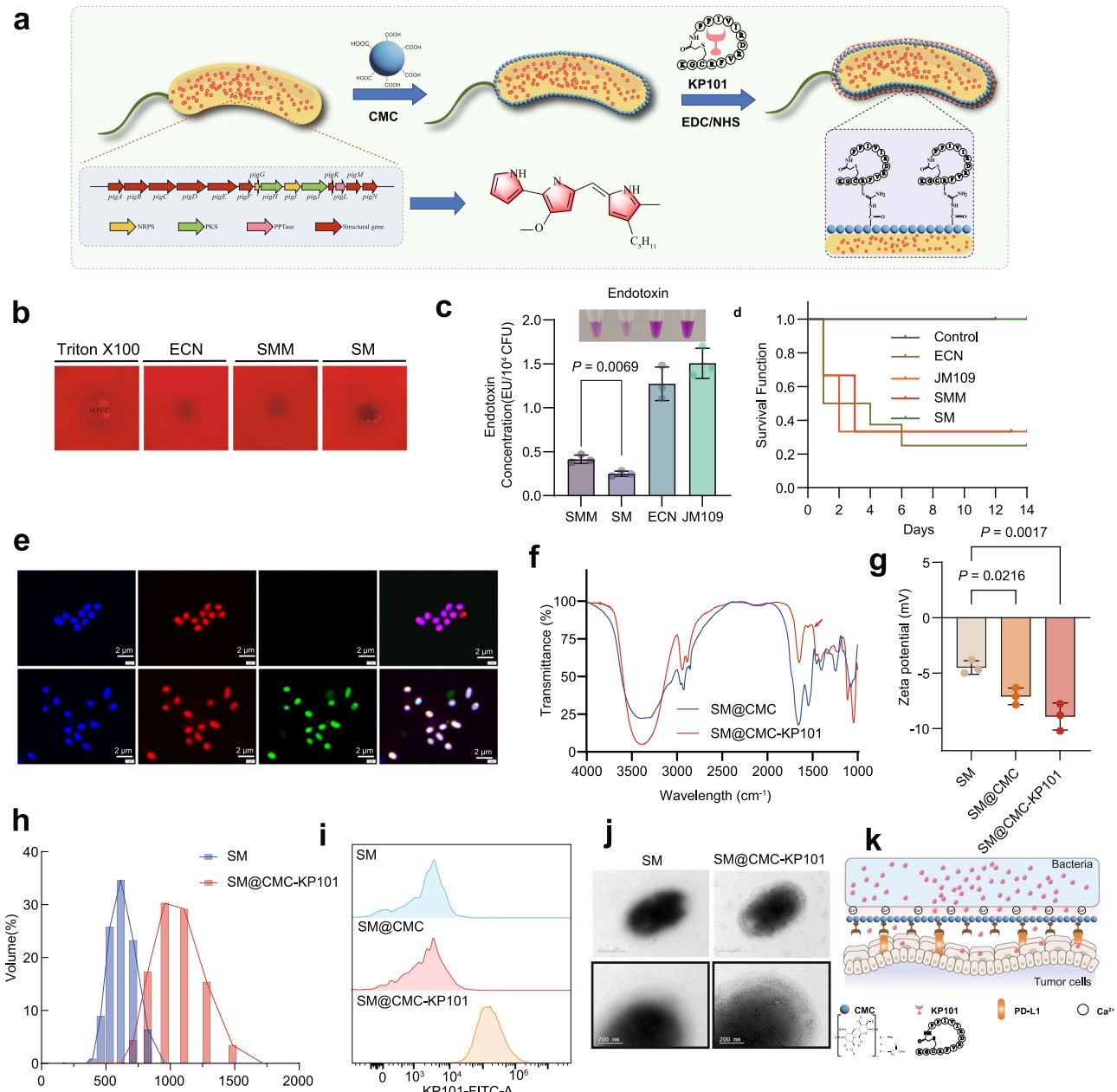

**Fig. 3 | The biohybrid strategy based on engineered *Serratia marcescens*.**
**a** Schematic illustration of the stepwise engineering of *S. mare* with carboxymethyl chitosan (CMC) via EDC/NHS coupling to anchor the small-molecule KP101. Created in BioRender. JI, L. (2026) https://BioRender.com/uma9zrk. All non-BioRender elements in this figure were independently created by the authors. **b** Hemolytic characteristics of different treatment groups under Triton × 100, ECN (*Escherichia coli* Nissle 1917), SMM (*S. mare*), SM (*S. mare* with *msbB* knockout). **c** Comparison of endotoxin levels among SMM, SM, ECN, and JM109. Data are presented as mean ± SD of *n* = 3 biologically independent samples. Statistical significance between SMM and SM was determined by a two-sided unpaired Student's *t*-test. The exact *P* value is indicated within the figure. **d** Survival curves of mice inoculated with SMM and SM mutant on overall survival intravenous injection with $1 \times 10^9$ CFU bacteria. **e** Bright-field and confocal fluorescence images (Rhodamine, FITC, and merged) comparing wild-type *S. mare* and SM@CMC-KP101 functionalized with carboxymethyl chitosan (CMC) or CMC-KP101 (scale bar, 2 μm). Similar results were obtained in 3 independent experiments. **f** FTIR analysis showing the characteristic spectral properties of SM@CMC and SM@CMC-KP101, indicating

successful formulation. **g** Zeta potential measurements of SM, SM@CMC, and SM@CMC-KP101, demonstrating surface charge characteristics. Data represent the mean ± SD of *n* = 3 independent experiments. Statistical significance was determined by a two-sided one-way ANOVA followed by Dunnett's post-hoc test. Exact *P* values are indicated within the figure. **h** Particle size measurements (by volume) comparing SM and SM@CMC-KP101, further confirming nanoparticle formation upon KP101 conjugation. **i** Flow cytometry bar chart comparing fluorescence signals among different treatment groups, illustrating cell-surface modifications induced by SM, SM@CMC, and SM@CMC-KP101. **j** Representative transmission electron microscopy images showing typical bacterial morphology and structural features; the right panel is a higher-magnification view (scale bar, 200 nm). Similar results were obtained in 3 independent experiments. **k** Schematic illustration of the SM@CMC-KP101 system interacting with tumor cells, indicating the role of CMC and KP101 in targeting and inducing anti-tumor effects. Created in BioRender. JI, L. (2026) https://BioRender.com/uma9zrk. All non-BioRender elements in this figure were independently created by the authors. Source data are provided as a Source Data file.

abundant carboxyl groups of CMC provide a rich resource for subsequent modification reactions.

To further enhance tumor targeting, we conjugated the cyclic peptide KP101-which exhibits high affinity for tumor-associated PD-L1 receptors[41]-onto SM@CMC via EDC/NHS coupling, yielding SM@CMC-KP101 biohybrids. FITC-labeled KP101 was employed for fluorescence microscopy and flow cytometry, confirming specific peptide–bacterium interactions (Fig. 3e). Infrared spectroscopy revealed a new characteristic peak at approximately 1500 cm$^{-1}$ in SM@CMC-KP101 (red arrow in Fig. 3f), which is typically assigned to amide groups (e.g., amide II band) or peptide bond vibrations. This finding indicates that KP101 was successfully conjugated to the CMC-modified bacterial surface, providing direct evidence of amide bond formation and thereby validating the construction of SM@CMC-KP101 biohybrids. Following peptide grafting, the particle size increased to 955.4 nm, whereas the zeta potential decreased to −8.917 mV (Fig. 3g, h). Flow cytometry analysis revealed a significant shift in FITC signal for SM@CMC-KP101, indicating successful binding of FITC-KP101 (Fig. 3i). The results of the transmission electron microscope (TEM) reveal that the bacterial surface, after being wrapped and modified with CMC and KP101, is rougher and shows distinct granular modifications, indicating the successful construction of our hybrid structure (Fig. 3j). Furthermore, this biohybrid strategy did not affect bacterial viability, with the survival rate being essentially consistent with that of the unmodified strains (Supplementary Fig. 10). Quantitative analysis revealed that approximately 6 nmol of KP101 were anchored to every $10^8$ colony-forming units (CFU), corresponding to ~$3.6 \times 10^7$ peptides per bacterium[6]. The interaction pattern between bacteria and tumor cells involves the KP101 peptide on the bacterial surface targeting the PD-L1 receptor, which is highly expressed on the surface of tumor cells, allowing for selective binding (Fig. 3k).

## Synergistic in vitro tumor elimination and immune potentiation by SM@CMC-KP101

To evaluate the therapeutic potential of SM@CMC-KP101, we first examined its tumor-killing capacity and immunomodulatory effects in vitro. The intrinsic multi-wavelength fluorescence of prodigiosin (λmax = 592 nm, λex = 434 nm) in SM enabled real-time visualization of cellular uptake without additional labeling (Supplementary Fig. 11). Microscopic evaluation revealed that B16F10 and CT26 tumor cells internalized the bacteria through an endocytic mechanism (Fig. 4a, b and Supplementary Figs. 12 and 13). CCK-8 assays using B16F10 and CT26 tumor cells demonstrated the cytotoxicity of SM@CMC-KP101 (IC$_{50}$ < $4 \times 10^7$ CFU/mL, equivalent to 500 nmol PG) in a concentration-dependent manner (Fig. 4c). Initial experiments using Annexin V/PI staining in B16F10 murine melanoma cells showed an apparent increase in cell death following treatment with prodigiosin (PG) alone and SM@CMC-KP101 (Supplementary Fig. 14). However, given the spectral overlap between prodigiosin's emission (λmax = 592 nm, λex = 434 nm) and the emission spectra of Propidium Iodide (PI), Fluorescein isothiocyanate (FITC) channels, and particularly considering the FITC-like emission of Annexin V, commonly used for apoptosis assays, these results are not conclusive evidence of apoptosis due to potential signal interference (Supplementary Fig. 14). Further mechanistic investigations are detailed in a subsequent section.

To determine the impact of surface modifications on bacterial cell entry, we directly compared the parental strain SMM, the attenuated mutant SM, the polymer-coated SM@CMC, and the KP101-functionalized SM@CMC-KP101 for their internalization efficiency in B16-F10 cells (Fig. 4d). Using an amikacin protection assay, we incubated B16-F10 cells with equal inocula of each strain for 1 h, then removed extracellular bacteria with 50 μg/mL amikacin[42]. After lysing the cells with 0.1% Triton ×100, intracellular colony-forming units (CFU) were enumerated (Fig. 4d). The SM strain showed no significant

difference in internalization compared to the parental SMM strain (both ≈0.09 - 0.10% internalization of the input dose). While CMC coating produced a slight increase, KP101 functionalization significantly enhanced bacterial entry, resulting in ≈0.19% internalization, representing a 107% improvement over SM under identical conditions (Fig. 4d). These results indicate that the enhanced cellular entry is primarily attributable to the KP101 peptide ligand rather than to msbB deletion or CMC coating.

Next, we assessed the immunostimulatory function of SM@CMC-KP101 in bone marrow-derived dendritic cells (BMDCs), which are pivotal antigen-presenting cells (APCs). In bone marrow-derived DC (BMDC) models, co-culture with SM@CMC-KP101 markedly enhanced the proportions of CD11c$^+$CD80$^+$CD86$^+$ subsets (1.08-fold increase vs. PBS treated group) (Fig. 4e). Representative flow cytometric plots of bone marrow-derived dendritic cells (BMDCs) stained for CD11c$^+$, CD80$^+$, and CD86$^+$ after the indicated treatments (Fig. 4e). The data also indicated that SM-EV (23.83%), ECN (26.74%), and PG (20.82%) induced BMDC activation to varying degrees (Fig. 4e). These findings emphasize the capacity of SM@CMC-KP101 to promote DC maturation and improve antigen-presenting activity. Reprogramming tumor-associated macrophages (TAMs) from anti-inflammatory M2 to pro-inflammatory M1 phenotype represents a critical strategy for reversing immunosuppressive microenvironments, we investigated macrophage polarization using Raw264.7 cells. SM@CMC-KP101 treatment significantly increased mRNA levels of M1 markers (TNFα, 62.94-fold; iNOS, 107.3-fold) with comparatively minimal changes in M2 markers (CD206, 3.3-fold; IL-10, 4.6-fold) (Fig. 4f–i). Similar trends were observed in bone marrow-derived macrophages (BMDMs), where SM@CMC-KP101 upregulated M1-associated genes (iNOS, TNFα) while leaving M2 markers (CD206, IL-10) largely unaffected (Supplementary Fig. 15). Collectively, these data highlight the dual tumoricidal and immunostimulatory capabilities of SM@CMC-KP101 in vitro.

We developed a biohybrid drug delivery system loaded with prodigiosin for direct tumor cell killing. Furthermore, this system exhibits multifunctional capabilities (Fig. 4j). Firstly, it significantly enhances dendritic cell (DC) maturation in vitro by increasing the proportion of CD11c$^+$CD80$^+$CD86$^+$ subpopulations (1.8-fold increase compared to controls), thereby promoting antigen cross-presentation. Additionally, our system markedly increases the mRNA expression levels of M1 markers in M2-type macrophages (both the RAW264.7 line and primary BMDMs) by several hundred folds in vitro relative to untreated groups, while maintaining minimal changes in the immunosuppressive cytokine IL-10. This multimodal immunomodulatory activity−combining direct tumor cell killing with reprogramming of both innate and adaptive immunity−positions this platform as a promising bacterial-mediated therapeutic approach for solid tumor eradication. These encouraging in vitro results led us to investigate whether this treatment could subsequently stimulate adaptive immunity in vivo.

## In vivo evaluation of SM@CMC-KP101 in the CT26 tumor model

To ensure the safe and effective use of SM@CMC-KP101, we performed dose-escalation studies via intratumoral (i.t.) and intravenous (i.v.) administration in CT26 tumor-bearing mice (Supplementary Fig. 16a, b). This route-dependent approach determined the maximum tolerated dose of SM@CMC-KP101 to be $1 \times 10^{10}$ CFU/mice (i.t.) and $1 \times 10^8$ CFU/mice (i.v.). Intratumoral injections used concentrations from $1 \times 10^9$ to $1 \times 10^{13}$ CFU/mL (100 μL/mice, equivalent to $1 \times 10^8$–$1 \times 10^{12}$ CFU/mice), while intravenous injections ranged from $1 \times 10^7$ to $1 \times 10^{11}$ CFU/mL (100 μL/mice, equivalent to $1 \times 10^6$–$1 \times 10^{10}$ CFU/mice) (Supplementary Fig. 16a, b). Based on these safety limits for SM@CMC-KP101, we conducted a 11-day therapeutic evaluation, monitoring tumor volume and body weight (Supplementary Fig. 16c–f). The optimal doses for antitumor activity without significant toxicity were

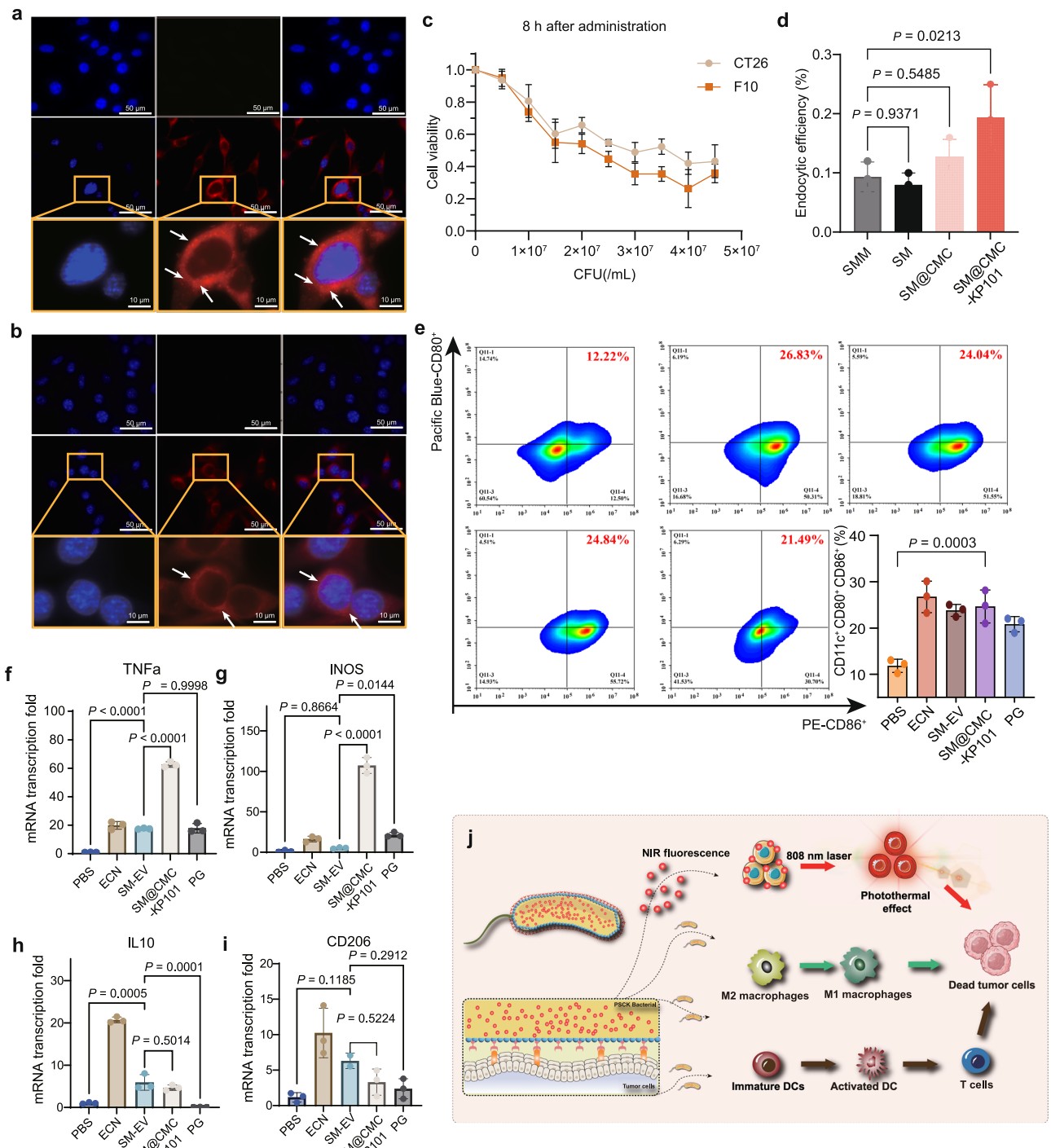

1 × 10^10 CFU/mice (*i.t.*) and 1 × 10^8 CFU/mice (*i.v.*), leading to ≥97% tumor volume reduction.

To ensure the safety of the experiment, we performed intratumoral injections of ECN (1 × 10^10 CFU/mice), VNP20009 (1 × 10^10 CFU/mice), SM-EV (1 × 10^10 CFU/mice), SM@CMC-KP101 (1 × 10^10 CFU/mice) (Fig. 5a). The injection area was illuminated with 4.5 w/cm² of 808 nm near-infrared light for 3 min to reach temperatures of 60 °C or higher. We also intravenously injected SM@CMC-KP101 (1 × 10^8 CFU/mice) in CT26 tumor-bearing mice. First, leveraging the inherent near-infrared (NIR) fluorescence emission of prodigiosin, we conducted in vivo imaging immediately following intratumoral administration of SM@CMC-KP101 (1 × 10^10 CFU/mice). Real-time observation revealed excellent in vivo imaging capabilities (Fig. 5b), demonstrating stable

biodistribution in major organs 24 h post-injection. Notably, both heat-inactivated and live SM@CMC-KP101 exhibited strong tracer effects, highlighting the robust diagnostic potential of our live bacterial drug. However, due to the low intravenous injection dose (1 × 10^8 CFU/mice), we were unable to observe clear bacterial localization via in vivo imaging. Therefore, our current imaging data is presented exclusively for intratumoral injections.

In subsequent therapeutic studies, intratumoral injection of SM@CMC-KP101 followed by 808 nm photothermal therapy achieved effective intratumoral temperatures (≈60 °C) while maintaining skin/normal-tissue temperatures at ≤43–45 °C (Supplementary Fig. 17). Following the treatment, normal organs showed no detectable bacteria at baseline; on day 1 (D1), they were either negative or exhibited

**Fig. 4 | Characterization and in vitro efficacy of the SM@CMC-KP101 delivery system. a, b** Confocal microscopy images showing the internalization of SM@CMC-KP101 by CT26 (**a**) and B16-F10 (**b**) tumor cells. DAPI staining (blue) indicates cell nuclei, and prodigiosin fluorescence (red) indicates the location of the bacteria. White arrows indicate bacteria within the cells. Scale bars: 50 μm (main panels), 10 μm (enlarged insets). Similar results were obtained in 3 independent experiments. **c** in vitro cytotoxicity of SM@CMC-KP101. Cell viability of CT26 and B16-F10 cell lines 8 h post-treatment with varying concentrations of SM@CMC-KP101. Data are presented as mean ± SD ($n = 3$ per group). **d** Quantification of bacterial internalization efficiency in B16-F10 cells. Data represent the percentage of internalized bacteria for SMM, SM, SM@CMC, and SM@CMC-KP101. Data represent the mean ± SD of $n = 3$ independent experiments. Statistical significance was determined by a two-sided one-way ANOVA followed by Dunnett's post-hoc test. Exact $P$ values are indicated within the figure. **e** Flow cytometric analysis of bone marrow-derived dendritic cells (BMDCs) treated with PBS, ECN (*Escherichia coli* Nissle 1917), SM-EV (*Serratia marcescens* JC11 Δ*msbB* cultured under 42 °C,

where the high-temperature environment resulted in the silencing of specific gene clusters, serving as a control for the empty vector), SM@CMC-KP101, or PG to assess BMDC activation (CD11c$^+$ CD80$^+$ CD86$^+$). The bar graph below shows the percentage of CD11c$^+$ CD80$^+$ CD86$^+$ cells for each treatment group. Data represent the mean ± SD of $n = 3$ independent experiments. Statistical significance was determined by a two-sided one-way ANOVA followed by Dunnett's post-hoc test. Exact $P$ values are indicated within the figure. **f–i** Relative mRNA levels of TNFα (**f**), iNOS (**g**), CD206 (**h**), and IL-10 (**i**) in BMDCs treated with PBS, ECN, SM-EV, SM@CMC-KP101 or PG, illustrating the immunomodulatory effects of each intervention. Data represent the mean ± SD of $n = 3$ independent experiments. Statistical significance was determined by a two-sided ordinary one-way ANOVA followed by Tukey's multiple comparisons test. Exact $P$ values are indicated within the figure. **j** Schematic representation of the proposed mechanisms of action of the SM@CMC-KP101 system. Created in BioRender. JI, L. (2026) https://BioRender. com/uma9zrk. All non-BioRender elements in this figure were independently created by the authors. Source data are provided as a Source Data file.

only transient low signals (<$10^2$ CFU/g), and by D14, all organs achieved complete bacterial clearance (Fig. 5c). Within tumors, the mean burden decreased from ≈$1.3 \times 10^9$ CFU/g before treatment to≈$4.25 \times 10^3$ CFU/g on D1, ≈5.5-log reduction, and all tumors were cleared by D14 (Fig. 5c). Concurrently, the robust hyperthermia created three distinct phases of tumor destruction: (i) an acute phase (0–48 h) characterized by visible necrosis and hemorrhagic crust formation; (ii) a transition phase (3–7 d) marked by complete scab detachment with underlying granulation tissue; and (iii) a resolution phase (10–15 d) resulting in full epithelialization without scarring (Fig. 5d). This near-complete bacterial clearance minimized infection risk while leaving behind bacterial debris that potentially activates local immune responses.

To elucidate the impact of SM@CMC-KP101 on the tumor immune microenvironment, we analyzed key immune cell populations. After 24 h, dendritic cells (DCs) isolated from the tumor-draining lymph nodes exhibited enhanced maturation, with the proportion of CD11c$^+$CD80$^+$CD86$^+$ DCs significantly increased in the SM@CMC-KP101 (*i.t.*) (24.6%) and SM@CMC-KP101 (*i.v.*) (16.8%) groups compared to the PBS treated group (9.9%) (Fig. 5e and Supplementary Fig. 19a). Similarly, the proportion of CD3$^+$CD8$^+$ T cells was elevated in the SM@CMC-KP101 (*i.t.*) (26.3%) and SM@CMC-KP101 (*i.v.*) (32.9%) groups relative to controls (21.0%) (Fig. 5f and Supplementary Fig. 19b). Furthermore, bone marrow analysis revealed a significant shift towards M1 macrophage polarization in SM@CMC-KP101-treated mice, characterized by a significantly elevated proportion of M1-like macrophages (CD45$^+$F4/80$^+$CD11b$^+$CD86$^+$) in both the SM@CMC-KP101 (*i.t.*) (20.1%) and SM@CMC-KP101 (*i.v.*) (19.4%) groups, compared to the PBS control group (12.1%) (Fig. 5g and Supplementary Fig. 19c). In contrast, the proportion of M2-like macrophages (CD45$^+$F4/80$^+$CD11b$^+$CD206$^+$) remained unchanged (Fig. 5h and Supplementary Fig. 19c). Splenic analysis in CT26-bearing mice revealed that the proportion of CD3$^+$ CD8$^+$ IFN$^+$ cells significantly increased in both the SM@CMC-KP101(*i.t.*) (16.17%) and SM@CMC-KP101(*i.v.*) (13.4%) groups (Fig. 5i and Supplementary Fig. 19d). Consistent with this enhanced cytotoxic potential, both intratumoral and intravenous photothermal therapy with SM@CMC-KP101 resulted in a preferential expansion of effector memory CD8$^+$ T cells (CD8$^+$ Tem, CD45$^+$CD3$^+$CD8$^+$CD44$^+$CD62L$^-$) to 37.8% and 38.2% (+179.6% and +182.5% vs control, respectively) (Fig. 5j and Supplementary Fig. 19f). The proportion of central memory CD8$^+$ T cells (CD8$^+$Tcm, CD45$^+$CD3$^+$CD8$^+$CD44$^+$CD62L$^+$) remained stable, suggesting a primary impact on the effector arm of the immune response (Supplementary Figs. 19f and 20a). In contrast, the proportion of central memory CD4$^+$ T cells (Fig. 5k, CD4$^+$ Tcm) increased to 17.9% and 17.5% (+63.2% and +59.2% vs. control, respectively) (Supplementary Fig. 19e), while the levels of effector memory CD4$^+$ T cells (CD4$^+$Tem) were unchanged (Supplementary Figs. 19e and 20b). Furthermore, both SM@CMC-KP101(*i.t.*) and SM@CMC-KP101 (*i.v.*) treatments significantly reduced

the proportion of splenic CD4$^+$/CD25$^+$/FoxP3$^+$ Treg cell subpopulations, decreasing from 14.2% in the control group to 8.6% and 8.4%, respectively (Fig. 5l and Supplementary Fig. 21). While ECN, VNP20009, and SM-EV also induced some DC activation and T cell presence, likely due to the inherent immunostimulatory properties of their bacterial components, their lack of prodigiosin loading limited their photothermal capabilities and subsequent therapeutic efficacy, suggesting that SM@CMC-KP101 uniquely combines immune activation with targeted photothermal ablation for enhanced antitumor effects.

Consistent with these immunomodulatory effects, longitudinal tumor volume analysis revealed striking therapeutic efficacy in the SM@CMC-KP101 with 808 nm laser cohort, achieving near-complete tumor regression (15.4 mm³ at day 11 vs 1534 mm³ in controls) accompanied by minor weight fluctuations (Fig. 5m, n and Supplementary Fig. 22). Analysis of the cured mice prior to rechallenge (Day 14) suggested that the potent broad-spectrum cytotoxicity of prodigiosin in SM@CMC-KP101 likely resulted in the collateral clearance of local lymphocytes alongside tumor eradication. To further validate the establishment of long-term immunological memory and functional anti-tumor immunity, we performed a tumor rechallenge assay (Supplementary Fig. 23). Mice from the SM@CMC-KP101 (i.t.+NIR and i.v.) groups that achieved complete regression were rechallenged with a high dose of CT26 cells ($1 \times 10^6$/mice) on Day 14 (Supplementary Fig. 23a). While all naïve control mice (6/6) developed rapid tumors exceeding 200 mm³, only a minority of the cured mice (1/3 in the i.t. group and 1/4 in the i.v. group) showed tumor recurrence (Fig. 5o, p and Supplementary Fig. 23b). Crucially, the long-term landscape at Day 28 revealed a comprehensive memory expansion, characterized by prominent CD8$^+$ central memory (TCM) upregulation alongside elevated CD8$^+$ TEM and CD4$^+$ memory subsets (Supplementary Fig. 23c–f). This shift confirms that SM@CMC-KP101 successfully facilitates the evolution from immediate cytotoxicity to durable, broad-spectrum systemic immunological memory.

To address potential risks associated with bacterial therapeutic approaches, we comprehensively assessed the safety profile of SM@CMC-KP101 through multiple methodological approaches. Serum inflammatory markers, including calreticulin (CRT) and pro-calcitonin (PCT) (Supplementary Fig. 27), remained within safe ranges, similar with those observed with alternative bacterial vectors. Intriguingly, while transient inflammatory responses were detected in initial blood count analyses, these changes were consistent with expected acute immunological reactions and spontaneously normalized within 15 days (Supplementary Fig. 28). Histological examinations of major organs revealed no treatment-related pathological alterations (Supplementary Fig. 29). Furthermore, H&E and Masson's trichrome staining of healed skin after photothermal therapy revealed that, compared to PBS controls, SM@CMC-KP101-treated skin exhibited

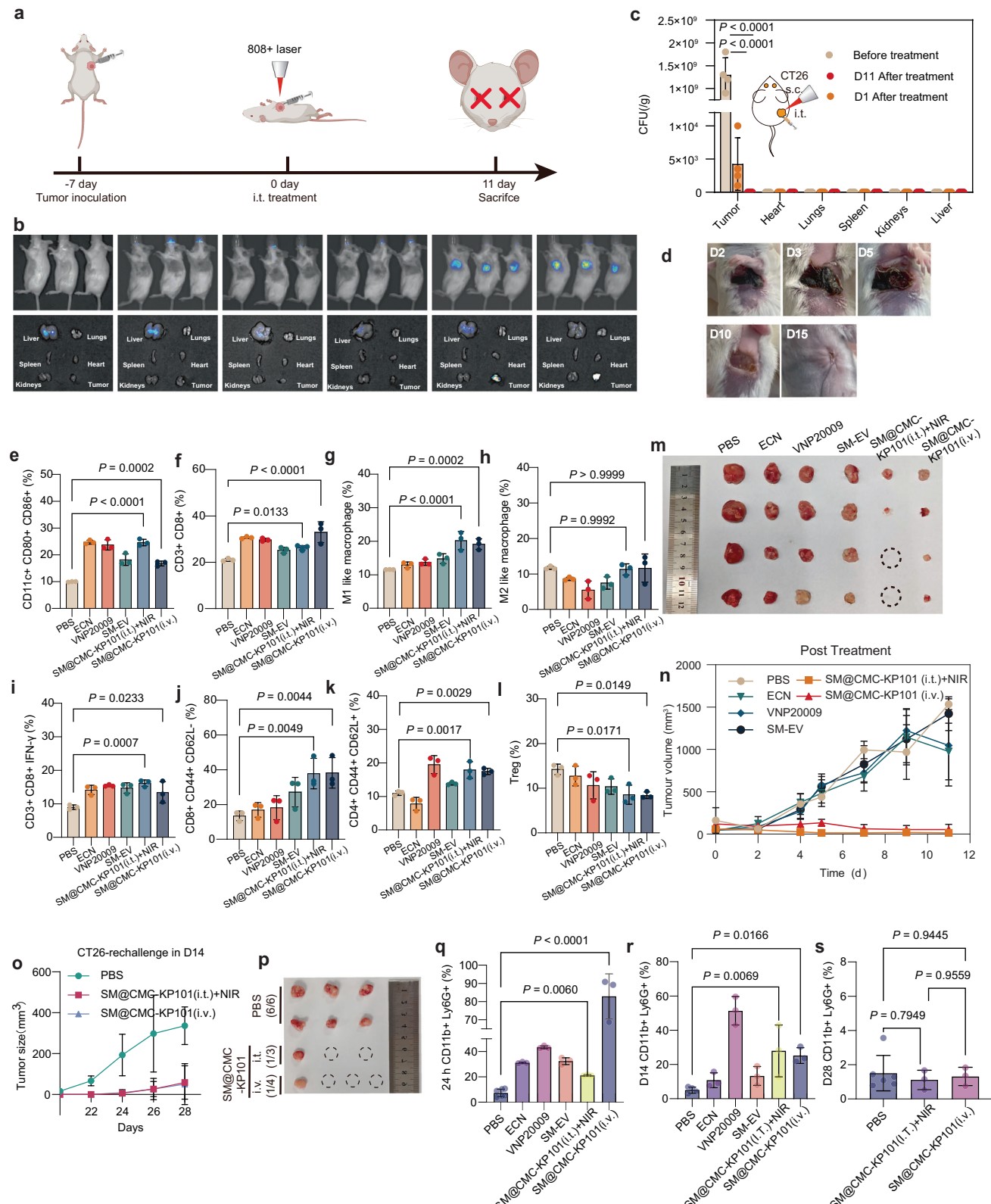

enhanced collagen deposition, maintenance of a continuous epidermis, and preservation of deeper vessel integrity, indicating a limited and self-resolving superficial effect without discernible damage to normal skin (Supplementary Fig. 30). Critically, post-treatment survival rates were excellent, and body weights remained stable throughout the experimental period. To further rigorously exclude the risk of sustained systemic infection or cytokine storm often associated with

bacterial therapies, we additionally monitored the kinetics of neutrophil levels (CD11b⁺ Ly6G⁺) in peripheral blood. Although transient elevation was observed at 24 h post-treatment (consistent with acute immune activation), levels declined significantly by day 14 and returned fully to baseline under normal physiological conditions by day 28 (Fig. 5q–s and Supplementary Fig. 31), with no significant difference compared to the PBS control. This kinetic profile confirms that

**Fig. 5 | Antitumor efficacy of SM@CMC-KP101 in the CT26 tumor model.**
**a** Schematic timeline of tumor inoculation, intratumoral (*i.t.*) treatment on day 0, followed immediately by 808 nm laser irradiation for 3 min, with the endpoint at day 11. Created in BioRender. JI, L. (2026) https://BioRender.com/uma9zrk. All non-BioRender elements in this figure were independently created by the authors.
**b** in vivo imaging of fluorescence in various organs at 0 and 24 h post-treatment with different formulations, indicating the biodistribution of SM@CMC-KP101.
**c** Bacterial counts (CFU/g) in tumor tissue and major organs before and after treatment. Data represent the mean ± SD of *n* = 4 mice per group. Statistical significance was determined by a two-sided one-way ANOVA followed by Dunnett's multiple comparisons test. Exact *P* values are indicated within the figure.
**d** Representative photographs of tumors at days 2, 3, 5, 10, and 15 post-treatment.
**e** Flow cytometric quantification of activated immune subsets in lymph nodes, including CD11c+CD80+CD86+ dendritic cells (DCs) and **f** CD3+CD8+ T lympho-cytes. **g** Quantification of M1-like macrophages (24 h bone marrow-derived), and **h** M2-like macrophages (24 h bone marrow-derived). **i** Frequencies of cytotoxic

T cells (CD8+IFN-γ+) (spleen-derived, day 11), **j** and memory T cells (CD8+CD44+CD62L−) (spleen-derived, day 11). **k** CD4+CD44+CD62L+cells (spleen-derived, day 11), **l** Treg (spleen-derived, day 11). **m** Tumor growth curves for various treatment groups over 11 days. Data represent the mean ± SD of four independent experiments. **n** Representative images of excised tumors from different treatment groups, showing size variations. **o** Tumor growth kinetics following a CT26 rechallenge on day 14 in cured mice after SM@CMC-KP101 treatment, when compared with PBS group. **p** Representative images of tumors harvested at the end-point; red dashed circles denote tumor-free sites, confirming the generation of protective immunological memory. **q–s** The percentage of CD11b+Ly6G+ neu-trophils analyzed by flow cytometry at 24 h (**q**), 14 days (**r**), and 28 days (**s**) post-treatment. All data represent the mean ± SD of *n* = 3 independent experiments. Statistical significance was determined by a two-sided one-way ANOVA followed by Dunnett's post-hoc test. Exact *P* values are indicated within the figure. Source data are provided as a Source Data file.

---

SM@CMC-KP101 elicits a controlled, transient immune response without persistent bacteremia-driven inflammation, supporting the long-term biosafety of this bacterial platform.

Our study demonstrates that SM@CMC-KP101 represents a breakthrough multimodal therapeutic platform that seamlessly integrates precise near-infrared imaging, targeted photothermal ablation, and immunogenic bacterial clearance. By achieving effective tumor eradication with minimal systemic toxicity and simultaneously reshaping the tumor microenvironment, this approach offers a promising strategy for safe and efficient cancer immunotherapy. The synergistic mechanism of SM@CMC-KP101 not only highlights its potential for clinical translation but also provides an effective strategy for precision oncological interventions.

## In vivo evaluation of SM@CMC-KP101 in the B16F10 tumor model

Given that F10 melanoma exhibits higher invasiveness, drug resistance, and immune evasion compared to CT26[43], making it considered a more challenging disease to cure in clinical settings. To demonstrate the versatility of our system and the consistency of therapeutic effects across different tumor models, we conducted animal experiments using B16F10 tumor-bearing mice. Notably, B16F10 melanoma is commonly studied in C57BL/6J mice, which have black fur. Due to the light-absorbing properties of this pigmentation, using 808 nm laser irradiation at 4.5 W−as previously applied in our CT26 model using BALB/c mice−would inevitably induce non-specific photothermal effects. Therefore, for this model, we opted for a purely intravenous approach without laser irradiation.

To evaluate and compare the therapeutic potential of SM@CMC-KP101 against ECN, VNP20009, SMM, SM, and SM@CMC, we performed an efficacy study employing intravenous (*i.v.*) administration in the B16F10 melanoma model. Following the safety dose of $1 \times 10^8$ CFU/mouse established in our prior CT26 studies in CT26 tumor-bearing BALB/c mice (Supplementary Fig. 16), which was confirmed to be both effective and safe. Although most literature reports doses of $1 \times 10^6$ to $1 \times 10^7$ CFU/mouse, lower doses yield limited antitumor effects. Therefore, to enable direct therapeutic comparison, we administered the uniform dosage of $1 \times 10^8$ CFU/mouse. We administered PBS, ECN, VNP20009, SMM, SM, SM@CMC, and SM@CMC-KP101 intravenously at this dosage on days 0, 3, 6, and 10 (Fig. 6a). with mice being euthanized 14 days post-treatment for subsequent analysis. Analysis of the survival curves (Fig. 6b) revealed varying tolerance levels in C57BL/6 mice to the administered bacterial strains at the established dosage of $1 \times 10^8$ CFU/mouse; unfortunately, the ECN-treated group exhibited a 50% mortality rate on the first day post-injection, with complete mortality after the second injection, and the VNP20009 group showed a 25% mortality rate on day one, increasing to 83.3% by day two, also reaching complete mortality after the second

administration (Fig. 6b). Tumor growth was monitored over the 14-day period, and the SM@CMC-KP101 group exhibited a marked reduction in tumor volume compared to the PBS treated groups (Fig. 6c, d and Supplementary Fig. 32). At the study endpoint, the average tumor size in the SM@CMC-KP101-treated group was 138.9 mm3, compared to 511.92 mm$^3$ in the SM-treated group and 555.37 mm$^3$ in the SM@CMC-treated group. While the non-attenuated control strain SMM demonstrated a longer survival period, all mice in this group dead by day six. In contrast, the attenuated SM strain, along with the CMC-modified SM@CMC and SM@CMC-KP101 groups, exhibited excellent survival rates, with no observed mortality during the 14-day treatment period, and these groups showed minimal body weight fluctuations (Fig. 6e). Notably, separate validation experiments showed that single-component treatments (KP101 alone or CMC alone) exhibited negligible anti-tumor efficacy, with tumor growth kinetics indistinguishable from the control group (Supplementary Fig. 33). Using these baselines, we calculated the Coefficient of Drug Interaction (CDI) to quantify the synergy. Relative to KP101, SM@CMC-KP101 exhibited strong synergy (CDI = 0.2528; threshold <0.7), whereas its interaction with CMC was additive (CDI = 1.0046). To further assess the targeting efficiency of SM and its modified versions in comparison to ECN and VNP20009, CFU counts were conducted on homogenized normal and tumor tissues from three mice per group one day following intravenous injection. The results revealed that the attenuated SM strain demonstrated comparable tumor targeting efficiency to both ECN and VNP20009, with a single-dose administration achieving a bacterial load of $1.97 \times 10^7$ CFU/g in the tumor area, while ECN and VNP yielded counts of $2.78 \times 10^7$ CFU/g and $1.96 \times 10^7$ CFU/g, respectively (Fig. 6f). The modified groups, SM@CMC and SM@CMC-KP101, exhibited even higher counts of $2.08 \times 10^7$ CFU/g and $8.98 \times 10^7$ CFU/g, respectively. These findings suggest that the SM strain effectively targets tumors, with SM@CMC-KP101 showing the highest targeting efficiency (Fig. 6f). Additionally, the SM strain and its modified derivatives showed a trend toward lower residual rates in normal tissues, particularly the heart and liver, compared to ECN and VNP20009, although these differences were not statistically significant (Fig. 6f).

Flow cytometric analysis was performed to evaluate the impact of SM@CMC-KP101 on immune cell populations. After 24 h, dendritic cells (DCs) were isolated from the tumor-draining lymph nodes, with their maturation markers CD80+ and CD86+ assessed. All experimental groups (PBS, ECN, VNP20009, SMM, SM, SM@CMC, and SM@CMC-KP101) showed a significant increase in the proportion of CD11c + CD80+ CD86+ dendritic cell (Fig. 6g and Supplementary Fig. 34a), CD69+ CD8+ T cell (Fig. 6g and Supplementary Fig. 34b), and CD69+ CD4+ T cell (Fig. 6g and Supplementary Fig. 34c) subsets in the tumor-draining lymph nodes 24 h post-treatment, resulting from the exposure to bacterial surface antigens. This indicates that the developed SM@CMC-KP101 treatment system, similar to the established

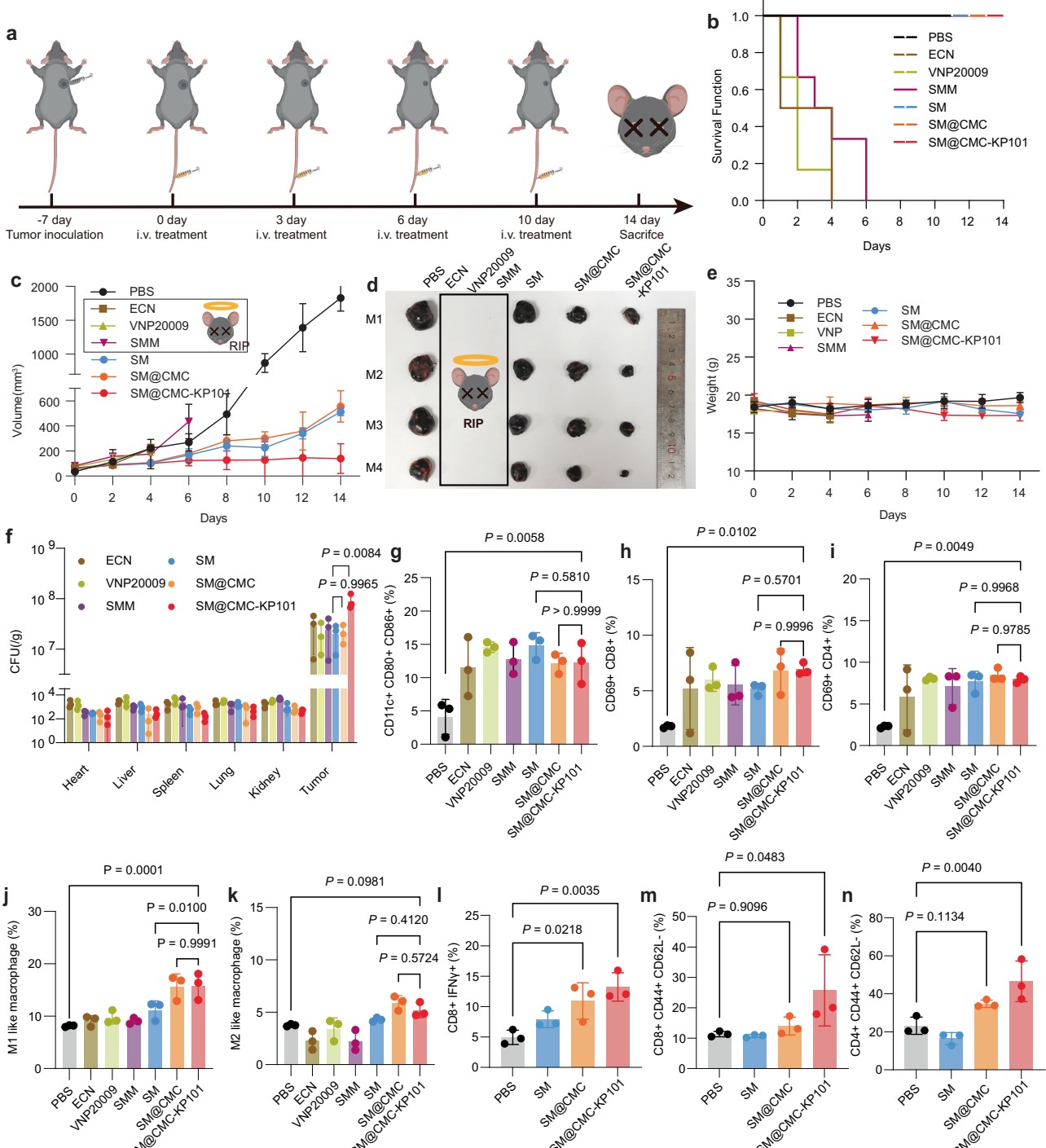

**Fig. 6 | Efficacy of SM@CMC-KP101 in a B16F10 melanoma model via intravenous administration. a** Schematic illustration of the experimental timeline depicting tumor inoculation on day-7, followed by intravenous (*i.v.*) administration of treatments on days 0, 3, 6, and 10, with sacrifice on day 14. Created in BioRender. JI, L. (2026) https://BioRender.com/uma9zrk. All non-BioRender elements in this figure were independently created by the authors. **b** The survival curves illustrating the survival rates of mice (*n* = 6) treated with PBS, ECN, VNP20009, SMM, SM, SM@CMC, and SM@CMC-KP101. **c** Tumor growth curves showing tumor volume (mm³) over time for each treatment group. Data represent the mean ± SD of four mice per group. **d** Representative photographs of excised tumors from each treatment group at day 14. **e** Mouse body weight (**g**) for each treatment group over the course of the experiment. Data represent the mean ± SD of four mice per group.

**f** Bacterial counts (CFU/g) in various organs (heart, liver, spleen, lung, kidney, and tumor) following treatment. Flow cytometric analysis of immune cell populations in tumor microenvironment: **g** Quantification of CD11c⁺CD80⁺CD86⁺ dendritic cells (DCs). **h** Quantification of CD69⁺ CD8⁺ T lymphocytes. **i** Quantification of CD69⁺ CD4⁺ T lymphocytes. **j** Quantification of M1-like macrophages. **k** Quantification of M2-like macrophages. **l** Frequencies of cytotoxic T cells (CD8⁺IFN-γ⁺). **m** Statistical percentage of effector memory T cells (CD8⁺CD44⁺CD62L⁻). **n** Statistical percentage of effector memory T cells CD4⁺CD44 ⁺ CD62L⁻ cells. All data represent the mean ± SD of *n* = 3 biologically independent experiments. Statistical significance was determined by a two-sided one-way ANOVA with Dunnett's (**f** and **l–n**) or Tukey's (**g–k**) post-hoc test. Exact *P* values are indicated within the figure. Source data are provided as a Source Data file.

ECN and VNP systems, promoted antigen presentation and T-cell activation. Specifically, the SM@CMC-KP101 group displayed CD11c⁺CD80⁺CD86⁺ dendritic cell, CD69⁺CD8⁺, and CD69⁺CD4⁺ cell subset proportions of 12.25, 6.93, and 8.32%, respectively, which were significantly higher than the PBS treated group proportions of 4.09, 1.78, and 2.27%, respectively (Fig. 6g–i). Bone marrow analyses at 24 h revealed a significant rise in CD45⁺CD11b⁺F4/80⁺CD86⁺ myeloid cells, indicating a shift toward a more proinflammatory M1-like phenotype. Specifically, the percentage of M1-like macrophages (CD11b⁺F4/80⁺CD86⁺) in the SM@CMC-KP101 group increased to 15.85%, compared to 8.15% in the PBS control group (Fig. 6j and Supplementary Fig. 34d). In comparison, the ECN and VNP20009 groups showed M1-like macrophage percentages of 9.13% and 9.74%, respectively, indicating no significant activation (Fig. 6j and Supplementary Fig. 34d). Conversely, the proportions of CD11b⁺F4/80⁺CD206⁺ cells (M2-like macrophages) showed no statistically significant difference among all groups (Fig. 6k and Supplementary Fig. 34d). These results suggest that the treatment primarily promotes M1 macrophage polarization in the bone marrow early on, with SM@CMC-KP101 inducing a slightly stronger effect than ECN and VNP20009, while exerting no statistically significant effect on M2 macrophage populations.

After observing the short-term immune activation, we shifted our focus to the long-term effects at 14 days post-treatment. Due to the complete mortality observed in the ECN, VNP, and SMM groups, subsequent analyses at 14 days were limited to the PBS, SM, SM@CMC, and SM@CMC-KP101 groups. The proportion of CD8 + T cells expressing IFNγ (CD3⁺CD8⁺IFNγ⁺) increased significantly in the SM@CMC-KP101 group (13.24%), compared to the PBS control group (4.96%), with the SM@CMC group also showing an increase (10.91%) (Fig. 6l and Supplementary Fig. 34e). Moreover, assessment of memory T cell subsets demonstrated a notable expansion of CD45⁺CD3⁺CD8⁺CD44⁺CD62L⁻ effector memory T cells (CD8⁺ Tem) in the SM@CMC-KP101 group (25.73%), compared to the PBS control group (11.37%) (Fig. 6m; Supplementary Fig. 34g). In B16F10 mice receiving intravenous SM@CMC-KP101, CD8⁺ T effector memory (Tem) cells showed a remarkable activation increase of 126.4% compared to the control group. This was accompanied by increases in central memory (Tcm) CD8⁺ T cells (56.9%) (Supplementary Figs. 34g and 35a), CD4⁺ Tcm cells (29.5%) (Supplementary Figs. 34f and 35b), and CD4⁺ Tem cells (100.9%) (Fig. 6n and Supplementary Fig. 34f).

In summary, the SM@CMC-KP101 treatment promotes comprehensive immune responses characterized by the maturation of dendritic cells, robust activation of T cells, and M1 macrophage polarization. Notably, the treatment resulted in a significant increase in the proportion of splenic CD8⁺ IFNγ⁺T cell subsets, alongside enhanced populations of effector memory T cells (CD8⁺ Tem) and central memory T cells (CD4⁺ Tem). Importantly, hematoxylin and eosin (H&E) staining revealed no obvious tissue damage, indicating the excellent biosafety profile of this approach (Supplementary Fig. 36). Together, these findings underscore the therapeutic potential of SM@CMC-KP101 in eliciting a sustained anti-tumor immune response and improving long-term tumor control in the B16F10 melanoma model.

## Mechanistic insights into the antitumor effects of SM@CMC-KP101

To investigate the mechanisms of SM@CMC-KP101-induced tumor cell killing, we performed proteomic analysis and various cellular and molecular assays in B16F10 tumor cells. SM@CMC-KP101 treatment induces significant changes in protein expression, activates a combination of cell death pathways, and ultimately leading to tumor cell killing through a multi-faceted mechanism. Proteomic analysis revealed that SM@CMC-KP101 treatment resulted in the upregulation of 152 proteins and downregulation of 89 proteins (Fig. 7a,

Supplementary Figs. 37–40, and Supplementary Data 1). Volcano plot analysis of proteomic data (Fig. 7a and Supplementary Data 1) showed the overall changes in protein expression and revealed that MAP kinase-activated protein kinase 5 (Mapkapk5, MK5), a key regulator of the HSP27 phosphorylation involved in cell survival and stress response[44,45], was the most significantly downregulated protein, with a decrease of 8.98-fold. In addition, proteomic analysis identified significant upregulation of ROS synthesis pathways (Maff 5.5-fold upregulated; Mafk 2.1-fold upregulated) and oxidative stress markers (Fos 62.1-fold upregulated; Jun 2.49-fold upregulated), prompting us to validate intracellular ROS levels. Due to the fluorescence interference between the drug and the FITC channel, we used the CellROX Deep Red fluorescent dye to quantify ROS production levels. Results from fluorescence microscopy showed a stronger generation of ROS (Supplementary Fig. 41). Furthermore, it revealed lysosomal dysfunction markers (Ctsz: 61.5% decrease, Npc2: 60.1% decrease, Ctsh: 69.1% decrease), suggesting disruption of lysosomal function. Kyoto Encyclopedia of Genes and Genomes (KEGG) pathway enrichment analysis (Fig. 7b) revealed significant activation of pathways related to autophagy, mitophagy, and necroptosis. Specifically, mitophagy and autophagy-related proteins such as Transcription factor Jun (Jun), Microtubule-associated protein 1 light chain 3 alpha (Map1lc3a), and Microtubule-associated protein 1 light chain 3 beta (Map1lc3b) were significantly upregulated. Additionally, proteins associated with necroptosis, including Interferon gamma receptor 1 (Ifngr1), Tumor necrosis factor receptor superfamily member 1A (Tnfrsf1a), and various components of the CHMP family, were also markedly increased (Fig. 7b). Cell viability assay (Fig. 7c and Supplementary Fig. 42) demonstrated that autophagy inhibitor (Chloroquine), apoptosis inhibitor (Z-VAD-FMK), necroptosis inhibitor (Nec-1), and ROS scavenger (Trolox) rescued cells from SM@CMC-KP101-induced cell death both in B16-F10 and CT26 cell lines. Specifically, the necroptosis inhibitor Nec-1 increased the survival rate of F10 cells from 28% to 94%, and CT26 cells from 8% to 59%. Additionally, the apoptosis inhibitor Z-VAD raised the survival rate of F10 cells from 28% to 100% and that of CT26 cells from 8% to 72%, indicating the occurrence of apoptosis and necroptosis (Fig. 7c). In addition, western blot analyses showed the effects of SM@CMC-KP101 and PG on proteins involved in mitophagy, autophagy, and necroptosis, which aligned with the proteomic findings (Fig. 7b and d). Specifically, SM@CMC-KP101 and PG treatment increased the levels of PINK1 and LC3-II/I, while decreasing p62 and CTSB, indicating the induction of mitophagy and autophagic flux. Furthermore, SM@CMC-KP101 and PG treatment increased the levels of phosphorylated mixed lineage kinase domain-like protein (p-MLKL), a marker of necroptosis (Fig. 7d). Although caspase-3 and cleaved caspase-3 show a decreasing trend, suggesting that caspase-dependent apoptosis may not be the dominant cell death mechanism, the overall impact of apoptosis should not be excluded (Fig. 7d). Consistent with these findings, transmission electron microscopy (TEM) revealed swollen mitochondria and a significant presence of vacuoles within tumor cells, along with evidence of impaired mitophagy after SM@CMC-KP101 treatment (Fig. 7e). Collectively, these data indicate that SM@CMC-KP101 treatment induces mitophagy and ROS production, activates stress-related pathways, and ultimately leads to cell death via necroptosis, with a potential contribution from apoptosis.

Given the most significant downregulation of Mapkapk5 observed in our proteomic analysis, and the activation of this stress response pathway by SM@CMC-KP101, we next sought to investigate the specific role of Mapkapk5 in SM@CMC-KP101-induced cell death. We designed three siRNAs targeting different Mapkapk5 gene sites (Supplementary Table 6). qPCR analysis confirmed that transfection with each of the three Mapkapk5 siRNAs resulted in over 80% silencing efficiency of MAPK5 mRNA expression (Supplementary Fig. 43). CCK8 assay (Fig. 7f) showed that knockdown of Mapkapk5 significantly

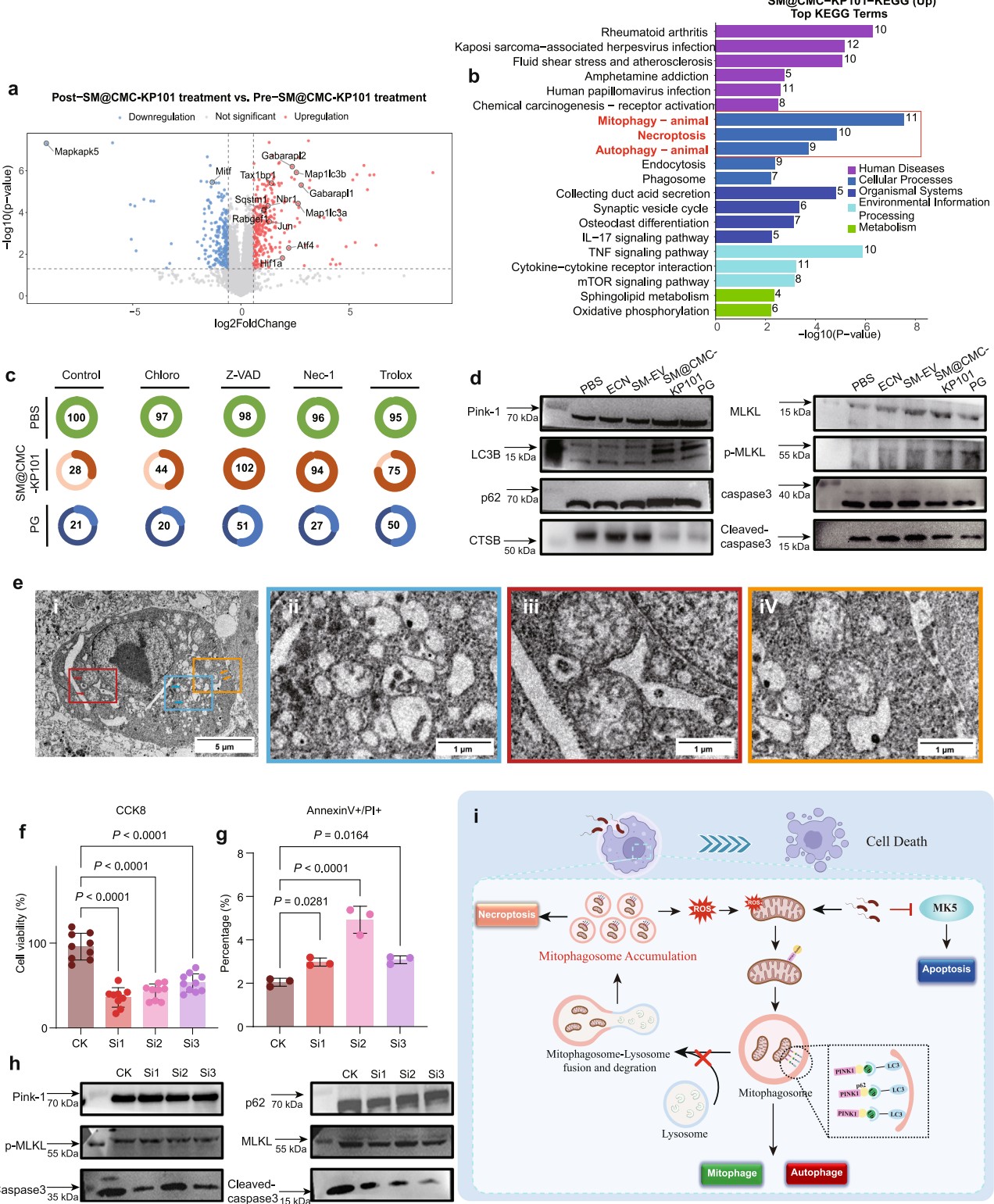

decreased cell viability to 35.89, 42.44, and 53.10% of WT, respectively, suggesting that Mapkapk5 expression is critical for cell survival. To further elucidate the mechanism by which Mapkapk5 regulates SM@CMC-KP101-induced cell death, we further assessed the impact of Mapkapk5 knockdown on cell apoptosis. Quantification of Annexin V/PI positive cells (Fig. 6g and Supplementary Fig. 44) revealed that Mapkapk5 knockdown increased the percentage of Annexin V/PI double-positive cells from 2.047% to 2.977%, 4.927%, and 3.083%, respectively, indicating an increase in cell apoptosis. Western blot

analyses (Fig. 7h) showed that MK5 knockdown did not significantly alter the levels of mitophagy markers (p62, PINK1) or necroptosis markers (p-MLKL, MLKL) (Fig. 7h), suggesting that MK5 knockdown is not linked to mitophagy or necroptosis. However, Western blot analysis revealed a surprising result: the levels of caspase-3 and cleaved caspase-3 were downregulated following MK5 knockdown (Fig. 7h), contrary to what is typically observed during classical apoptosis[46,47]. Intriguingly, MK5 knockdown resulted in the same phenotype as SM@CMC-KP101 and PG treatment, where caspase-3 and cleaved

**Fig. 7 | Mechanism study based on proteomic analysis of B16f10 administered with SM@CMC-KP101. a** Volcano plot of differentially expressed genes comparing SM@CMC-KP101 vs PBS treated groups, with significantly upregulated (red) and downregulated (blue) transcripts labeled for key factors involved in ROS production. **b** KEGG pathway enrichment analysis of differentially expressed proteins, highlighting pathways related to mitophagy, necroptosis, and autophagy. **c** Cell viability assay showing the rescue effect of autophagy inhibitor (Chloroquine), apoptosis inhibitor (Z-VAD-FMK), necroptosis inhibitor (Nec-1), and ROS scavenger (Trolox) on SM@CMC-KP101-induced cell death. **d** Western blot analyses of PINK-1, LC3-I/II, p62, and CTSB, caspase3, cleaved caspase3, MLKL, and p-MLKL in cells treated with PBS, ECN, SM-EV, SM@CMC-KP101, or PG, revealing activation of mitophagy-and stress-related pathways. β-Actin serves as the loading control. Uncropped blots are available in Source Data file. **e** Transmission electron microscopy (TEM) images showing morphological changes in B16F10 cells, including swollen mitochondria (i & ii), increased vacuolization (iii), and impaired mitophagy (iv) after SM@CMC-KP101 treatment. Similar results were obtained in 3 independent experiments. **f** CCK8 assay showing cell viability of B16F10 cells transfected with MK5 siRNAs (Si1, Si2, Si3) or treated with equal volumes of transfection reagent (WT). Data represent the mean ± SD of $n = 10$ biologically independent experiments. Statistical significance was determined by a two-sided one-way ANOVA followed by Dunnett's post-hoc test. Exact $P$ values are indicated within the figure. **g** Quantification of Annexin V/PI staining of B16F10 cells transfected with MK5 siRNAs (Si1, Si2, Si3) or treated with equal volumes of transfection reagent (WT). Data represent the mean ± SD of $n = 3$ biologically independent experiments. Statistical significance was determined by a two-sided one-way ANOVA followed by Dunnett's post-hoc test. Exact $P$ values are indicated within the figure. **h** B16F10 cells were transfected with MK5 siRNAs (Si1, Si2, Si3) or treated with equal volumes of transfection reagent (WT). Protein expression was assessed by Western blot using antibodies against the indicated proteins: Pink1, p62, p-MLKL, MLKL, Caspase3, and Cleaved-Caspase3. Uncropped blots are available in Source Data file. **i** Schematic diagram illustrating the proposed mechanism by which SM@CMC-KP101 induces tumor cell death. Created in BioRender. JI, L. (2026) https://BioRender.com/uma9zrk. All non-BioRender elements in this figure were independently created by the authors. Source data are provided as a Source Data file.

caspase-3 were also downregulated. This suggests that MAPKAPK5 downregulation induces cell death via a caspase-3-independent mechanism.

In summary, SM@CMC-KP101 exerts its cytotoxic effects on B16F10 tumor cells through a multifaceted mechanism (Fig. 7i). First, SM@CMC-KP101 may induce mitochondrial dysfunction, leading to the generation of reactive oxygen species (ROS). In response, cells attempt to eliminate damaged mitochondria through mitophagy; however, SM@CMC-KP101 appears to interfere with the fusion and degradation of mitophagosomes in lysosomes, resulting in the accumulation of mitophagosomes. Proteomic analysis revealed a significant downregulation of Mapkapk5, which thereby induces caspase-3-independent apoptosis and mirrors the phenotype observed with SM@CMC-KP101 treatment. Collectively, these results suggest that SM@CMC-KP101 induces tumor cell death via a combination of mitophagy-driven necroptosis and caspase-3-independent apoptosis, with Mapkapk5 downregulation playing a critical role.

## Discussion

In this work, we explored an underutilized but promising bacterial chassis—*Serratia marcescens*—to address key challenges in bacterial cancer therapy: limited chassis options, the need for exogenous drug modification, insufficient direct tumor-killing, and safety concerns. The high levels of prodigiosin biosynthesis (8.3 ± 0.5 g/L) highlight *S. marcescens*' robust metabolic capacity, positioning it as a viable alternative to traditional platforms such as *Salmonella typhimurium* VNP20009 and *Escherichia coli* Nissle1917, which have primarily focused on tumor targeting and immunostimulation but rely on exogenous modifications for therapeutic loading[6,8,15]. Unlike these platforms and classic oncolytic bacteria that depend on intratumoral proliferation, our system uses *S. marcescens* as a controllable drug delivery vehicle. This enhances both safety and control through predictable dosing via pre-loaded prodigiosin and an external trigger, and by mitigating risks of inflammation and systemic infection with reduced anaerobic growth and a photothermal clearance mechanism. This controlled approach allows efficient drug delivery while minimizing off-target effects.

Here, we address these limitations by developing an attenuated *Serratia marcescens* platform that leverages intrinsic biosynthetic pathways for the high-yield production of prodigiosin. Prodigiosin, which we report here as a biosynthesized near-infrared fluorescent probe, endows our SM@CMC-KP101 system with both photothermal capabilities and near-infrared tracking functionality, positioning it as a theranostic live drug carrier. The 808 nm laser, operating within the NIR-I window, allows for enhanced penetration (4–6 mm) compared to visible light[48–51]. Prodigiosin, produced in situ by our engineered bacteria, acts as a fully biosynthesized NIR photothermal agent. While recognizing its limitations—photodegradation and lower efficiency compared to agents like indocyanine green—we are exploring chemical modifications to improve its properties. Despite these, in situ prodigiosin generation, coupled with 808 nm laser penetration, offers a promising PTT strategy. Furthermore, demonstrating a significant amplification effect, scaling to a 5 L fed-batch fermenter yielded $5 \times 10^{15}$ CFU per batch, demonstrating a scalable pathway towards clinical-grade GMP manufacturing.

SM@CMC-KP101 demonstrates significant versatility and efficacy as an antitumor therapeutic. Both intratumoral injection followed by photothermal therapy and systemic intravenous administration of SM@CMC-KP101 result in potent antitumor effects. This is coupled with a pronounced immunostimulatory response, characterized by enhanced dendritic cell maturation, robust activation of CD8$^+$ T cells, systemic neutrophil expansion, and reprogramming of tumor-associated myeloid cells towards an M1 phenotype. This ability to elicit a strong systemic immune response and establish functional immunological memory, regardless of the administration route, underscores the potential of SM@CMC-KP101 for broad clinical application. Further mechanistic investigation reveals that SM@CMC-KP101 possesses a potent ability to reprogram the tumor microenvironment and directly kill tumor cells. Specifically, SM@CMC-KP101 induces both mitophagy-driven necroptosis and caspase-3-independent apoptosis in tumor cells, enhancing direct cytotoxic effects. This multifaceted cell death induction, driven in part by Mapkapk5 downregulation and interference with mitophagosome degradation, augments overall antitumor efficacy. These cytotoxic effects, combined with the observed immune modulation, highlight the synergistic nature of SM@CMC-KP101, positioning it as a promising candidate for cancer immunotherapy through both direct tumor cell killing and immune-mediated mechanisms.

In conclusion, our biohybrid bacterial drug delivery platform offers an advancement in precision cancer therapy, combining potent anticancer efficacy with immunomodulation while addressing safety concerns via rapid photothermal-mediated bacterial clearance. The results of this study not only demonstrate the versatility of *Serratia marcescens* for therapeutic applications but also encourage further exploration into the development of other biosynthetic platforms that can integrate natural products with tailored engineering approaches. Such advancements could play a crucial role in advancing the next generation of bacterial therapies, making them safer, more targeted, and significantly more efficient for treating various malignancies.

## Methods

### Ethical statement

All animal experiments were performed in strict accordance with the National Institutes of Health (NIH) guidelines for the care and use of

laboratory animals. The animal study protocols were reviewed and approved by the Institutional Animal Care and Use Committee (IACUC) of Nanjing University (approval numbers: IACUC-D2202146).

Mice were housed in a specific pathogen-free facility under controlled conditions: a 12-h light/dark cycle (lights on from 8:00 a.m. to 8:00 p.m.), an ambient temperature of 18–22 °C, and 50–60% humidity, with unrestricted access to standard chow and water. The maximum permitted tumor volume was 2000 mm³, and this limit was strictly adhered to throughout all experiments. Mice were humanely euthanized if their tumor volume approached or exceeded this limit, or upon reaching the predetermined experimental endpoint of 14 days.

## Materials

Prodigiosin standard was purchased from MedChemExpress (Monmouth Junction, USA). Cell Counting Kit-8 (CCK-8), Annexin V-FITC/PI apoptosis detection kit, and Yopro-1-V-FITC/PI apoptosis detection kit were obtained from KeyGEN Biotechnology (Nanjing, China). Fluorochrome-conjugated antibodies, including BV480 Rat Anti-Mouse CD45 (#752417), FITC Rat Anti-CD11b(M1/70) (#557396), APC Rat Anti-Mouse CD4(RM4-5) (#553051), Ms CD8a PE 53-6.7 100ug (#553032), Ms CD44 BV421 IM7 50ug (#563970), and PE-Cy7 Rat Anti-Mouse CD62L(MEL-14) (#560516) from BD Biosciences; FIXABLE VIABILITY DYE EF780 (#65-0865-14), and CD206 (MMR) Monoclonal Antibody (MR6F3), PE (#12-2061-82) from eBioscience; Fc Receptor Blocking Solution, Mouse (#abs9477-200T) from Absin; and FITC anti-mouse FOXP3 (#11-5773-82) and anti-mouse CD25-APC (#17-0251-81) from Invitrogen. Finally, CD3-FITC (#100204), CD4-PE (#100512), CD8a-Pacific Blue (#100725), CD11c-APC (#117310), CD80-FITC (#104705), CD86-Pacific Blue (#105022), IFN-γ-APC (#505810), and CD69-FITC (#104505) were acquired from BioLegend (San Diego, USA). The following primary antibodies were used for Western blotting: Pink1 (Abmart, PK05715S), Cathepsin D (Beyotime, AF1645), β-Actin (Beyotime, AF2815), LC3B (Beyotime, AB2023), Hsp27 (Beyotime, AF0183), p-Hsp27(S82) (Cell Signaling Technology, 9709), p38 (Beyotime, AF1111), p-p38 (Cell Signaling Technology, 4511T), caspase3 (Abclonal, A19664), cleaved-caspase3 (Abmart, TA7022), Phospho-MLKL(Ser358) (Cell Signaling Technology, #37333), MLKL (Cell Signaling Technology, #37705), RIPK3 (Abclonal, A5431), and p-RIPK3 (Abclonal, AP1260). All antibodies were used according to manufacturer's recommendations and listed in Supplementary Tables 3 and 4. The Chromogenic LAL Endotoxin Assay Kit was purchased from Beyotime Biotechnology (Shanghai, China). KP101 peptide (sequence: FFIVIRDRVFRCG) was custom synthesized by Hefei Peptide Library Co., Ltd (Hefei, China). CellROX™ Deep Red fluorescent probe and TCEP hydrochloride were obtained from Thermo Fisher Scientific (Waltham, USA) and Sigma-Aldrich (St. Louis, USA), respectively.

## Development of high-prodigiosin producing *Serratia marcescens* strains via iterative UV mutagenesis and stepwise screening

The wild-type *Serratia marcescens* strain NRRL B-1481 was purchased from the American Type Culture Collection (ATCC; Catalog No. 14041). UV mutagenesis and iterative screening[52] were performed to obtain high-prodigiosin (PG)–producing *Serratia marcescens* mutants. Fresh colonies were inoculated into LB broth (10 g/L tryptone, 5 g/L yeast extract, 10 g/L NaCl) and cultured at 30 °C to late exponential phase (~16 h). Cells were harvested, washed twice with sterile saline, and resuspended to OD600 = 1.0. To establish the lethal curve and define the working dose, 10-μL aliquots were spread on sterile Petri dishes (lids open) and irradiated under the germicidal UV lamp of a laminar-flow hood for 0, 10, 30, 50, 70, or 100 s. Immediately after irradiation, suspensions were serially diluted ($10^3$–$10^5$) in saline, 100 μL was plated on LB agar, and plates were incubated at 30 °C for 16 h protected from light. Lethality was calculated as $1 - N/N0$ based on

colony-forming units (N0, non-irradiated control), and the exposure giving ≥80% lethality (approximately 10 s) was used in all subsequent rounds. For round-1 mutagenesis, UV-treated cultures were plated, incubated in the dark, and intensely red colonies were picked for primary screening in 24-well plates (1 mL fermentation medium per well; 30 °C, 200 rpm, 30 h). The fermentation medium contained (per liter) sucrose $20 \times g$, CaCl₂ $3 \times g$, peptone $10 \times g$, and glutamic acid $2 \times g$; the same medium was used for plate-based screening. After fermentation, 2 mL acidified methanol (adjusted to pH 3.0 with HCl) was added to each culture, samples were sonicated for 20 min, and 200 μL of the extract was transferred to 96-well plates for spectrophotometric quantification at 535 nm. PG concentrations were calculated from a calibration curve prepared with an authentic prodigiosin standard (MedChemExpress, MCE) and converted to titers (g/L). Top producers from the 24-well screen were re-screened in 250-mL Erlenmeyer flasks containing 50 mL of fermentation medium (30 °C, 200 rpm, 30 h) to confirm production and stability; promising isolates were stored as 20% glycerol stocks at −80 °C. The best-producing isolate from round 1 was used as the parental strain for round 2, and the top strain from round 2 served as the parent for round 3, repeating the same workflow (dose confirmation−UV mutagenesis−plate selection−24-well screening−flask re-screening−quantification). This stepwise strategy yielded the final high-prodigiosin producing strain SM.

## Cell lines

The B16F10 melanoma cells (Cat# IM-M002) were purchased from IMMOCELL (Xiamen, Fujian, China) and CT26 colon carcinoma (ATCC CRL-2638) authenticated cell lines were purchased from ATCC. RAW264.7 (ATCC TIB-71) were purchased from ATCC. Cells were confirmed mycoplasma-free. Cells were cultured in incubators at 37 °C with an atmosphere of humidified 5% CO₂. B16F10 and CT26 cells were grown in DMEM supplemented with 10% (vol/vol) fetal bovine serum (FBS), 1 × GlutaMax, 1% (vol/vol) MEM non-essential amino acids solution (Gibco-11140050), and 100 U ml⁻¹ penicillin-streptomycin. No commonly misidentified cell lines were used in this study.

## Bacterial culture and genetic manipulation

*Serratia marcescens* JC11 (herebefore referred to as SMM) was cultured in LB medium to exponential phase (200 rpm, 30 °C). Then, bacteria were washed twice with 0.9% NaCl and resuspended in PBS with serial dilution (10, $10^2$, $10^3$, $10^4$, $10^5$, and $10^6$) for counting. The bacterial count was determined by the corresponding optical density (OD at 600 nm) measured by plate reader (Infinite 200 PRO, USA). At the same time, the number of bacteria was determined by counting the colony-forming units (CFUs) after culturing the serial bacterialdilution on selective LB agar plates at 37 °C overnight.

The *msbB* gene encoding lipid A modification enzyme was disrupted in SMM via homologous recombination using the suicide plasmid pDM4 (obtained from our laboratory collection)[53]. To achieve this, Upstream and downstream homology arms flanking *msbB* were amplified by high-fidelity PCR (PCR primer sequences are provided in Supplementary Table 5) and cloned into SacI/SalI-digested pDM4 (chloramphenicol-resistant) through seamless assembly. The recombinant plasmid was introduced into *E. coli* S17-λpir for conjugative transfer. Donor (S17-λpir/pDM4-Δ*msbB*) and recipient SMM cultures were mixed (3:1 ratio) and co-incubated on LB agar with 0.22 μm membranes (37 °C for 4 h, then 28 °C for 16 h). Transconjugants were selected on LB plates containing 50 μg/mL ampicillin (to suppress donor growth) and 50 μg/mL chloramphenicol. Putative single-crossover mutants were subjected to sucrose counter-selection (10% sucrose in low-salt LB) to eliminate plasmid-retaining strains. Double-crossover mutants were confirmed by PCR amplification of the *msbB* locus and sequencing using the verification primers Δ*msbB*-yz-s (5′-GTTCCGCATCTCCTCCAACTTCAAC-3′) and Δ*msbB*-yz-a

(5′-CAACTGCATCGGTTGAAATACCTCTACC-3′). The Attenuate strain was designated *Serratia marcescens* JC11 Δ*msbB* (SM).

**Hemolysis assay.** The hemolytic activity of the bacterial strains was evaluated using Columbia blood agar plates. Briefly, the Columbia Blood Agar Base was purchased from Sangon Biotech (BBI Life Sciences, China; Catalog No. B681083) and prepared according to the manufacturer's instructions, supplemented with 5% defibrinated sheep blood. ECN, SMM, and SM strains were cultured in LB liquid broth until the optical density at 600 nm ($OD_{600}$) reached 1.0. Subsequently, 5 μL of each bacterial suspension was spotted onto the blood agar plates. A 5 μL drop of Triton ×-100 (1% v/v) was spotted as a positive control for hemolysis. The plates were then incubated at 37 °C for 12–24 h. Following incubation, the plates were visually inspected and photographed to evaluate the formation of hemolytic clearance zones.

### Bacterial culture and genetic manipulation

*Serratia marcescens* JC11 (herebefore referred to as SMM) was cultured in LB medium to exponential phase (200 rpm, 30 °C). Then, bacteria were washed twice with 0.9% NaCl and resuspended in PBS with serial dilution ($10, 10^2, 10^3, 10^4, 10^5,$ and $10^6$) for counting. The bacterial count was determined by the corresponding optical density (OD at 600 nm) measured by plate reader (Infinite 200 PRO, USA). At the same time, the number of bacteria was determined by counting the colony-forming units (CFUs) after culturing the serial bacterial dilution on selective LB agar plates at 37 °C overnight.

The *msbB* gene encoding lipid A modification enzyme was disrupted in SMM via homologous recombination using the suicide plasmid pDM4 (obtained from our laboratory collection)[53]. To achieve this, Upstream and downstream homology arms flanking *msbB* were amplified by high-fidelity PCR (PCR primer sequences are provided in Supplementary Table 5) and cloned into SacI/SalI-digested pDM4 (chloramphenicol-resistant) through seamless assembly. The recombinant plasmid was introduced into E. coli S17-λpir for conjugative transfer. Donor (S17-λpir/pDM4-Δ*msbB*) and recipient SMM cultures were mixed (3:1 ratio) and co-incubated on LB agar with 0.22 μm membranes (37 °C for 4 h, then 28 °C for 16 h). Transconjugants were selected on LB plates containing 50 μg/mL ampicillin (to suppress donor growth) and 50 μg/mL chloramphenicol. Putative single-crossover mutants were subjected to sucrose counter-selection (10% sucrose in low-salt LB) to eliminate plasmid-retaining strains. Double-crossover mutants were confirmed by PCR amplification of the *msbB* locus and sequencing using the verification primers Δ*msbB*-yz-s (5′-GTTCCGCATCTCCTCCAACTTCAAC-3′) and Δ*msbB*-yz-a (5′-CAACTG-CATCGGTTGAAATACCTCTACC-3′). The Attenuate strain was designated *Serratia marcescens* JC11 Δ*msbB* (SM). The newly generated materials in this study are available from the corresponding author upon reasonable request.

### In-vitro thermokilling assay of SM@CMC-KP101

SM@CMC-KP101 ($\approx 1 \times 10^{11}$ CFU mL⁻¹ and $1 \times 10^9$ CFU mL⁻¹) were exposed to 37, 45, 50, 55, or 60 °C for 0 min (control), 3 min, or 5 min. Samples were cooled on ice for 1 min immediately after heating. Aliquots were serially diluted, plated in triplicate on LB agar, and incubated at 30 °C for 24 h before CFU enumeration.

### Determination of encapsulation efficiency

Bacterial encapsulation efficiency was determined by quantifying unencapsulated and total bacterial loads. Briefly, CMC-microencapsulated bacteria were centrifuged ($8000 \times g$, 15 min), and the supernatant containing unencapsulated cells was serially diluted, spotted onto agar plates (10 μL/spot, triplicate), and incubated for colony counting (CFU). To measure total bacteria, microcapsules were lysed with 0.5 M EDTA (37 °C, 30 min), and the released cells were

similarly quantified. Encapsulation efficiency (%) was calculated as:

$$\left(1 - \frac{CFU_{supernatant}}{CFU_{total}}\right) \times 100 \tag{1}$$

### Construction of SM@CMC-KP101

Leveraging chemical modification techniques for targeted peptide functionalization, we coated bacterial surfaces with carboxymethyl cellulose (CMC) to introduce reactive groups for further conjugation. First, bacteria were cultured in LB medium at 37 °C with shaking (250 rpm) until reaching the logarithmic growth phase, collected via centrifugation ($3000 \times g$, 4 °C, 5 min), and washed twice with PBS to remove residual medium. The bacterial pellet was resuspended in a 10 g/L CMC solution (e.g., MW 700000) and stirred magnetically at 4 °C for 30 min to ensure uniform coating of CMC on the bacterial surface. After incubation, $2 \times$ bacterial volume of 0.2 M $CaCl_2$ was added to crosslink the CMC layer, stabilizing its attachment to the bacterial surface.

Next, the bacterial pellet was collected by centrifugation and resuspended in PBS (same volume as the pellet). EDC (5 μM) and NHS (5 μM) were added to the suspension and incubated at room temperature for 30 min to activate the carboxyl groups on the CMC layer. KP101 peptide were then added to the activated bacterial suspension, allowing the amide bond formation on the bacterial surface. The reaction was carried out at room temperature for 2 h. Prepared peptide-modified bacteria (e.g., KP101@bacteria) were purified by centrifugation ($3220 \times g$, 4 °C, 5 min) and washed three times with PBS. To investigate the impact of surface-bound CMC and KP101 on *S. marcescens* growth, SM, SM@CMC, and SM@CMC-KP101 were serially diluted and plated in triplicate on LB agar, and incubated at 30 °C for 24 h before CFU enumeration.

### Photothermal effect measurements

The photothermal effects of PG solution at different concentrations (5 mM, 2 mM, 1 mM) and in different solutions were evaluated by using 808 nm laser at 4.5 W/cm² for 180 s. The aqueous suspensions of SM at different concentrations ($1 \times 10^9$, $1 \times 10^{10}$, and $1 \times 10^{11}$ CFU/mL) under continuous 808 nm laser irradiation (4.5 W/cm²) for 180 s. Moreover, the photothermal stability of PG and SMM for four on/off irradiation cycles were monitored. The temperature was recorded with an infrared thermal camera.

### Number of KP101 on bacterial surface

First, a standard curve between fluorescence intensity and the concentration of FITC labeled KP101 was established using a fluorescence microplate reader (Infinite 200 PRO, USA) with excitation at 488 nm and emission at 520 nm. Then, the fluorescence intensity of FITC labeled KP101 on the *S. marcescens* surface was measured via Infinite 200 PRO, and the molar concentration was calculated using the regression equation of the standard curve. Next, *S. marcescens* cell number was determined using the spread plate method. The average number of KP101 molecules per bacterial cell was calculated as:

$$N_{avr} = \frac{cv\,N_A}{N} \tag{2}$$

Navr represents the average number of KP101 on the *S. marcescens* surface, c represents the molar concentration of KP101, v represents the volume of SM@CMC-KP101 for measurement, NA is Avogadro's constant $6.02 \times 10^{23}$ mol⁻¹, and N represents the number of VNP cells for measurement.

## Endotoxin detection

Endotoxin levels were quantified using a chromogenic Limulus amebocyte lysate (LAL) assay kit (C0276S, Beyotime Biotechnology) following the manufacturer's protocol[54]. Briefly, samples and endotoxin standards (0.01–1.00 EU/mL, prepared via serial dilution of a 20 EU/mL stock in endotoxin-free water) were mixed with LAL reagent and incubated at 37 °C (9–25 min, depending on expected endotoxin range). Chromogenic substrate was added, followed by a second incubation (6 min), and reactions were terminated with acidic buffer (0.4 M HCl). Absorbance at 545 nm was measured after adding stabilization reagents. A linear standard curve ($R^2 \geq 0.98$) was validated, and sample concentrations were calculated within the curve's range. All steps utilized endotoxin-free consumables, with negative controls ensuring assay validity.

## Detection of cell apoptosis

Cell apoptosis was detected using an Annexin V-FITC Apoptosis Detection Kit (KeyGEN Biotech Co., Ltd.) following the manufacturer's protocol. Briefly, B16F10 cells were seeded onto six-well plates at $2.0 \times 10^6$ cells/well for 24 h. Then, SM@CMC-KP101 was added (final concentration = $4.0 \times 10^7$ CFU/mL), followed by incubation for 8 h. Moreover, PBS, PG, and SM@CMC-KP101 were administered in equivalent proportions. After incubation at 37 °C for 8 h, the cells were collected and stained with the Annexin V-FITC apoptosis detection by fluorescence microscopy (ZEISS LSM880) was performed according to the recommended protocol.

## Cell viability assay (CCK8)

To assess the cytotoxic effect of heat-inactivated bacteria on tumor cells in vitro, CT26 and F10 cells were seeded in 96-well plates at a density of $5 \times 10^3$ cells per well and incubated overnight at 37 °C in a humidified atmosphere containing 5% $CO_2$.

SM@CMC-KP101 were cultured to mid-log phase and then heat-inactivated by incubation at 60 °C for 3 min. The concentration of the stock solution of heat-inactivated bacteria was $4 \times 10^9$ CFU/mL. The heat-inactivated bacteria were then diluted in cell culture medium to the indicated working concentrations, with a final concentration of $4 \times 10^7$ CFU/mL based on a 1:100 dilution of the stock.

After overnight incubation of cells, the culture medium was replaced with fresh medium containing various concentrations of heat-inactivated SM bacteria (0, $1 \times 10^7$, $2 \times 10^7$, $3 \times 10^7$, $4 \times 10^7$, $5 \times 10^7$ CFU/mL). After 8 h of incubation, 10 μL of CCK8 reagent (KeyGEN Biotechnology, China) was added to each well, and the plates were incubated for an additional 2 h at 37 °C. The absorbance was measured at 450 nm using a microplate reader (Bio-Rad, USA). Cell viability was calculated as a percentage relative to the untreated control.

## Bacterial internalization assay

To quantify bacterial internalization by B16-F10 murine melanoma cells, an amikacin protection assay was performed. B16-F10 cells were seeded in 24-well plates at a density of $5 \times 10^4$ cells per well and incubated overnight at 37 °C in a humidified atmosphere containing 5% $CO_2$. The following day, SMM, SM, SM@CMC, and SM@CMC-KP101 were then washed twice with sterile PBS and resuspended to a final concentration of $1 \times 10^8$ CFU/mL in DMEM supplemented with 10% FBS.

Cells were infected with each bacterial strain at a multiplicity of infection (MOI) of 20:1 (bacteria:cell). The plates were centrifuged at $500 \times g$ for 5 min to synchronize the infection and incubated for 1 h at 37 °C with 5% $CO_2$. Following incubation, the cell culture medium was removed, and the cells were washed three times with sterile PBS to remove non-adherent bacteria. To kill extracellular bacteria, cells were incubated with DMEM containing 50 μg/mL amikacin for 1 h at 37 °C.

After amikacin treatment, the cells were washed three times with sterile PBS. To release internalized bacteria, cells were lysed with 200 μL of 0.1% Triton ×-100 in PBS for 10 min at 37 °C. The lysate was then serially diluted in PBS, and 10 μL aliquots were plated on LB agar plates. The plates were incubated overnight at 37 °C, and the number of colony-forming units (CFU) was enumerated. The percentage of internalized bacteria was calculated as (CFU recovered/CFU input) × 100. All conditions were performed in triplicate, and data are presented as mean ± standard deviation.

## In vitro macrophage activation

M2 macrophages were polarized from Raw264.7 cells and BMDM as previously described[55]. Briefly, $1 \times 10^6$ Raw264.7 cells were cultured in a 12-well plate in DMEM medium containing IL-4 (20 ng/mL) for 7 - 8 days to form M2 macrophages. In parallel, bone marrow-derived macrophages (BMDMs) were generated. Bone marrow cells were flushed from the femurs and tibias of C57BL/6J and cultured in petri dishes in DMEM supplemented with 10% FBS, 1% penicillin/streptomycin, and 20 ng/mL M-CSF for 7 days to differentiate into macrophages. After differentiation, BMDMs were treated with 20 ng/mL IL-4 for 48 h to induce M2 polarization. The obtained M2 macrophages were washed and incubated with PBS, ECN (final concentratin = $1 \times 10^6$ CFU/mL), SM-EV (final concentratin = $1 \times 10^6$ CFU/mL), SM@CMC-KP101 (final concentratin = $1 \times 10^6$ CFU/mL), and PG (final concentratin = 12.5 nM) for 8 h, respectively. After 8 h, the media was aspirated and wells were washed six times with sterile ice-cold PBS. Total RNA from cell samples was isolated according to the manufacturer's instructions. The relative expression of target mRNAs (iNOS, TNFα, CD206, IL10) was normalized to β-actin as an internal control and calculated using the $Log_2(2^{-\Delta\Delta Ct})$ method. RT–qPCR primer sequences are provided in Supplementary Table 7.

## In vitro BMDC stimulation

Bone marrow-derived dendritic cells (BMDCs) were generated from the bone marrow of 6–8-week-old female C57BL/6J mice using a modified version of previously described methods[56]. Briefly, murine femurs and tibias were aseptically harvested, and bone marrow cells were flushed into culture medium. After centrifugation to remove debris, cells were resuspended in complete culture medium consisting of RPMI-1640 supplemented with 10% FBS (or serum-free alternative), 1% penicillin/streptomycin, and 20 ng/mL GM-CSF. Cells were cultured at 37 °C in a humidified atmosphere containing 5% $CO_2$, with a half-medium change and replenishment of GM-CSF on day 3. On days 6–8, non-adherent cells were collected and used as BMDCs. BMDCs were stimulated with microbial strains or other experimental conditions, and their activation was assessed by flow cytometry or other functional assays. The obtained BMDCs were washed and incubated with PBS, ECN (final concentratin = $1 \times 10^6$ CFU/mL), SM-EV (final concentratin = $1 \times 10^6$ CFU/mL), SM@CMC-KP101 (final concentratin = $1 \times 10^6$ CFU/mL), and PG (final concentratin = 12.5 nM) for 8 h, respectively. After 8 h, the media was aspirated and wells were washed six times with sterile ice-cold PBS. Flow cytometric analysis was performed to assess BMDC activation. The following antibodies were used: CD11c-APC (#117310), CD86-PE (105022), and CD80-Pacific Blue (104723). Cells were initially gated on the lymphocyte population based on forward scatter (FSC) and side scatter (SSC) characteristics. Doublets were excluded using FSC-H/FSC-A and SSC-H/SSC-A plots. CD11c$^+$ cells were then gated from the single-cell population, and their activation status was determined by co-expression of CD80$^+$ and CD86$^+$.

## Animal models and in vivo experiments

Female BALB/c and C57BL/6J mice (6–8 weeks old) were purchased from Hangzhou Ziyuan Laboratory Animal Technology Co., Ltd. To establish syngeneic tumor models, BALB/c mice and C57BL/6J mice were subcutaneously injected into the left axillary region with $1.0 \times 10^6 1.0 \times 10^6$ CT26 cells and B16F10 cells, respectively, suspended in 100 μL of PBS. Tumor volume was measured using a digital caliper

and calculated as volume = length × width2 × π/6. Experiments were conducted when the tumor volume reached 100 mm³. Euthanize the mice when the tumor volume is about to exceed 2000 mm³ or has already exceeded 2000 mm³, or after 14 days of culture.

To evaluate the in vivo photothermal therapeutic efficacy of ECN, VNP20009, SM-EV, SM@CMC-KP101, and PG in CT26-bearing BALB/c were administered via intratumoral injection at $1.0 \times 10^{10}$ CFU/mice. After injection, the tumor underwent 808 nm NIR laser irradiation (4.5 W/cm² for 3 min), and the corresponding temperature change was recorded using a digital infrared thermal camera. Beside, B16F10 bearing C57BL/6J mice were administered via intravenous injection at $1.0 \times 10^{10}$ CFU/mice. The antitumor efficacy was evaluated on the fourteenth day following treatment administration of varying durations (0, 3, 6, or 10 days), necessitated by the substantial tumor volume observed in the control cohort. Throughout the 14-day experimental period, systematic measurements of body weight and tumor dimensions were recorded, alongside continuous monitoring of survival rates among tumor-bearing subjects across different treatment modalities. Upon conclusion of the study, the subjects were humanely euthanized, and tumor specimens were harvested and subsequently preserved in 10% neutral buffered formalin. The preserved tissue samples underwent paraffin embedding, sectioning at 5 µm thickness, and hematoxylin and eosin (H&E) staining for histological examination via digital microscopy.

## Evaluation of synergistic effects

To evaluate the nature of the interaction between the components, the Coefficient of Drug Interaction (CDI) was calculated based on the tumor volumes measured on day 14 post-treatment. The CDI was determined using the formula:

$$CDI = \frac{E_{AB}}{E_A \times E_B} \tag{3}$$

where AB represents the ratio of the tumor volume in the combination group to that of the control group, and A and B denote the ratios of the tumor volumes in the respective single-agent groups to the control group[57]. According to established pharmacological standards, a CDI value less than 1 indicates a synergistic effect, a value equal to 1 indicates an additive effect, and a value greater than 1 implies an antagonistic effect.

## DC and T-cell stimulation in tumor-draining lymph node (TDLN) in vivo

The mouse model was established as described earlier, and lymph nodes from the tumor-draining regions were excised 24 h after intravenous injection and subsequent sacrifice of the mice. Aseptically excise the TDLNs and immediately place them in cold complete RPMI-1640 medium (supplemented with 2% FBS). Maintain the lymph nodes on ice or at 4 °C throughout the procedure. Place the lymph nodes in a 70 µm cell strainer positioned over a 50 mL Falcon tube. Add 2–5 mL of cold complete RPMI-1640 to the strainer and gently disrupt the lymph node tissue by pressing it against the mesh using the plunger of a sterile 5 mL syringe (or a rubber pestle) in circular motions. Wash the cell strainer with an additional 2–3 mL of cold PBS to maximize cell recovery. Collect the filtered cell suspension and centrifuge at 300–400 × g for 5 min at 4 °C. This cell suspension was treated with CD11c-APC (#117310), CD80-FITC (#104705), CD86-Pacific Blue (#105022) antibodies, followed by flow cytometry analysis to detect dendritic cells (DCs). CD11c+ cells were gated on single cells, with activation status evaluated by CD80 and CD86 co-expression. Additionally, the same cell suspension was processed with CD3-FITC (#100204) or CD69-FITC (#104505), CD4-PE (#100512), CD8a-Pacific Blue (#100725) antibodies to identify T cells. CD69+ cells were gated on single cells, then separated into CD4+ and CD8+ populations[58].

Representative flow cytometry gating methods are summarized in Supplementary Fig. 19b, e, f.

## Macrophage Phenotyping (24 h) in 24 h post-treated bone marrow

The mouse model was constructed as previously described. At 24 h post-treatment, the femurs and tibias were aseptically excised, and excess tissue was removed. Using a syringe with a 25 G needle, the bone marrow was flushed from the bones with cold complete RPMI-1640 medium (supplemented with 2% FBS). The resulting cell suspension was gently pipetted to break up any clumps and then filtered through a 70 µm cell strainer positioned over a 50 mL Falcon tube to remove debris. The cell strainer was washed with an additional 2–3 mL of cold PBS to maximize cell recovery. After collection, the filtered cell suspension was centrifuged at 300–400 × g for 5 min at 4 °C. The pellet was resuspended in appropriate buffer for further analysis.

For phenotyping, the cell suspension was stained with the following fluorochrome-conjugated antibodies: BV480 Rat Anti-Mouse CD45 (#752417), FITC Rat Anti-CD11b (M1/70) (#557396), and CD206 (MMR) Monoclonal Antibody (PE, #12-2061-82) from eBioscience. Flow cytometry was performed to determine the percentages of M1 (CD45 + CD11b + F4/80 + CD86 +) and M2 (CD45 + CD11b + F4/80 + CD206 +) macrophages in the bone marrow[59,60]. Representative flow cytometry gating methods are summarized in Supplementary Supplementary Fig. 19c.

## Comprehensive assessment of T cell subpopulations in spleen

After the treatment finished, spleens were aseptically excised on day 11 following photothermal therapy in the CT26 tumor model and on day 14 after intravenous injection therapy in the B16F10 model. The excised spleens were immediately placed in cold complete RPMI-1640 medium (supplemented with 2% FBS), maintaining their temperature on ice or at 4 °C throughout the procedure. To prepare splenocyte suspensions, each spleen was placed in a 70 µm cell strainer positioned over a 50 mL tube. 2–5 mL of cold complete RPMI-1640 was added to the strainer, and the spleen tissue was gently disrupted by pressing it against the mesh using the plunger of a sterile 5 mL syringe in circular motions. The cell strainer was washed with an additional 2–3 mL of cold PBS or RPMI to maximize cell recovery. The filtered cell suspension was collected and centrifuged at 300–400 × g for 5 min at 4 °C.

Splenocytes were stained with specific surface markers for flow cytometry analysis. For (a) splenocyte-derived T cell subsets, live CD45⁺ cells were gated on CD3⁺ T cells and then subdivided into CD4+ (stained with APC Rat Anti-Mouse CD4, #553051) and CD8⁺ populations (stained with Ms CD8a PE, #553032). Memory phenotypes were defined within the CD4⁺ and CD8⁺ populations as central memory (TCM: CD44⁺ CD62L⁺, using Ms CD44 BV421, #563970, and PE-Cy7 Rat Anti-Mouse CD62L, #560516) and effector memory (TEM: CD44⁺ CD62L⁻). For (b) splenocyte-derived Treg cells, live CD45+ cells were first gated on singlets and then on CD4⁺ T cells. Regulatory T cells (Treg) were identified as CD25⁺ (stained with anti-mouse CD25-APC, #17-0251-81) and FoxP3+ (stained with FITC anti-mouse FOXP3⁺, #11-5773-82) cells within the CD4⁺ population. For (c) splenocyte-derived cytotoxic T cell function, after gating on singlets and CD3⁺ T cells (stained with CD3-FITC, #100204), CD8⁺ T cells were identified and further analyzed for intracellular IFN-γ expression using IFN-γ-APC (#505810) following stimulation. Representative flow cytometry gating methods are summarized in Supplementary Supplementary Fig. 19a, d, g.

## Detection of cellular ROS levels

ROS levels were determined using a CellROX™ Deep Red Flow Cytometry Assay Kit (ThermoFisher) following the manufacturer's protocol. The treatment processes were the same as those in the apoptosis assay. Then, the medium was removed, and the cells were washed 3

# Article

times with PBS and probed with 5 μM CellROX™ Deep Red in serum-free DMEM. After incubation in 5% $CO_2$ at 37 °C for 30 min, the cells were collected, and fluorescence was detected by using the PE-Texas Red channel of a flow cytometer (Agilent, NovoCyte).

### SiRNA-mediated gene silencing of MAPKAPK5

B16-F10 melanoma cells were cultured in DMEM supplemented with 10% fetal bovine serum (FBS) and penicillin-streptomycin at 37 °C in a humidified atmosphere containing 5% $CO_2$. Three siRNAs targeting MAPKAPK5 were synthesized by GenScript, and the specific sequences are provided in Supplementary Table 6. Transfection was performed using Lipofectamine™ 3000 (Thermo Fisher Scientific) according to the manufacturer's instructions. Cells were seeded in 6-well plates at $1.0 \times 10^5$ cells and incubated overnight. For each well, 50 nM of siRNA was mixed with 2 μL of transfection reagent in 200 μL of Opti-MEM and allowed to incubate for 10 min at room temperature before being added to the cells. The cells were then incubated for an additional 48 h at 37 °C with 5% $CO_2$.

To validate the efficiency of MAPKAPK5 gene silencing, total RNA was extracted from the cells using EZ-10 Total RNA Mini-Preps Kit (Sangon Biotech, NO. B618583) following the manufacturer's protocol. Complementary DNA (cDNA) was synthesized using the MightyScript First Strand cDNA Synthesis Master Mix (Sangon Biotech, NO. B639251). Quantitative PCR (qPCR) was conducted using MAPKAPK5 primers purchased from Beyotime, with expression levels normalized to β-actin. The relative expression levels were quantified using the ΔΔCT method.

### DIA proteomic analysis.

B16F10 cells were seeded onto six-well plates at $2.0 \times 10^6$ cells/well for 24 h. Then, PBS or SM@CMC-KP101 was added (final concentration = $4.0 \times 10^7$ CFU/mL; $n = 3$ independent biological replicates per group), followed by incubation for 12 h. After incubation at 37 °C for 12 h, the cells were collected and frozen for subsequent analysis.

Frozen samples (approximately 100 mg) were quickly ground into a fine powder in liquid nitrogen and subsequently lysed in 300 μL lysis buffer supplemented with phosphatase inhibitors and 1 mM PMSF. The samples were further lysed with sonication (1 s on/1 s off intervals at 80 W for 2 min on ice). After sonication, samples were centrifuged at $12,000 \times g$ for 10 min at 4 °C to remove insoluble particles. Protein concentrations were determined using the BCA assay. For SDS-PAGE analysis, an equal amount of protein from each sample was separated by 12% SDS-PAGE gels and stained with Coomassie Brilliant Blue to evaluate sample quality.

Based on the protein concentrations, 50 μg of protein from each sample was diluted to a uniform concentration and volume. Samples were reduced with 5 mM DTT at 55°C for 30 min, cooled to room temperature, and alkylated with 10 mM iodoacetamide in the dark for 15 min. Proteins were then precipitated with 6 volumes of pre-cooled acetone at −20 °C overnight and centrifuged at $8000 \times g$ for 10 min at 4 °C. The precipitate was redissolved in 50 mM $NH_4HCO_3$ and digested with Trypsin (1:50 w/w, Trypsin:protein) at 37 °C overnight. Digestion was stopped by adjusting the pH to approximately 3 with phosphoric acid, and samples were desalted using SOLA™ SPE 96-well plates. After drying under vacuum, the samples were spiked with iRT standard peptides (Biognosys) at a 1:20 v/v ratio prior to MS analysis.

Proteomic analyses were performed by Shanghai OE Biotech Co., Ltd. Peptides were separated using a Vanquish Neo UHPLC system (Thermo Fisher Scientific). Mobile phase A consisted of 0.1% formic acid in water, and mobile phase B consisted of 0.1% formic acid in 80% acetonitrile. The separation was performed using a high-throughput gradient strategy (total 22.6 min): 4% B for 0.5 min, 8% B at 0.6 min, 8.5% B at 0.9 min, 22.5% B at 13.9 min, 35% B at 20.8 min, 55% B at 21.2 min, and holding at 99% B from 21.7 to 22.6 min. The flow rate was dynamically adjusted from 1.3 μL/min to 0.8 μL/min during separation and 2.5 μL/min during the washing phase.

The separated peptides were analyzed using an Orbitrap Astral mass spectrometer (Thermo Fisher Scientific) in Data-Independent Acquisition (DIA) mode. The MS parameters were set as follows: Orbitrap resolution at 240,000; full MS scan range of 380−980 m/z; MS/MS scan range of 150−2000 m/z; isolation window of 2 m/z; and higher-energy collisional dissociation (HCD) collision energy at 25%. The cycle time was set to 0.6 s.

For data parsing and protein identification, the raw DIA data were processed using DIA-NN software. The spectra were searched against the UniProt *Mus musculus* database (release 2024.2.1, combined with iRT sequences). The search parameters were set as follows: trypsin was specified as the cleavage enzyme with a maximum of 1 missed cleavage allowed; carbamidomethylation on cysteine was set as a fixed modification, while oxidation of methionine and acetylation of the protein N-terminus were set as variable modifications. Both the peptide-spectral match (PSM) false discovery rate (FDR) and the protein-level FDR were strictly controlled at ≤ 1% (0.01).

### Statistics and reproducibility.

All quantitative data are presented as the mean ± standard deviation (SD) from at least three independent biological replicates unless otherwise specified. No statistical methods were used to predetermine sample sizes; rather, sample sizes for in vivo studies ($n \geq 3$ mice per group) were determined based on standards generally accepted in the field and our previous experience with similar tumor models.

For animal experiments, mice were randomly assigned to different treatment groups when their tumor volumes reached approximately 100 mm³. Investigators were blinded to the group allocation during data collection and outcome assessment. No animals or data points were excluded from the analyses, except for mice that reached the humane endpoint (tumor volume ≥2000 mm³), which were euthanized and recorded as events for survival analysis.

Statistical analyses were performed using GraphPad Prism software (version 10.0). Statistical significance between two independent groups was evaluated using a standard two-tailed unpaired Student's *t*-test. For comparisons among multiple groups, a two-sided one-way analysis of variance (ANOVA) was used, followed by either Dunnett's post-hoc test (for comparing multiple treatment groups to a single control) or Tukey's post-hoc test (for multiple pairwise comparisons), as specified in the corresponding figure legends. Animal survival distributions were estimated and plotted using the Kaplan–Meier method. A *P* value of <0.05 was considered statistically significant. Exact *P* values are provided in the figure legends.

### Reporting summary

Further information on research design is available in the Nature Portfolio Reporting Summary linked to this article.

## Data availability

The mass spectrometry-based proteomics data generated in this study have been deposited in the ProteomeXchange Consortium via the PRIDE partner repository under accession code PXD074961. The differentially expressed gene data generated in this study are provided in Supplementary Data 1. All other data supporting the findings of this study are available within the paper and its Supplementary Information. Source data for all corresponding main text and Supplementary Figures (including the uncropped and unprocessed scans of blots) are provided with this paper. Source data are provided with this paper.

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

## Acknowledgements

This work was supported by the National Natural Science Foundation of China (22293052, 22025701, and 92353301 to J.Z.; 22477057 to X.X.W.; 22177048 to W.W.), the National Key R&D Program of China (2023YFA1508900 to X.X.W.), the Natural Science Foundation of Jiangsu Province (BK20232020 to J.Z.), the Jiangsu Provincial Science and Technology Plan Special Fund (BM2023008 to W.W.), the Nanjing Science and Technology Program (202305003 to J.Z.), the Fundamental and Interdisciplinary Disciplines Breakthrough Plan of the Ministry of Education of China (JYB2025XDXM507 to J.Z.), the Fundamental Research Funds for the Central Universities (KG202510 to W.W., 2024300401 to X.X.W.), and the Yachen Foundation of Nanjing University (to J.Z.).

## Author contributions

Conceptualization, J.L.H., Z.T.Z., W.L. and W.W.; Methodology, J.L.H., Z.J.Y., W.J. and W.X.X.; Experimental execution, J.L.H., J.T.Q., Q.Z.H., G.S.Q., W.Y.Q. and H.H.M.; Investigation, J.L.H., J.T.Q., G.S.Q., Q.Z.H., H.H.M., W.Y.Q. and W.L.; Writing–original draft, J.L.H.; Writing–review & editing, W.X.X., L.C. and W.W.; Funding acquisition, M.Y.L., Z.J., and W.W.; Resources, W.X.X., Z.J. and W.W.; Supervision, Z.J. and W.W.

## Competing interests

The authors declare no competing interests.
