## [Transparent Peer Review file · Nature Communications]

A live biohybrid bacterial therapy based on engineered *Serratia marcescens*

Corresponding Author: Professor Wei Wei

Version 0:

Reviewer comments:

Reviewer #1

(Remarks to the Author)

This manuscript describes an engineered *Serratia marcescens* strain with enhanced prodigiosin production and attenuated virulence, which is applied as a live bacterial vector for anti-tumor immunotherapy and photothermal therapy. The topic is novel and interesting, and the experimental data, while preliminary in some respects, suggest potential value for cancer theranostics.

However, the manuscript contains numerous inconsistencies, ambiguities, and errors—both conceptual and technical—that undermine its scientific rigor and interpretability. In its current form, the manuscript does not meet the standards for publication in a scientific journal. Although the manuscript contains numerous other issues, I have chosen to highlight only a few illustrative examples.

1. Inconsistent nomenclature and group labeling

In lines 271–275, the descriptions of the experimental groups are inaccurate and confusing. For example, SM@CMC-KP101, which refers to the prodigiosin-producing strain, is incorrectly abbreviated again as “SM-PG,” a term previously defined as the non-producing (empty vector) control. Such inconsistencies appear throughout the manuscript and figures, making it difficult for the reader to follow the experimental design.

2. Memory T cell definition in Figure 4g

The memory T cells are labeled as CD3⁺CD8⁺CD44⁺CD62⁺, which is ambiguous. It is unclear whether the authors are referring to central memory or effector memory T cells. Given that these cells are isolated from spleen, the CD62L⁺ gating in the figure legend is conceptually inconsistent with known tissue localization. Moreover, the figure and its legend are inconsistent in this labeling. The authors must clarify both the phenotype and the anatomical context to support any claims regarding memory T cell activation.

3. Unsupported illustration in Figure 3k

Figure 3k illustrates activation of CD8⁺ T cells, yet no corresponding in vitro data are presented in the figure or preceding text to support this claim. While in vivo CD8⁺ T cell activation is shown later (e.g., in Figure 4g), its inclusion in the schematic of in vitro results is misleading unless clearly labeled as hypothetical or supported by data.

4. Unjustified generalizations

The claim that high temperatures “effectively eradicated bacterial carriers” (line 247) is based on in vivo CFU reduction (Figure 4i), but the only in vitro support is qualitative (Figure S4b), showing colony fading and no direct CFU quantification. A more rigorous, quantitative assay would strengthen this conclusion.

5. Abbreviation use

Numerous abbreviations appear without being defined on first use (e.g., DA, ECN, VNP20009, PG, etc.), and several are used inconsistently across the manuscript and figures (e.g., SM vs. SMT). A glossary or consistent formatting is needed. I recommend that the manuscript be revised thoroughly to address the above concerns. The authors should focus on ensuring consistency in terminology, providing adequate experimental support for mechanistic claims, and improving the clarity and precision of figure legends and group definitions. After these major revisions, the manuscript may be reconsidered for publication.

Reviewer #2

(Remarks to the Author)

This study proposes a biohybrid bacterial therapy using engineered *Serratia marcescens* to achieve high-yield prodigiosin synthesis, tumor-targeted delivery, photothermal ablation, and immunomodulation. The study addresses an innovative topic with a well-designed experimental framework. However, critical data gaps (e.g., dose-response curves, photothermal safety, mechanistic crosstalk) and insufficient discussion of translational challenges must be addressed. The work holds significant potential for publication post-revision.

1. Clarify the basis for bacterial dosage (e.g., 4×10^9 CFU) with dose-response experiments (e.g., tumor suppression rates across CFU gradients).
2. Address potential thermal damage to normal tissues (e.g., skin, vasculature) at 60°C. Include long-term toxicity data or thermal tolerance thresholds.
3. Supplement Figure 4i with CFU counts in major organs (e.g., liver, spleen) to confirm systemic clearance post-photothermal treatment.
4. Elucidate the molecular interplay between p38/MK5-HSP27 signaling and mitophagy (e.g., via genetic knockout or pathway-specific inhibitors).
5. Assess immune memory (e.g., anti-bacterial antibodies) or autoimmune risks post-treatment.
6. Compare *S. marcescens* with other bacterial vectors (e.g., VNP20009, EcN) in terms of tumor-targeting efficiency and immune activation to highlight its unique advantages.
7. Discuss limitations of photothermal therapy (e.g., laser penetration depth, clinical equipment requirements) and challenges in scaling bacterial production.
8. Benchmark the reported prodigiosin yield (8.3 g/L) against prior studies.
9. The numbering of the supplementary figures in the supplementary information file appears inconsistent. For example, line 49, Figure S15 is followed by Figure S18, and Figure S18 is followed by Figure S16.
10. Specify sources of critical reagents (e.g., KP101 peptide synthesis vendor) and include animal ethics approval numbers.

Reviewer #3

(Remarks to the Author)

I generally agree with the authors' interpretations of the results, and find them of general interest and interest in the area of therapeutic bacteria. They have introduced the use of a bacterium not currently being studied for its anti-tumor properties, and also provide a novel image modality with NIR prodigiosin and a novel mode of bacterial elimination. This bacterium was able to kill cells *in vitro* and had anti-tumor activity against B16F10 murine melanoma and CT26 murine colon carcinoma models *in vivo*. This particular melanoma is one that has limited sensitivity to checkpoint inhibitors such as PD-1, which makes that model more difficult to demonstrate a therapeutic effect with immunomodulators. The bacteria also promoted maturation of dendritic cells, recruitment of T-cells, and favorable macrophage polarization, which may improve the immunotherapeutic effect in these or other models. In B16, apoptosis and mitophagy-induced necrosis also occurred and may offer combined advantages in the use of these bacteria.

However, there are some specific areas of the conclusions that seem to be implied, but are not specifically addressed.

The improved production at a lower temperature *in vitro* was not demonstrated to occur at higher temperatures. C57 mice are 36 to 38°C and BalbC are 36.4 to 38.4°C - how much do they make at mouse temperatures?

Bacterial entry into cells was purportedly shown (see comment for line 155 below), but there was no comparison with the wild type, msbB, or BP101. Is the strain improved in its ability to enter cells, and if so, was that improvement due to the PB101 peptide?

Similarly, the attenuated strain with CMC and BP101 was shown to target tumors, but it cannot be determined if that was a preexisting phenotype, or if the msbB, CMC, or PB101 caused any significant improvement.

RESULTS section

A discussion of Figure 1 is absent.

Line 87. The "mutational breeding" used in the study is not described.

Line 89. Physiological temperatures for production of prodigiosin that is the comparative production of prodigiosin at physiological temperature (37°C)? This is especially important if it is down-regulated at higher temperatures (e.g., 42°C).

Line 104 refers to "Building on the hypoxia-targeting capability of *S. marcescens*" while line 116 it states: "and does not proliferate under hypoxic conditions". Please explain what appears to be a contradiction.

The ability to grow under hypoxic conditions is widely believed to be a component of *Clostridium*, *Salmonella* and other bacteria's ability to target tumors, and is not generally linked to toxicity except in the cases of colonizing pre-existing abscesses. How is this an improvement?

Line 90. How was elevated expression of the PigP operon achieved?

Line 147-149. I suggest indicating this refers to a schematic diagram as is indicated in Figure Legend 2. Also in Figure 2f, I am unable to read the magnification bars in this version.

Line 155. Figure 3a needs a higher resolution photo to be convincing as to bacteria having been internalized. Higher magnification and counter staining with a nuclear stain such as DAPI would be helpful. There is also a common assay using gentamicin (assuming your strain is sensitive) DOI: 10.1016/0076-6879(94)36030-8.

An additional comment in this regard as stated above: there are no controls for the ability of the wild type to be internalized due to the peptide (e.g, comparison with WT) or the effect of the msbB, CMC coating and BP101 itself on internalization.

Line 556. Please change tumor models to tumor cells.

Line 160 and Figure Legend 3c. Which cell line is this?

Figure Legend 4. Please indicate this is the CT26 tumor model.

Lines 210-211. Please clearly identify the doses being administered by the different routes of injection - e.g., 100 ul of 1×10^{11} CFU/ml = 1×10^{10} CFU/mouse (or tumor if it).

Lines 221-222. A good tumor/normal tissue ratio is shown, but in the absence of a WT and other comparators, it is unable to be interpreted as an improvement of the wild type. Any such comparison should also individually include the steps in modification, including the delta-msbB, CMC coating, and the addition of KP101.

Figure Legend 5. Please indicate this is the B16F10 tumor model.

Version 1:

Reviewer comments:

Reviewer #2

(Remarks to the Author)

The authors have undertaken a comprehensive and rigorous revision of the manuscript. I recommend acceptance of the manuscript in its current form for publication. The revisions have thoroughly addressed the initial concerns, and the work now meets the high standards of the journal.

Reviewer #3

(Remarks to the Author)

The authors have substantially improved the manuscript and have satisfactorily answered my original questions.

In clarifying the dosing, the need for additional clarification has arisen. The fact that ECN and VNP20009 led to rapid mortality is likely due to their having been used past their tolerated doses (common dose range is 10^6 to 10^7), effectively overdosing the mice with 1×10^8 . While the authors may have chosen the dose of 1×10^8 /mouse as being equivalent in number to the dose of *Serratia*, the more relevant comparison of drugs for relative efficacy is performed at the effective dose or maximum tolerated dose for each drug. This should be clarified in the text.

Reviewer #4

(Remarks to the Author)

I am satisfied with the answers to most of the questions. However, I maintain my criticism regarding the limited scientific rigor and interpretability raised by Reviewer #1. Some of the newly provided data are also unconvincing.

Memory T cells

My impression is that these experiments were performed only once; they should be independently reproduced.

Figure 3j is described as showing antigen-specific T cells, but no supporting evidence is provided. Tumor-specific T cells

should be evaluated as H-2Ld-MuLV gp70 peptide (SPSYVYHQF) tetramer-binding CD8+ T cells (Tet+ CD8+). T cells in the draining lymph node were characterized only by general surface markers. Therefore, it is unclear whether they were tumor-specific or potentially bacteria-specific. Why were tumor-infiltrating lymphocytes (TILs) not analyzed? Figures 4k-l and 4o-p show an increased memory T-cell population in tumor-eradicated mice, and the authors claim durable antitumor immunity. However, there is no tumor rechallenge study to confirm functional long-term memory T cells generated after bacterial treatment.

There is no significant difference in the immune profile between SM@CMC-KP101 (i.t.) and SM@CMC-KP101 (i.v.) (Fig. 4f-m), which is inconsistent with the therapeutic efficacy shown in Fig. 4p (i.t. shows two tumor-eradicated mice, whereas i.v. shows none).

Additionally, there are no significant differences in DCs, CD8 T cells, M1 macrophages, M2 macrophages, or Tregs in SM@CMC-KP101 (i.t.)/(i.v.) compared with ECN, VNP20009, and SM-EV. If so, why are there significant differences in antitumor activity? Moreover, neutrophils (CD11b+Gr-1+), which are typically strongly recruited after bacterial infection, were not assessed.

The synergy index was not calculated (PMCID: PMC11303714). The combination may be additive rather than synergistic.

Figure 4b appears to have an incorrect label for SM@CMC-KP101 (DA).

Version 2:

Reviewer comments:

Reviewer #4

(Remarks to the Author)

The authors have carried out a thorough and careful revision of the manuscript. I recommend acceptance for publication in its present form. The revisions comprehensively resolve the original concerns, and the study now satisfies the Nat Comms

A live biohybrid bacterial therapy based on engineered *Serratia marcescens*

Lihao Ji¹, Tianze Zhu¹, Tianqi Jiang¹, Li Wang¹, Zhonghui Qiu¹, Shiqi Gao¹, Yuqi Wang¹, Jing
Wang², Jingyi Zhang², Haomiao Huang¹, Yunlong Mao⁵, Chen Lin², Jing Zhao^{2,3,4,*}, Xiuxiu
Wang^{1,2,4,**}, Wei Wei^{1,3,4,***}

¹ School of Life Sciences, Nanjing University, Nanjing 210093, P. R. China.

² State Key Laboratory of Coordination Chemistry, Chemistry and Biomedicine Innovation Center (ChemBIC),
School of Chemistry and Chemical Engineering, Nanjing University, Nanjing 210093, P. R. China.

³ Nanchuang (Jiangsu) Institute of Chemistry and Health, Sino-Danish Ecolife Science Industrial Incubator,
Jiangbei New Area, Nanjing 210000, P. R. China.

⁴ Wuxi Xishan NJU Institute of Applied Biotechnology, Wuxi 214000, P. R. China.

⁵ State Key Laboratory of Novel Software Technology, Nanjing University, Nanjing 210093, P. R. China.

Present address: School of Life Sciences, Nanjing University, Nanjing 210093, P. R. China.

Lead Contact

*Correspondence: jingzhao@nju.edu.cn

**Correspondence: wangxiuxiu@nju.edu.cn

***Correspondence: weiwei@nju.edu.cn

**REVIEWER COMMENTS**

Reviewer #1 (Remarks to the Author):

This manuscript describes an engineered *Serratia marcescens* strain with enhanced prodigiosin
production and attenuated virulence, which is applied as a live bacterial vector for anti-tumor
immunotherapy and photothermal therapy. The topic is novel and interesting, and the experimental
data, while preliminary in some respects, suggest potential value for cancer theranostics.

However, the manuscript contains numerous inconsistencies, ambiguities, and errors—both
conceptual and technical—that undermine its scientific rigor and interpretability. In its current
form, the manuscript does not meet the standards for publication in a scientific journal. Although
the manuscript contains numerous other issues, I have chosen to highlight only a few illustrative
examples.

Q1. Inconsistent nomenclature and group labeling

In lines 271–275, the descriptions of the experimental groups are inaccurate and confusing. For
example, SM@CMC-KP101, which refers to the prodigiosin-producing strain, is incorrectly
abbreviated again as "SM-PG," a term previously defined as the non-producing (empty vector)
control. Such inconsistencies appear throughout the manuscript and figures, making it difficult for
the reader to follow the experimental design.

**Response:**

We thank the reviewer for pointing out the confusion caused by inconsistent abbreviations (e.g.,
the misuse of "SM+PG" for SM@CMC-KP101). In response, we conducted a manuscript-wide
audit covering the main text, figure panels/legends, tables, and Supplementary Information. We
have replaced all ambiguous or conflicting terms with a single, standardized nomenclature and
ensured one-to-one correspondence between each abbreviation and its full experimental definition.

To make the study design fully transparent, we now include a summary table (Table S1:
Abbreviation-to-sample mapping) listing every abbreviation used in the manuscript and its precise
meaning.

We believe these revisions eliminate ambiguity and allow the experimental groups to be
followed unambiguously throughout the manuscript and figures.

**Revisions Made:**

• Lines 87-89: Starting from *S. marcescens* NRRLB-1481, we developed an engineered strain,

*S. marcescens* JC11 (hereafter referred to as SMM), through iterative UV irradiation and
 visual screening for intensified red pigmentation, indicating enhanced prodigiosin
 biosynthesis (Figure S1, S2).

• Lines 118-121: To establish a robust, dual-layered safety profile for this approach, we first
 engineered an attenuated strain based on SMM, by deleting the *msbB* gene (hereafter
 referred to as SM), which is critical for the biosynthesis of endotoxic lipid A; the successful
 knockout was confirmed by PCR (Figure 1i).

• Lines 133-137: Like *Escherichia coli* Nissle1917 (ECN) and *Salmonella typhimurium* VNP20009
 (VNP20009), SM is a facultative anaerobe; however, under strict anaerobic conditions at 37°C,
 SM growth is significantly inhibited (Figure S8). Interestingly, the degree of anaerobic growth
 inhibition observed for SM was less pronounced than that of ECN, and roughly equivalent to
 VNP20009 (Figure S8).

• Lines 152-154: Therefore, we selected CMC for encapsulation, naming the resulting composite
 SM@CMC, where the abundant carboxyl groups of CMC provide a rich resource for subsequent
 modification reactions.

• Lines 155-157: To further enhance tumor targeting, we conjugated the cyclic peptide
 KP101-which exhibits high affinity for tumor-associated PD-L1 receptors-onto SM@CMC
 via EDC/NHS coupling, yielding SM@CMC-KP10 1 biohybrids.

• Lines 183-185: Initial experiments using Annexin-V/PI staining in B16F10 murine melanoma
 cells showed an apparent increase in cell death following treatment with prodigiosin (PG) alone
 and SM@CMC-KP101 (Figure S14).

• Lines 233-234: To ensure the safe and effective use of SM@CMC-KP101, we performed
 dose-escalation studies via intratumoral (i.t.) and intravenous (i.v.) administration in CT26
 tumor-bearing mice (Figure S18a, b).

Table S1 Abbreviation-to-sample mapping

Abbreviation	Full name
Control	PBS treated
ECN	Escherichia coli Nissle1917
JM109	Escherichia coli JM109
VNP20009	Salmonella typhimurium VNP20009
SMM	Serratia marcescens JC11 with pre-synthesized prodigiosin
SM	Serratia marcescens JC11 Δ msbB with pre-synthesized prodigiosin
SM-EV	SM without PG production (empty-vector control), which can be obtained by incubating at 42°C for 24 hours.
SM@CMC	SM coated with CMC
SM@CMC-KP101	SM@CMC linked with KP101
PG	Prodigiosin
SM@CMC-KP101(i.t.)	SM@CMC-KP101 administered intratumorally (i.t.).
SM@CMC-KP101(i.v.)	SM@CMC-KP101 administered intravenously (i.v.).

Q2. Memory T cell definition in Figure 4g
The memory T cells are labeled as CD3+CD8+CD44+CD62+, which is ambiguous. It is unclear
whether the authors are referring to central memory or effector memory T cells. Given that these
cells are isolated from spleen, the CD62L+ gating in the figure legend is conceptually inconsistent
with known tissue localization. Moreover, the figure and its legend are inconsistent in this labeling.
The authors must clarify both the phenotype and the anatomical context to support any claims
regarding memory T cell activation.

**Response:**

We thank the reviewer for this insightful comment. We've clarified the specific phenotypes of
central memory (Tcm) and effector memory (Tem) CD8⁺ T cells derived from the spleen in this
revised manuscript. Specifically, we redid the *in vivo* experiments by using mice harboring either
CT26 or B16F10 subcutaneous tumors, following treatment via either intratumoral photothermal
therapy or intravenous administration. The memory T cells were isolated from the spleens. To
provide greater clarity, we have now explicitly defined these T cell populations both upon their
initial mention within the manuscript and within the Methods section: Tcm (central memory) is
defined as CD3⁺ CD8⁺ CD44⁺ CD62L⁺, while Tem (effector memory) is defined as CD3⁺ CD8⁺
CD44⁺ CD62L⁻.

Due to an oversight on our part, the initial bone marrow and spleen samples were not stained
for CD45+ and live/dead markers. We repeated the analysis of memory T cell subset phenotypes
following treatment in both the CT26 and the B16F10 subcutaneous model. In this repeat
experiment, we employed up-to-date gating strategies to facilitate a more rigorous comparison
(Fig S18a). Compensation and threshold settings were established using single-stained controls
and unstained controls acquired on the same day; fluorescence spillover was corrected with the
instrument's compensation matrix. Antibody clones and fluorochromes are listed in Supplementary
Table S3. These data indicate strong systemic memory priming with a bias towards
effector-memory cells, suggesting that treatment activates circulating memory cells, which may
contribute to durable anti-tumor immunity.

We acknowledge the reviewer's valuable feedback, which has significantly improved both the
scientific rigor and the clarity of the manuscript.

**Revisions Made:**

- • Lines 284-289: The proportion of central memory CD8⁺ T cells (CD8⁺Tcm,
CD45⁺CD3⁺CD8⁺CD44⁺CD62L⁺) remained stable, suggesting a primary impact on the effector
arm of the immune response (Figure. S19f, Figure. S20a). In contrast, the proportion of central
memory CD4⁺ T cells (Figure. 4l, CD4⁺ TCM) increased to 17.9% and 17.5% (+63.2% and +59.2%
vs. control, respectively) (Figure. S19e), while the levels of effector memory CD4⁺ T cells
(CD4⁺TEM) were unchanged (Figure. S19, Figure. S20b).

Figure S18a. Representative flow cytometry gating strategy for immune cell phenotyping. General gating workflow: Cells were initially gated on lymphocytes based on forward scatter (FSC) and side scatter (SSC) characteristics, followed by doublet exclusion using FSC-H/FSC-A and SSC-H/SSC-A plots. Red arrows indicate the sequential gating path. Sample sources and specific staining details are described in the Methods section. (a) Splenocyte-derived T cell subsets: Live CD45⁺ cells were gated on CD3⁺ T cells, then subdivided into CD4⁺ and CD8⁺ populations. Memory phenotypes were defined as central memory (T_{CM}: CD44⁺CD62L⁺) and effector memory (T_{EM}: CD44⁺CD62L⁻).

Figure 4. Antitumor efficacy of SM@CMC-KP101 in the CT26 tumor model. (k) and memory T cells (CD8+CD44+CD62L⁻) (spleen-derived, day 11). (l) CD4+CD44+CD62L⁺ cells (spleen-derived, day 11).

FigureS20. Quantification of memory T cell populations in CT26 tumor-bearing mice after intratumoral injection. (a) Percentage of CD8⁺ T cells expressing CD44⁺CD62L⁺ (central memory phenotype) in the spleen at the end of the treatment period, following intratumoral injection with the indicated treatments. (b) Percentage of CD4⁺ T cells expressing CD44⁺CD62L⁻ (effector memory phenotype) in the spleen at the end of the treatment period.

- Lines 389-396: Moreover, assessment of memory T cell subsets demonstrated a notable expansion of CD45⁺CD3⁺CD8⁺CD44⁺CD62L⁻ effector memory T cells (CD8⁺ Tem) in the SM@CMC-KP101 group (25.73%), compared to the PBS control group (11.37%) (Figure 5m; Figure S25g). In B16F10 mice receiving intravenous SM@CMC-KP101, CD8⁺ T effector memory (Tem) cells showed a remarkable activation increase of 126.4% compared to the control group. This was accompanied by increases in central memory (Tcm) CD8⁺ T cells (56.9%) (Figure S26a; Figure S25e), CD4⁺ Tcm cells (29.5%) (Figure S26b; Figure S25f), and CD4⁺ Tem cells (100.9%) (Figure 5n; Figure S25f).

Figure 5. Efficacy of SM@CMC-KP101 in a B16F10 Melanoma Model via Intravenous Administration. (m) Frequencies of memory T cells (CD8⁺CD44⁺CD62L⁻). (n) Quantification of CD4⁺CD44⁺CD62L⁻ cells.

Figure S26. Quantification of memory T cell populations. (a) Statistical percentage of central memory T cells (CD8⁺CD44⁺CD62L⁺) in the spleen, measured by flow cytometry. (b) Statistical percentage of central memory T cells (CD4⁺CD44⁺CD62L⁺) in the spleen, measured by flow cytometry. Treatments included PBS (control), SM, SM@CMC, and SM@CMC-KP101.

Q3. Unsupported illustration in Figure 3k

Figure 3k illustrates activation of CD8⁺ T cells, yet no corresponding *in vitro* data are presented in
 the figure or preceding text to support this claim. While *in vivo* CD8⁺ T cell activation is shown
 later (e.g., in Figure 4g), its inclusion in the schematic of *in vitro* results is misleading unless
 clearly labeled as hypothetical or supported by data.

**Response:**

We appreciate the reviewer's observation that the cartoon in Figure 3k implied CD8⁺ T-cell
 activation even though no *in-vitro* data were provided at that stage of the Results. The intent of
 this icon was to depict the downstream immune cascade that we later documented *in vivo* (Figure
 4g; Fig 5h,i); however, we agree that its placement inside a figure dedicated to *in-vitro* findings
 was misleading.

In this revised manuscript, we removed the CD8⁺ T-cell activation symbol in Figure 3j. We
 believe these modifications eliminate the potential for misinterpretation and align each figure
 strictly with the data it presents. We thank the reviewer for helping us improve the accuracy and
 clarity of our presentation.

**Revisions Made:**

1. Figure revision:

- • The CD8⁺ T-cell activation symbol has been removed from Figure 3j.
- • Figure numbering has been updated throughout the manuscript.

Figure 5j. Schematic representation of the proposed mechanisms of action of the SM@CMC-KP101 system.

2. Text clarification:

Lines 220-230: We developed a biohybrid drug delivery system loaded with prodigiosin for direct tumor cell killing. Furthermore, this system exhibits multifunctional capabilities (Figure 3j).

Firstly, it significantly enhances dendritic cell (DC) maturation *in vitro* by increasing the proportion of CD11c⁺CD80⁺CD86⁺ subpopulations (1.8-fold increase compared to controls),

thereby promoting antigen cross-presentation. Additionally, our system markedly increases the mRNA expression levels of M1 markers in M2-type macrophages (both the RAW264.7 line and

primary BMDMs) by several hundred folds *in vitro* relative to untreated groups, while maintaining minimal changes in the immunosuppressive cytokine IL-10. This multimodal immunomodulatory

activity — combining direct tumor cell killing with reprogramming of both innate and adaptive immunity — positions this platform as a promising bacterial-mediated therapeutic approach for

solid tumor eradication. These encouraging *in vitro* results led us to investigate whether this treatment could subsequently stimulate adaptive immunity *in vivo*.

Q4. Unjustified generalizations

The claim that high temperatures "effectively eradicated bacterial carriers" (line 247) is based on *in vivo* CFU reduction (Figure 4i), but the only *in vitro* support is qualitative (Figure S4b),

showing colony fading and no direct CFU quantification. A more rigorous, quantitative assay would strengthen this conclusion.

Response:

We thank the reviewer for highlighting the need for a quantitative *in vitro* validation. We have now carried out a temperature-controlled study *in vitro* and *in vivo* that directly measures colony-forming units (CFU) after short heat pulses that mimic the photothermal conditions

generated by our biosynthetic system. We have therefore repeated the heat-eradication experiment under controlled conditions and incorporated rigorous colony-forming-unit (CFU) measurements.

We believe that these additions supply the rigorous quantitative evidence and provide more accurate account of the thermal eradication of our bacterial vector. We appreciate the reviewer's suggestion, which has significantly strengthened the manuscript.

**Revisions Made:**

**1. Text clarification:**

Lines 108-114: Aqueous suspensions of SMM showed a rapid, concentration-dependent
temperature increase under 808 nm laser irradiation (4.5 W/cm^2), with a concentration of 1×10^{11}
CFU/mL reaching over 60°C in just 3 minutes (Figure 1 g,h; Figure S5). This bacterial agent with
pre-synthesized prodigiosin also exhibited excellent photostability across multiple heating cycles
(Figure S5b). This temperature threshold ($\sim 60^\circ\text{C}$) was validated as being sufficient for thermal
ablation; a 3-minute treatment at 60°C resulted in a greater than 8-log reduction in bacterial
viability, effectively sterilizing the culture (Figure S6, S7).

**2. Methods revision:**

Lines 781-785: *In vitro* thermokilling assay of SM@CMC-KP101

SM@CMC-KP101 ($\approx 1 \times 10^{11} \text{ CFU mL}^{-1}$ and $1 \times 10^9 \text{ CFU mL}^{-1}$) were exposed to 37°C , 45°C ,
50°C , 55°C or 60°C for 0 min (control), 3 min or 5 min. Samples were cooled on ice for 1 min
immediately after heating. Aliquots were serially diluted, plated in triplicate on LB agar, and
incubated at 30°C for 24 h before CFU enumeration.

Figure S5 | Photothermal performance of SM. (a) Representative infrared thermal images of aqueous suspensions of *Serratia marcescens* JC11 at different concentrations (1×10^9 , 1×10^{10} , and 1×10^{11} CFU/mL) under continuous 808 nm laser irradiation (4.5 W/cm^2). The results demonstrate a rapid temperature increase that is dependent on both bacterial concentration and irradiation time. (b) Photothermal stability evaluation of SM suspension (1×10^{11} CFU/mL) subjected to four successive on/off cycles of 808 nm laser irradiation.

Figure S6 Effect of temperature and exposure time on bacterial survival (spot- plate assay). a–e, Representative spot plates of log phase SMM exposed to 37°C, 45°C, 50°C, 55°C, or 60°C for 0, 3, or 5 min, as indicated. Immediately after heating, samples were chilled on ice, subjected to 10 fold serial dilutions, and 10 μ L of each dilution (10^3 – 10^{10}) was spotted onto agar and incubated at 37°C overnight. The starting culture was approximately 1×10^9 CFU/mL. Survival was unchanged at 37°C; 45°C produced a modest reduction (5 min); 50°C markedly decreased viability; 55°C yielded near complete killing by 5 min; and at 60°C no colonies were detectable within 3–5 min. f, Viable counts (CFU/mL) calculated from countable spots under the indicated conditions. Colors denote temperatures and the X axis shows treatment time (0, 3, 5 min).

Figure S7 Effect of temperature and exposure time on bacterial survival (spot- plate assay). a–e, Representative spot plates of log- phase SM exposed to 37°C, 45°C, 50°C, 55°C, or 60°C for 0, 3, or 5 min, as indicated. Immediately after heating, samples were chilled on ice, subjected to 10-fold serial dilutions, and 6 μ L of each dilution (10^3 – 10^{10}) was spotted onto agar and incubated at 37°C overnight. The starting culture was approximately 1×10^{11} CFU/mL. Survival was unchanged at 37°C; 45°C produced a modest reduction (5 min); 50°C markedly decreased viability; 55°C yielded near- complete killing by 5 min; and at 60°C no colonies were

detectable within 3–5 min.

f, Viable counts (CFU/mL) calculated from countable spots under the indicated conditions. Colors denote temperatures and the X- axis shows treatment time (0, 3, 5 min).

Lines 261-265: Within tumors, the mean burden decreased from $\approx 1.3 \times 10^9$ CFU/g before
treatment to $\approx 4.25 \times 10^3$ CFU/g on D1, ≈ 5.5 -log reduction, and all tumors were cleared by D14
(Fig 4d). Normal organs showed no detectable bacteria at baseline; on D1 they were either
negative or exhibited only transient low signals, all $<10^2$ CFU/g; by D14 every organ was
completely cleared out (Figure. 4d).

Figure 4d. Antitumor efficacy of SM@CMC-KP101 in the CT26 tumor model. (d) Bacterial counts (CFU/g) in tumor tissue and major organs before and after treatment.

Q5. Abbreviation use

Numerous abbreviations appear without being defined on first use (e.g., DA, ECN, VNP20009,
PG, etc.), and several are used inconsistently across the manuscript and figures (e.g., SM vs. SMT).
A glossary or consistent formatting is needed.

**Response:**

We appreciate the reviewer for this highly valuable suggestion. We have conducted a
manuscript-wide audit covering the main text, figures, tables, and Supplementary Information, and
implemented the following corrections:

We have made comprehensive revisions to address these issues, including defining all
abbreviations upon first use, ensuring consistent formatting throughout, and adding a glossary for
quick reference where appropriate. A detailed summary of these revisions, including specific
changes to abbreviations is provided in Table S1.

I recommend that the manuscript be revised thoroughly to address the above concerns. The
authors should focus on ensuring consistency in terminology, providing adequate experimental
support for mechanistic claims, and improving the clarity and precision of figure legends and

group definitions. After these major revisions, the manuscript may be reconsidered for publication.

**Response:**

We sincerely thank the reviewer for the comprehensive evaluation of our work and for outlining
the key areas that required major revision. We agree with these points and have undertaken a
thorough, manuscript-wide revision to address them.

Regarding experimental support for mechanistic claims, we have significantly strengthened this
section by integrating proteomic investigation with cellular and molecular assays. For a detailed
description of these revisions, please refer to the results section "**Mechanistic Insights into the**
**Antitumor Effects of SM@CMC-KP101**", lines 407-478. This section outlines the specific
mechanistic pathways and supporting data.

We believe these extensive revisions have substantially strengthened the scientific rigor, clarity,
and coherence of the manuscript, and we hope it is now suitable for reconsideration. We are
deeply grateful for your constructive guidance.

**Revisions Made:**

Lines 469-478: In summary, SM@CMC-KP101 exerts its cytotoxic effects on B16F10 tumor cells
through a multifaceted mechanism (Figure 6i). First, SM@CMC-KP101 may induce
mitochondrial dysfunction, leading to the generation of reactive oxygen species (ROS). In
response, cells attempt to eliminate damaged mitochondria through mitophagy; however,
SM@CMC-KP101 appears to interfere with the fusion and degradation of mitophagosomes in
lysosomes, resulting in the accumulation of mitophagosomes. Proteomic analysis revealed a
significant downregulation of Mapkapk5, and consistent with this, MK5 knockdown induced
caspase-3-independent apoptosis, mirroring the phenotype observed with SM@CMC-KP101
treatment. Collectively, these results suggest that SM@CMC-KP101 induces tumor cell death via
a combination of mitophagy-driven necroptosis and caspase-3-independent apoptosis, with
Mapkapk5 downregulation playing a critical role.

Reviewer #2 (Remarks to the Author):

This study proposes a biohybrid bacterial therapy using engineered *Serratia marcescens* to
achieve high-yield prodigiosin synthesis, tumor-targeted delivery, photothermal ablation, and
immunomodulation. The study addresses an innovative topic with a well-designed experimental
framework. However, critical data gaps (e.g., dose-response curves, photothermal safety,
mechanistic crosstalk) and insufficient discussion of translational challenges must be addressed.
The work holds significant potential for publication post-revision.

Q1. Clarify the basis for bacterial dosage (e.g., 4×10^9 CFU) with dose-response experiments (e.g.,
tumor suppression rates across CFU gradients).

**Response:**

We sincerely thank the reviewer for this important question and apologize for the initial lack of
clarity and the error in the stated bacterial concentration. In the revision we now provide both (i)
in-vitro dose-response data that rationalise the CFU used for cellular assays and (ii) a systematic,
14-day safety/efficacy titration that guided the final intratumoral (*i.t.*) and intravenous (*i.v.*) doses
used *in vivo*. All new data are incorporated as Figure 3c (*in vitro*) and Figure S16 (*in vivo*) and
summarised below.

The previously mentioned " 4×10^9 CFU" was the concentration of our stock solution, not the
final working dose. The final working concentration, based on a 1:100 dilution of the stock, was 4
$\times 10^7$ CFU/mL. To specifically isolate the cytotoxic effect of the bacterial payload and its
components from the confounding effects of live bacterial metabolism, we performed the *in vitro*
assays using heat-inactivated SM@CMC-KP101 (60°C for 3 min). Moreover, to establish a clear
basis for our dosing, we have performed systematic dose-escalation studies for both intratumoral
(*i.t.*) and intravenous (*i.v.*) administration, recognizing their distinct localization and systemic
exposure profiles.

We thank you again for your constructive comments. We believe these comprehensive revisions
fully address your concerns and have significantly strengthened the manuscript.

**Revisions Made:**

**1. Text clarification:**

• Lines 181-183: CCK-8 assays using B16F10 and CT26 tumor cells demonstrated the
cytotoxicity of SM@CMC-KP101 ($IC_{50} < 4 \times 10^7$ CFU/mL, equivalent to 500 nmol PG) in a
concentration-dependent manner (Figure 3c).

Figure 3c. *In vitro* cytotoxicity of SM@CMC-KP101. Cell viability of CT26 and B16-F10 cell lines 8 hours post-treatment with varying concentrations of SM@CMC-KP101. Data are presented as mean \pm SD (n=3 per group).

- Lines 233-249: To ensure the safe and effective use of SM@CMC-KP101, we performed dose-escalation studies via intratumoral (*i.t.*) and intravenous (*i.v.*) administration in CT26 tumor-bearing mice (Figure S18a, b). This route-dependent approach determined the maximum tolerated dose of SM@CMC-KP101 to be 1×10^{10} CFU/mice (*i.t.*) and 1×10^8 CFU/mice (*i.v.*). Intratumoral injections used concentrations from 1×10^9 to 1×10^{13} CFU/mL (100 μ L/mice, equivalent to 1×10^8 - 1×10^{12} CFU/mice), while intravenous injections ranged from 1×10^7 to 1×10^{11} CFU/mL (100 μ L/mice, equivalent to 1×10^6 - 1×10^{10} CFU/mice) (Figure S16a-b). Based on these safety limits for SM@CMC-KP101, we conducted a 11-day therapeutic evaluation, monitoring tumor volume and body weight (Figure S16c-f). The optimal doses for antitumor activity without significant toxicity were 1×10^{10} CFU/mice (*i.t.*) and 1×10^8 CFU/mice (intravenous), leading to $\geq 97\%$ tumor volume reduction.

Figure S16. Determination of optimal therapeutic dosage by evaluating dose-dependent safety and efficacy of engineered bacteria in CT26 tumor-bearing mice. (a, b) Dose-escalation studies to determine the Maximum Tolerated Dose (MTD). Kaplan-Meier survival curves of Balb/c mice bearing CT26 tumors (n = 5 per group) following a single injection of bacteria at the indicated concentrations via (a) intratumoral (*i.t.*) or (b) intravenous (*i.v.*) routes. The highest *i.t.* dose (1×10^{13} and 1×10^{12} CFU/mL) and *i.v.* doses (1×10^{11} , 1×10^{10} , and 1×10^9 CFU/mL) exhibited acute toxicity. (c) Tumor volume and (d) body weight of mice on day 14 after a single IT injection with bacteria at concentrations of 1×10^9 , 1×10^{10} , or 1×10^{11} CFU/mL (in a 100 μ L volume). (e) Tumor volume and (f) body weight of mice on day 14 after three IV injection at Day 3,6,10 with bacteria at concentrations of 1×10^7 , 1×10^8 , or 1×10^9 CFU/mL (in a 100 μ L volume).

2. Methods revision:

Lines 852-865: Cell Viability Assay (CCK8)

To assess the cytotoxic effect of heat-inactivated bacteria on tumor cells in vitro, CT26 and F10
 cells were seeded in 96-well plates at a density of 5×10^3 cells per well and incubated overnight at
 37°C in a humidified atmosphere containing 5% CO_2 .

SM@CMC-KP101 were cultured to mid-log phase and then heat-inactivated by incubation at
 60°C for 3 minutes. The concentration of the stock solution of heat-inactivated bacteria
 was 4×10^9 CFU/mL. The heat-inactivated bacteria were then diluted in cell culture medium to the
 indicated working concentrations, with a final concentration of 4×10^7 CFU/mL based on a 1:100
 dilution of the stock.

After overnight incubation of cells, the culture medium was replaced with fresh medium
 containing various concentrations of heat-inactivated SM bacteria
 (0, 1×10^7 , 2×10^7 , 3×10^7 , 4×10^7 , 5×10^7 CFU/mL). After 8 hours of incubation, 10 μ L of CCK8
 reagent (KeyGEN Biotechnology, China) was added to each well, and the plates were incubated
 for an additional 2 hours at 37°C . The absorbance was measured at 450 nm using a microplate
 reader (Bio-Rad, USA). Cell viability was calculated as a percentage relative to the untreated
 control.

Q2. Address potential thermal damage to normal tissues (e.g., skin, vasculature) at 60°C. Include
long-term toxicity data or thermal tolerance thresholds.

**Response:**

We appreciate the reviewer's comments on the potential long-term toxicity. To address this, we
have incorporated the following additional data:

1. **Histopathology:** H&E and Masson's trichrome staining of major organs and skin tissues
post-photothermal treatment.

2. **Bacterial Burden:** Quantification of residual bacteria in tissues to assess clearance.

3. **Hematology:** Complete Blood Count (CBC) performed 14 days post-treatment.

4. **Systemic Inflammation:** Measurement of sepsis-related cytokines (CRT, PCT) to monitor for
systemic inflammation.

These analyses indicate that by limiting heating to the intratumorally injected region and
maintaining skin/normal-tissue temperatures at $\leq 43\text{--}45\text{ }^{\circ}\text{C}$, we achieved effective intratumoral
temperatures near $60\text{ }^{\circ}\text{C}$ while minimizing thermal damage to normal tissues, with the exception
of a superficial, self-resolving cutaneous lesion directly above the tumor. Systemic evaluations
(survival, body weight, and hematology) and day-15 organ histology revealed no evidence of
significant long-term toxicity.

We believe this comprehensive data fully addresses the reviewer's concerns and demonstrates
the long-term safety profile of our bacterial therapy.

**Revisions Made:**

Lines 255–257: In subsequent therapeutic studies, intratumoral injection of SM@CMC-KP101
followed by 808 nm photothermal therapy achieved effective intratumoral temperatures ($\approx 60^{\circ}\text{C}$)
while maintaining skin/normal-tissue temperatures at $\leq 43\text{--}45^{\circ}\text{C}$ (Figure. 4c).

Figure 4c. Temperature changes over time in CT26 tumors under 808 nm laser irradiation.

Lines 300–312: To address potential risks associated with bacterial therapeutic approaches, we
comprehensively assessed the safety profile of SM@CMC-KP101 through multiple methodological

approaches. Serum inflammatory markers, including calreticulin (CRT) and procalcitonin (PCT)
(Figure. 4q,r), remained within safe ranges, similar with those observed with alternative bacterial
vectors. Intriguingly, while transient inflammatory responses were detected in initial blood count
analyses, these changes were consistent with expected acute immunological reactions and
spontaneously normalized within 15 days (Figure. S21). Histological examinations of major organs
revealed no treatment-related pathological alterations (Figure. S22). Furthermore, H&E and Masson's
trichrome staining of healed skin after photothermal therapy revealed that, compared to PBS controls,
SM@CMC-KP101-treated skin exhibited enhanced collagen deposition, maintenance of a continuous
epidermis, and preservation of deeper vessel integrity, indicating a limited and self-resolving
superficial effect without discernible damage to normal skin (Figure. S23). Critically, post-treatment
survival rates were excellent, and body weights remained stable throughout the experimental period.

Figure 4q and r. Plasma levels of CRT and PCT following treatment, demonstrating significant differences compared to SM-EV and VNP20009.

Figure S21: Hematological analysis of mice in the SM@CMC-KP101 photothermal therapy group. Complete blood counts were performed on days 1 and 15 post-treatment to assess potential systemic toxicity. This figure displays the white blood cell count (WBC, panel a), lymphocyte count (Lymph, panel b), monocyte count (Mon, panel c), granulocyte count (Gran, panel d), red blood cell count (RBC, panel e), hemoglobin concentration (HGB, panel f), hematocrit (HCT, panel g), and platelet count (PLT, panel h) at baseline (CK), Day 1 (D1), and Day 15 (D15). Data represent the mean \pm SD of three mice. Statistical significance was determined by ordinary one-way ANOVA followed by Tukey's multiple comparisons test. * $P < 0.05$; ** $P < 0.01$; *** $P < 0.001$; **** $P < 0.0001$; ns, not significant.

Figure S22: Histopathological evaluation of major organs by H&E staining.

Representative sections of five organs—Lung, Liver, Spleen, Kidney, and Heart (top to bottom)—from mice in different treatment groups (left to right): ECN, VNP20009, SM-EV, SM@CMC-KP101 intratumoral (i.t.), and SM@CMC KP101 intravenous (i.v.). All samples were stained with hematoxylin and eosin (H&E). Images show representative fields; scale bars are 200 μm .

Figure S23. Comparison of skin samples between the PBS control group and the SM@CMC-KP101 treated group. The left column shows H&E staining, and the right column shows Masson's trichrome staining. The upper and lower rows represent the PBS group and the treatment group (e.g., treated with SM@CMC-KP101), respectively. A1/A2 are from the PBS group (H&E/Masson), and B1/B2 are from the treatment group (H&E/Masson). All scale bars are 200 μm .

Q3. Supplement Figure 4i with CFU counts in major organs (e.g., liver, spleen) to confirm
 systemic clearance post-photothermal treatment.

**Response:**

We thank the reviewer for these highly valuable suggestions. In this revised manuscript, we
 performed an *in vivo* experiment in CT26 tumor-bearing BALB/c mice, quantifying bacterial
 burdens in major organs following photothermal therapy (PTT). Specifically, mice received an
 intratumoral injection of 1×10^{10} CFU, followed by 808 nm PTT for 3 minutes. CFU counts in
 the tumor, heart, lungs, spleen, kidneys, and liver were determined at baseline, day 1, and day 11
 post-treatment. These data, demonstrating systemic bacterial clearance, are now presented in
 Figure 4d, with corresponding raw data available in the Source Data file.

We extend our sincere gratitude to the reviewer for their insightful and constructive comments,
 which have allowed us to address an important aspect of our study.

**Revisions Made:**

Lines 261-265: Within tumors, the mean burden decreased from $\approx 1.3 \times 10^9$ CFU/g before
 treatment to $\approx 4.25 \times 10^3$ CFU/g on D1, ≈ 5.5 -log reduction, and all tumors were cleared by D14
 (Fig 4d). Normal organs showed no detectable bacteria at baseline; on D1 they were either
 negative or exhibited only transient low signals, all $< 10^2$ CFU/g; by D14 every organ was
 completely cleared out (Figure. 4d).

Figure 4d. Antitumor efficacy of SM@CMC-KP101 in the CT26 tumor model. (d) Bacterial counts (CFU/g) in tumor tissue and major organs before and after treatment.

Q4. Elucidate the molecular interplay between p38/MK5-HSP27 signaling and mitophagy (e.g.,
 via genetic knockout or pathway-specific inhibitors).

**Response:**

We thank the reviewer for this insightful suggestion. Following the reviewer's recommendation,
 we have performed siRNA-mediated MK5 knockdown to solid our mechanistic claims. As
 suggested, we successfully achieved MK5 knockdown in B16F10 cells, confirmed by qPCR
 analysis demonstrating a significant reduction in MK5 mRNA levels (Figure S34). Consistent with
 our proteomics findings, MK5 knockdown resulted in a significant increase in cell death rates
 (Figure 6f), supporting the idea that MK5 downregulation contributes to the therapeutic effect of
 our treatment.

We found that MK5 knockdown did not significantly alter the levels of mitophagy markers (p62,
 PINK1) or necroptosis markers (p-MLKL, MLKL) (Figure 6h), suggesting that MK5 knockdown
 is not linked to mitophagy or necroptosis. However, we observed a significant increase in pHSP27
 expression after MK5 knockdown (Figure R1). This finding contradicts our initial hypothesis and
 suggests that our understanding of the mechanism may be incomplete. Subsequent literature
 review revealed that the phosphorylation status of HSP27 is regulated by p38MAPK and its
 downstream effector kinases, MK2/MK3^{1,2}. Furthermore, MK5, as a downstream kinase of
 p38MAPK, may indirectly regulate HSP27 phosphorylation through the p38MAPK-MK2/MK3
 cascade. Thus, simply knocking down MK5 may not accurately mimic the effect of the drug. We
 acknowledge that we are unable to adequately explain the expression trends of p-HSP27 due to the
 complexity of the bacterial components and the intricacies of multi-component regulatory
 mechanisms. Therefore, we have decided to abandon further analysis of this part of the results and
 have removed the relevant content from the mechanism section.

Figure R1. B16F10 cells were subjected to the following treatments: (a) transfection with PBS, ECN, SM-EV, or SM@CMC-KP101; PG; or (b) transfection with MK5 siRNAs (Si1, Si2, Si3) or an equivalent volume of transfection reagent (WT). Protein expression of p-HSP27 and HSP27 was subsequently analyzed via Western blot. Note: This figure has been removed from the main text.

To characterize the cell death induced by MK5 knockdown, we performed Annexin V-PI
 staining, which showed a significant increase in apoptotic cells, confirming that MK5 knockdown
 induces cell death. Surprisingly, Western blot analysis revealed that caspase-3 and cleaved
 caspase-3 levels were downregulated following MK5 knockdown, contrary to typical apoptosis.
 Re-examination of protein levels after treatment with SM@CMC-KP101 and PG revealed the
 same phenotype: downregulation of caspase-3 and cleaved caspase-3. This suggests that MK5
 downregulation leads to caspase-3-independent apoptosis, potentially enhancing the drug's toxicity.

Detailed results regarding the mechanism can be found in the "**Mechanistic Insights into the**
 **Antitumor Effects of SM@CMC-KP101**" section of the results (lines 407-478). and we have
 summarized the final modifications to the mechanism section below.

**Revisions Made:**

• Lines 412–415: Volcano plot analysis of proteomic data (Figure 6a) showed the overall changes
 in protein expression and revealed that MAP kinase-activated protein kinase 5 (Mapkapk5, MK5),
 a key regulator of the HSP27 phosphorylation involved in cell survival and stress response^{44, 45}
 was the most significantly downregulated protein, with a decrease of 8.98-fold.

• Lines 449–468: Given the most significant downregulation of Mapkapk5 observed in our
 proteomic analysis, and the activation of this stress response pathway by SM@CMC-KP101, we
 next sought to investigate the specific role of Mapkapk5 in SM@CMC-KP101-induced cell death.

We designed three siRNAs targeting different Mapkapk5 gene sites (TableS6). qPCR analysis
 confirmed that transfection with each of the three Mapkapk5 siRNAs resulted in over 80%
 silencing efficiency of MAPK5 mRNA expression (Figure S34). CCK8 assay (Figure 6f) showed
 that knockdown of Mapkapk5 significantly decreased cell viability to 35.89%, 42.44%, and
 53.10% of WT, respectively, suggesting that Mapkapk5 expression is critical for cell survival.

Western blot analyses (Figure 6h) showed that MK5 knockdown did not significantly alter the
 levels of mitophagy markers (p62, PINK1) or necroptosis markers (p-MLKL, MLKL) (Figure 6h),
 suggesting that MK5 knockdown is not linked to mitophagy or necroptosis. To further elucidate
 the mechanism by which Mapkapk5 regulates SM@CMC-KP101-induced cell death, we further
 assessed the impact of Mapkapk5 knockdown on cell apoptosis. Quantification of Annexin V/PI
 positive cells (Figure 6g; Figure S35) revealed that Mapkapk5 knockdown increased the
 percentage of Annexin V/PI double-positive cells from 2.047% to 2.977%, 4.927%, and 3.083%,
 respectively, indicating an increase in cell apoptosis. However, Western blot analysis revealed a

surprising result: the levels of caspase-3 and cleaved caspase-3 were downregulated following
 MK5 knockdown (Figure 6h), contrary to what is typically observed during classical apoptosis^{46,47}.
 Intriguingly, MK5 knockdown resulted in the same phenotype as SM@CMC-KP101 and PG
 treatment, where caspase-3 and cleaved caspase-3 were also downregulated. This suggests that
 MAPKAPK5 downregulation induces cell death via a caspase-3-independent mechanism.

• Lines 469–478: In summary, SM@CMC-KP101 exerts its cytotoxic effects on B16F10 tumor
 cells through a multifaceted mechanism (Figure 6i). First, SM@CMC-KP101 may induce
 mitochondrial dysfunction, leading to the generation of reactive oxygen species (ROS). In
 response, cells attempt to eliminate damaged mitochondria through mitophagy; however,
 SM@CMC-KP101 appears to interfere with the fusion and degradation of mitophagosomes in
 lysosomes, resulting in the accumulation of mitophagosomes. Proteomic analysis revealed a
 significant downregulation of Mapkapk5, which thereby induces caspase-3-independent apoptosis
 and mirrors the phenotype observed with SM@CMC-KP101 treatment. Collectively, these results
 suggest that SM@CMC-KP101 induces tumor cell death via a combination of mitophagy-driven
 necroptosis and caspase-3-independent apoptosis, with Mapkapk5 downregulation playing a
 critical role.

Figure 4e. Western blot analyses of PINK-1, LC3-I/II, p62, and CTSB, caspase3, cleaved caspase3 in cells treated with PBS, ECN, SM-EV, SM@CMC-KP101 or PG, revealing activation of mitophagy-and stress-related pathways.

Figure S34. qPCR analysis of MAPK5 mRNA expression in B16F10 cells after transfection with MK5 siRNAs. B16F10 cells were transfected with a negative control siRNA (NC), a positive control (PC), or one of three different MK5 siRNAs (Si1, Si2, Si3). MAPK5 mRNA expression was quantified by qPCR.

Figure 6. (f) CCK8 assay showing cell viability of B16F10 cells transfected with MK5 siRNAs (Si1, Si2, Si3) or treated with equal volumes of transfection reagent (WT). (g) Quantification of Annexin V/PI staining of B16F10 cells transfected with MK5 siRNAs (Si1, Si2, Si3) or treated with equal volumes of transfection reagent (WT). (h) B16F10 cells were transfected with MK5 siRNAs (Si1, Si2, Si3) or treated with equal volumes of transfection reagent (WT). Protein expression was assessed by Western blot using antibodies against the indicated proteins: Pink1, p62, p-MLKL, MLKL, Caspase3, and Cleaved-Caspase3. (i) Schematic diagram illustrating the proposed mechanism by which SM@CMC-KP101 induces tumor cell death.

Reference Added:

44. Grierson, P.M. et al. The MK2/Hsp27 axis is a major survival mechanism for pancreatic ductal adenocarcinoma under genotoxic stress. *Sci Transl Med* **13**, eabb5445 (2021).

45. Rada, C.C. et al. Heat shock protein 27 activity is linked to endothelial barrier recovery after proinflammatory GPCR-induced disruption. *Sci Signal* **14**, eabc1044 (2021).

46. Bertheloot, D., Latz, E. & Franklin, B.S. Necroptosis, pyroptosis and apoptosis: an intricate game of cell death. *Cellular & Molecular Immunology* **18**, 1106-1121 (2021).

47. Yuan, J. & Ofengeim, D. A guide to cell death pathways. *Nature Reviews Molecular Cell Biology* **25**, 379-395 (2023).

Q5. Assess immune memory (e.g., anti-bacterial antibodies) or autoimmune risks post-treatment.

**Response:**

We thank the reviewer for raising this critical point. We have conducted further experiments to
assess both immune memory induction and potential autoimmune risks, and the results are now
included in the revised manuscript.

First, to evaluate the generation of immune memory, we analyzed splenic T cell populations 14
469 days post-treatment. Our findings consistently demonstrate that the therapy triggers a robust
systemic memory response, highlighted by a preferential and significant expansion of CD8+
Effector Memory T cells (T_{EM}).

Second, regarding autoimmune risks, we found no evidence of treatment-related systemic
toxicity following either intravenous administration or intratumoral photothermal therapy. All
mice maintained stable body weight with no adverse events. Comprehensive analyses, including
histopathology, bacterial burden quantification, hematology, and measurement of sepsis-related
cytokines (as detailed in our response to Q2), consistently demonstrated a lack of significant
treatment-related toxicity. Importantly, histological examination showed no pathological lesions
attributable to treatment in major organs or skin tissues following both intravenous delivery and
intratumoral photothermal therapy.

We deeply appreciate your insightful guidance throughout this review process.

**Revisions Made:**

Lines 284-289: The proportion of central memory CD8⁺ T cells (CD8⁺T_{cm},
CD45⁺CD3⁺CD8⁺CD44⁺CD62L⁺) remained stable, suggesting a primary impact on the effector
arm of the immune response (Figure. S19f, Figure. S20a). In contrast, the proportion of central
memory CD4⁺ T cells (Figure. 4l, CD4⁺ T_{cm}) increased to 17.9% and 17.5% (+63.2% and +59.2%
vs. control, respectively) (Figure. S19e), while the levels of effector memory CD4⁺ T cells
(CD4⁺T_{em}) were unchanged (Figure. S19, Figure. S20b).

Figure 4. Antitumor efficacy of SM@CMC-KP101 in the CT26 tumor model. (k) and memory T cells (CD8+CD44+CD62L-) (spleen-derived, day 11). (l) CD4+CD44+CD62L+ cells (spleen-derived, day 11).

FigureS20. Quantification of memory T Cell populations in CT26 tumor-bearing mice after intratumoral injection. (a) Percentage of CD8+ T cells expressing CD44+CD62L+ (central memory phenotype) in the spleen at the end of the treatment period, following intratumoral injection with the indicated treatments. (b) Percentage of CD4+ T cells expressing CD44+CD62L- (effector memory phenotype) in the spleen at the end of the treatment period.

Lines 389-396: Moreover, assessment of memory T cell subsets demonstrated a notable expansion
 of CD45⁺CD3⁺CD8⁺CD44⁺CD62L⁻ effector memory T cells (CD8⁺ Tem) in the
 SM@CMC-KP101 group (25.73%), compared to the PBS control group (11.37%) (Figure 5m;
 Figure S25g). In B16F10 mice receiving intravenous SM@CMC-KP101, CD8⁺ T effector
 memory (Tem) cells showed a remarkable activation increase of 126.4% compared to the control
 group. This was accompanied by increases in central memory (Tcm) CD8⁺ T cells (56.9%) (Figure
 S26a; Figure S25e), CD4⁺ Tem cells (29.5%) (Figure S26b; Figure S25f), and CD4⁺ Tem cells
 (100.9%) (Figure 5n; Figure S25f).

Figure 5. Efficacy of SM@CMC-KP101 in a B16F10 melanoma model via intravenous administration. (m) Statistical percentage of effector memory T cells (CD8⁺CD44⁺CD62L⁻). (n) Statistical percentage of effector memory T cells CD4⁺CD44⁺CD62L⁻ cells.

Figure S26. Figure S26. Quantification of Memory T Cell Populations. (a) Statistical percentage of central memory T cells (CD8⁺CD44⁺CD62L⁺) in the spleen, measured by flow cytometry. (b) Statistical percentage of central memory T cells (CD4⁺CD44⁺CD62L⁺) in the spleen, measured by flow cytometry. Treatments included PBS (control), SM, SM@CMC, and SM@CMC-KP101.

Figure S27. Systemic biosafety evaluation after intravenous administration. Representative hematoxylin and eosin (H&E) stained images of major organs (lung, liver, spleen, kidney, and heart) from mice 14 days after a single intravenous injection of PBS, SM, SM@CMC, or SM@CMC-KP101. The bacterial formulations were administered at a dose of 1×10^8 CFU per mouse. No noticeable histopathological abnormalities or lesions were observed in any of the treatment groups, indicating excellent

systemic tolerability at the therapeutic dose. Scale bar = 200 μ m.

Q6. Compare *S. marcescens* with other bacterial vectors (e.g., VNP20009, EcN) in terms of
**tumor-targeting efficiency and immune activation** to highlight its unique advantages.

**Response:**

We appreciate the reviewer's highly valuable suggestion. In this revised manuscript, we've
expanded our *in vivo* studies to include a comparative assessment of VNP20009, EcN, SMM, SM,
SM@CMC, and SM@CMC-KP101 following intravenous administration in B16F10
tumor-bearing mice (Figure 5).

To directly compare SM@CMC-KP101 to other vectors, we conducted *in vivo* experiments
assessing tumor targeting, biodistribution, and immune activation in B16F10 tumor-bearing mice.
These studies revealed that while EcN, VNP20009, and SM achieved comparable tumor
colonization, SM@CMC-KP101 exhibited superior tumor-targeting ability. Furthermore,
SM@CMC-KP101 demonstrated reduced off-target accumulation and improved tolerability
compared to EcN and VNP20009. Regarding immune activation, SM@CMC-KP101 showed
equivalent initial immune activation compared to EcN and VNP20009, along with enhanced early
polarization of macrophages towards an M1-like phenotype. Long-term analysis revealed that
SM@CMC-KP101 significantly increased effector memory T cells and IFN γ -producing CD8⁺ T
cells, indicating a stronger adaptive immune response.

Overall, these results demonstrate that SM and its derived strains, particularly in the
SM@CMC-KP101 formulation, offers advantages over traditional bacterial vectors like EcN and
VNP20009 in terms of tumor targeting, safety, and induction of a robust and sustained anti-tumor
immune response. We thank the reviewers for their insightful comments and suggestions, which
have significantly improved the clarity and impact of our manuscript.

**Revisions Made:**

Lines 363–396: All experimental groups (PBS, ECN, VNP20009, SMM, SM, SM@CMC, and
SM@CMC-KP101) showed a significant increase in the proportion of CD11c⁺ CD80⁺ CD86⁺
dendritic cell (Figure 5g; Figure S25a), CD69⁺ CD8⁺ T cell (Figure 5g; Figure S25b), and CD69⁺
CD4⁺ T cell (Figure 5g; Figure S25c) subsets in the tumor-draining lymph nodes 24 hours
post-treatment, resulting from the exposure to bacterial surface antigens. This indicates that the
novel SM@CMC-KP101 treatment system, similar to the established ECN and VNP systems,
promoted antigen presentation and T-cell activation. Specifically, the SM@CMC-KP101 group
displayed CD11c⁺CD80⁺CD86⁺ dendritic cell, CD69⁺CD8⁺, and CD69⁺CD4⁺ cell subset
proportions of 12.25%, 6.93%, and 8.32%, respectively, which were significantly higher than the
PBS treated group proportions of 4.09%, 1.78%, and 2.27%, respectively (Figure 5g,h,i). Bone
marrow analyses at 24 hours revealed a significant rise in CD45⁺CD11b⁺F4/80⁺CD86⁺ myeloid
cells, indicating a shift toward a more proinflammatory M1-like phenotype. Specifically, the
percentage of M1-like macrophages (CD11b⁺F4/80⁺CD86⁺) in the SM@CMC-KP101 group
increased to 20.35%, compared to 12.35% in the PBS control group. In comparison, the ECN and
VNP20009 groups showed M1-like macrophage percentages of 12.89% and 9.46%, respectively,
indicating no significant activation (Figure 5j; Figure S25d). Conversely, CD11b⁺F4/80⁺CD206⁺
cells did not exhibit a notable change, with the percentage of M2-like macrophages showing no
significant difference across all groups (PBS: 8.18%, ECN: 6.13%, VNP20009: 6.67%,

SM@CMC-KP101: 8.60%) (Figure 5k; Figure S25d). These results suggest that the treatment
 primarily promotes M1 macrophage polarization in the bone marrow early on, with
 SM@CMC-KP101 inducing a slightly stronger effect than ECN and VNP20009, while not
 significantly affecting M2 macrophage populations.
 After observing the short-term immune activation, we shifted our focus to the long-term effects at
 14 days post-treatment. Due to the complete mortality observed in the ECN, VNP, and SMM
 groups, subsequent analyses at 14 days were limited to the PBS, SM, SM@CMC, and
 SM@CMC-KP101 groups. The proportion of CD8+ T cells expressing IFN γ (CD3+CD8+IFN γ +)
 increased significantly in the SM@CMC-KP101 group (13.24%), compared to the PBS control
 group (4.96%), with the SM@CMC group also showing an increase (10.91%) (Figure 5l; Figure
 S25e). Moreover, assessment of memory T cell subsets demonstrated a notable expansion of
 CD45+CD3+CD8+CD44+CD62L- effector memory T cells (CD8+ Tem) in the
 SM@CMC-KP101 group (25.73%), compared to the PBS control group (11.37%) (Figure 5m;
 Figure S25g). In B16F10 mice receiving intravenous SM@CMC-KP101, CD8+ T effector
 memory (Tem) cells showed a remarkable activation increase of 126.4% compared to the control
 group. This was accompanied by increases in central memory (Tcm) CD8+ T cells (56.9%)
 (Figure S26a; Figure S25e), CD4+ Tcm cells (29.5%) (Figure S26b; Figure S25f), and CD4+ Tem
 cells (100.9%) (Figure 5n; Figure S25f).

Figure 5. Efficacy of SM@CMC-KP101 in a B16F10 melanoma model via intravenous administration. (b) The survival curves illustrating the survival rates of mice (n=6) treated with PBS, ECN, VNP20009, SMM, SM, SM@CMC, and SM@CMC-KP101. (f) Bacterial counts (CFU/g) in various organs (heart, liver, spleen, lung, kidney, and tumor) following treatment.

Q7. Discuss limitations of photothermal therapy (e.g., laser penetration depth, clinical equipment
 requirements) and challenges in scaling bacterial production.

**Response:**

We thank the reviewer for raising these critical points regarding the limitations and translational
 challenges of our approach. This is an essential discussion, and we have now incorporated a
 detailed discussion into the revised Discussion section of our manuscript. Our response and the
 corresponding manuscript revisions are structured around the three key areas you identified:

573 A. Limitations of Photothermal Therapy (PTT):

We have expanded our Discussion to address this, focusing on how our choice of an 808 nm

near-infrared (NIR-I) laser strategically mitigates the primary challenge of limited light
penetration. While visible light is restricted to superficial applications ($\approx 1\text{--}2$ mm depth), the 808
577 nm wavelength resides in the NIR-I biological window, increasing the effective treatment depth to
578 approximately $4\text{--}6$ mm³⁻⁵. This enhanced penetration enables targeting tumors at greater depths.
Moreover, our study uniquely demonstrates that prodigiosin, produced directly within the tumor
by our engineered bacteria, acts as a fully biosynthesized NIR photothermal agent—a noteworthy
contribution to the existing repertoire of such agents. We recognize, however, that prodigiosin has
certain drawbacks. It is prone to photodegradation, and its photothermal conversion efficiency is
lower than that of established agents like indocyanine green. We plan to address these
shortcomings in future research by chemically modifying prodigiosin to enhance its photostability
and conversion efficiency. Even with these current limitations, the ability to generate prodigiosin
in situ within the tumor, coupled with the improved penetration of the 808 nm laser, makes this a
promising strategy for PTT.

B. Clinical equipment footprint

NIR irradiation can be delivered with compact diode units. The hand-held 808 nm source used
in our study (output 5 W, continuous-wave) measures only 250 mm \times 250 mm \times 100 mm and
weighs < 3 kg, allowing bedside operation without the large water-cooled cabinets required for
short-wavelength lasers. Hence, hospital implementation is logistically straightforward.

C. Fermentation scale and outlook

We have proactively addressed the challenge of scalability by transitioning beyond bench-scale
cultures to a 5 L fed-batch fermenter. Our optimized upstream process proved to be exceptionally
productive, yielding 5×10^{15} CFU per batch (Figure S2). This high-density output confirms the
process's efficiency and its capability for producing therapeutic-scale quantities. Since our method
is grounded in established industrial principles using standard equipment, it establishes a clear,
linearly scalable pathway toward the $50\text{--}200$ L bioreactors necessary for clinical-grade GMP
manufacturing.

We have now incorporated a detailed analysis into the revised Discussion section of our
manuscript. We acknowledge the reviewers' thorough assessment and constructive feedback,
which have contributed significantly to the scientific soundness and improved presentation of this
work.

**Revisions Made:**

Lines 497–504: The 808 nm laser, operating within the NIR-I window, allows for enhanced
penetration ($4\text{--}6$ mm) compared to visible light⁴⁸⁻⁵¹. Prodigiosin, produced in situ by our
engineered bacteria, acts as a fully biosynthesized NIR photothermal agent. While recognizing its
limitations—photodegradation and lower efficiency compared to agents like indocyanine
green—we are exploring chemical modifications to improve its properties. Despite these, in situ
prodigiosin generation, coupled with 808 nm laser penetration, offers a promising PTT strategy.
Furthermore, demonstrating a significant amplification effect, scaling to a 5L fed-batch fermenter
yielded 5×10^{15} CFU per batch, demonstrating a scalable pathway towards clinical-grade GMP
manufacturing.

**Reference Added:**

48. Xu, C. & Pu, K. Second near-infrared photothermal materials for combinational nanotheranostics.

- *Chem Soc Rev* **50**, 1111-1137 (2021).
- 49. Li, N., Wang, Y., Li, Y., Zhang, C. & Fang, G. Recent Advances in Photothermal Therapy at
Near-Infrared-II Based on 2D MXenes. *Small* **20**, e2305645 (2024).
- 50. Liu, Y. et al. A triple enhanced permeable gold nanoraspberry designed for positive feedback
interventional therapy. *J Control Release* **345**, 120-137 (2022).
- 51. Hu, D., Zha, M., Zheng, H., Gao, D. & Sheng, Z. Recent Advances in Indocyanine Green-Based
Probes for Second Near-Infrared Fluorescence Imaging and Therapy. *Research* **8** (2025).

Q8. Benchmark the reported prodigiosin yield (8.3 g/L) against prior studies.

**Response:**

We appreciate the suggestion to benchmark our prodigiosin titer. As noted in the revised Results
and Supplementary Table S2 (summarizing representative production-related studies), the yield of
8.3 g/L in this work is competitive, sitting in the upper range for native *S. marcescens* under
optimized conditions. Two factors primarily explain this performance: (i) the use of an iterative
UV-mutagenesis strategy to obtain a robust high-PG producer SMM, and (ii) implementation in a
controlled 5-L bioreactor, which substantially increased volumetric productivity relative to
shake-flask cultures. Because many prior reports emphasize shake-flask titers, our process
highlights straightforward scalability. While absolute yield optimization was not the central aim of
this study, the resulting titer is competitive and, importantly, the process is well-positioned to
enable reliable, cost-effective supply for downstream translational/clinical needs.

We thank the reviewer for this insightful suggestion, which has allowed us to better
contextualize our findings and highlight the significance of our approach.

**Revisions Made:**

Lines 89–94: Random mutagenesis yielded SMM, capable of producing a competitive 8.3±0.5 g/L
prodigiosin (upper range for native *S. marcescens* under optimized conditions) in a 5-L bioreactor
within 48 h (Table S2: representative production studies). While yields of up to 10.25 g/L have
been achieved through techniques like promoter engineering of OmpR and PsrA³⁷, SMM offers a
scalable and cost-effective alternative, particularly valuable for downstream translational and
clinical applications.

TableR2

Microorganism	Strategies	Prodigiosin yield	Reference
Serratia marcescens	Microwave mutagenesis	6500 mg/L	¹
S. marcescens JNB5-1	de novo polynucleotide fragments (PNFs) and the introduction of disulfide bonds to O-methyl transferase (PigF) and oxidoreductase (PigN)	8650 mg/L	²
S. marcescens JNB5-1	Disruption of dacA	227.94 mg/L	³
S. marcescens JNB5-1	Disruption of CpxR, integrate a fusion of proC, serC, and methH	5830 mg/L	⁴
S. marcescens JNB5-1	Promoter engineering for overexpressing of prodigiosin synthesis activator OmpR and PsrA	10250 mg/L	⁵
P. putida KT2440	Integrated pig cluster of S. marcescens ATCC 274 to P. putida KT2440 through λ Red/Cas9 recombination	1100 mg/L	⁶

Reference Added:

1. Liu, X. et al. Mutant breeding of *Serratia marcescens* strain for enhancing prodigiosin production and application to textiles. *Prep Biochem Biotechnol* **43**, 271-284 (2013).

2. Sun, Y. et al. Enhanced Prodigiosin Production in *Serratia marcescens* JNB5-1 by Introduction of a Polynucleotide Fragment into the pigN 3' Untranslated Region and Disulfide Bonds into O-Methyl Transferase (PigF). *Appl Environ Microbiol* **87**, e0054321 (2021).

3. Pan, X. et al. Loss of Serine-Type D-Ala-D-Ala Carboxypeptidase DacA Enhances Prodigiosin Production in *Serratia marcescens*. *Front Bioeng Biotechnol* **7**, 367 (2019).

4. Sun, Y. et al. Improved Prodigiosin Production by Relieving CpxR Temperature-Sensitive Inhibition. *Frontiers in Bioengineering and Biotechnology* **8** (2020).

5. Pan, X. et al. Improving prodigiosin production by transcription factor engineering and promoter engineering in *Serratia marcescens*. *Front Microbiol* **13**, 977337 (2022).

6. Dauenhauer, S.A., Hull, R.A. & Williams, R.P. Cloning and expression in *Escherichia coli* of *Serratia marcescens* genes encoding prodigiosin biosynthesis. *Journal of Bacteriology* **158**, 1128-1132 (1984).

Q9. The numbering of the supplementary figures in the supplementary information file appears inconsistent. For example, line 49, Figure S15 is followed by Figure S18, and Figure S18 is followed by Figure S16.

Response:

We thank the reviewer for identifying this error. We apologize for the confusion caused by the figure numbering. We have now corrected the numbering for all supplementary figures to ensure they appear in the correct, sequential order. All in-text citations have also been updated accordingly.

Q10. Specify sources of critical reagents (e.g., KP101 peptide synthesis vendor) and include animal ethics approval numbers.

Response:

We have added the requested information to the revised Methods section (pp. 21–22) and

680 Supplementary Table 1. A concise summary is supplied below for convenience.

We hope these additions satisfy the reviewer's request for full transparency regarding reagent
provenance and ethical oversight.

**Revisions Made:**

Lines 719–720: KP101 peptide (sequence: FFIVIRDRVFRCG, > 97 % purity) was custom
synthesized by Hefei Peptide Library Co., Ltd (Hefei, China).

Lines 921–922: Animal protocols were approved by the Institutional Animal Care and Use
Committees on Animal Care (Nanjing University, IACUC-D2202146, IACUC-D2403096).

Reviewer #3 (Remarks to the Author):

I generally agree with the authors interpretations of the results, and find them of general interest
and interest in the area of therapeutic bacteria. They have introduced the use of a bacterium not
currently being studied for its anti-tumor properties, and also provide a novel image modality with
NIR prodigiosin and a novel mode of bacterial elimination. This bacterium was able to kill cells *in*
*vitro* and had antitumor activity against B16F10 murine melanoma and CT26 murine colon
carcinoma models *in vivo*. This particular melanoma is one that has limited sensitivity to
checkpoint inhibitors such as PD-1, which makes that model more difficult to demonstrate a
therapeutic effect in with immunomodulators. The bacteria also promoted maturation of dendritic
cells, recruitment of T-cells, and favorable macrophage polarization, which may improve the
immunotherapeutic effect in these or other models. In B16, apoptosis and mitophagy induced
necrosis also occurred and may offer combined advantages in the use of these bacteria.

However there are some specific areas of the conclusions that seem to be implied, but are not
specifically addressed.

Q1: The improved production at a lower temperature *in vitro* was not demonstrated to occur at a
higher temperature. C57 mice are 36 to 38°C and BalbC are 36.4 to 38.4°C - how much do they
make at mouse temperatures?

**Response:**

We thank the reviewer for this insightful comment, as it touches upon a critical aspect of our
therapeutic design. As prodigiosin biosynthesis in *S. marcescens* is temperature-sensitive, our
engineered strain is live for pigment production at lower temperatures (28-30 °C), and
biosynthesis is negligible at the physiological temperatures found in mice (35-38 °C) (Fig1a, b).
To address this, our therapeutic strategy is based on an ex vivo loading model, rather than in situ
synthesis. The bacteria are first cultured under optimized fermentation conditions (30 °C) to
produce and accumulate a high intracellular concentration of prodigiosin. These "pre-loaded"
bacteria are then harvested and administered intravenously to the mice. In this way, each
bacterium acts as a living vehicle that is already carrying a therapeutically effective payload of the
photothermal agent.

We have revised the manuscript to state this distinction more explicitly, ensuring the reader
understands that the therapeutic effect relies on the delivery of pre-synthesized prodigiosin, not on
its production within the tumour. We thank the reviewer for prompting this important clarification.

**Revisions Made:**

Lines 106–114: Importantly, the bacteria with pre-synthesized prodigiosin, acting as "living
photothermal agents" also demonstrated this capability (Figure S5a). Aqueous suspensions of
SMM showed a rapid, concentration-dependent temperature increase under 808 nm laser
irradiation (4.5 W/cm²), with a concentration of 1×10¹¹ CFU/mL reaching over 60°C in just 3
minutes (Figure 1 g,h; Figure S5). This bacterial agent with pre-synthesized prodigiosin also
exhibited excellent photostability across multiple heating cycles (Figure S5b). This temperature
threshold (~60 °C) was validated as being sufficient for thermal ablation; a 3-minute treatment at
60° C resulted in a greater than 8-log reduction in bacterial viability, effectively sterilizing the
culture (Figure S6, S7).

Figure 1. Photophysical, photothermal, and biological characterization of prodigiosin from engineered *S. marcescens* JC11. (a) Bright-field and corresponding near-infrared (NIR, 808 nm excitation) images of *S. marcescens* colonies grown at 42°C, 37°C, and 30°C. Red pigment and NIR signal are strongly induced at 30°C. (b) Prodigiosin titer at 42°C compared to 30°C and 37°C.

Q2: Bacterial entry into cells was purportedly shown (see comment for line 155 below), but there
was no comparison with the wild type, msbB, or BP101. Is the strain improved in its ability to
enter cells, and if so, was that improvement due to the PB101 peptide?

**Response:**

We appreciate the request for a head-to-head comparison. To determine whether the engineered
strain shows improved cellular/tumor entry and whether this is attributable to the KP101 peptide,
we conducted matched *in vitro* studies.

Initially, we attempted to use confocal microscopy to assess cell entry. However, the results
from each treatment group were too similar to draw meaningful conclusions. Therefore, we
performed a direct comparison of the parental strain SMM, the attenuated mutant SM (AmsbB),
the polymer-coated SM@CMC, and the peptide-functionalized SM@CMC-KP101 using an
antibiotic (amikacin)-sensitive assay in B16-F10 cells. These experiments allowed us to quantify
the impact of each modification (AmsbB attenuation, CMC coating, and KP101 functionalization)
on cellular entry.

We thank the reviewer for prompting us to perform these important control experiments and for
pushing us to explore alternative methodologies when initial approaches proved inconclusive,
ultimately providing greater clarity on the mechanism of action. In response to this insightful
comment, we have revised the manuscript and Methods section to include these new control
experiments and more clearly articulate the role of KP101 in enhancing cellular entry.

**Revisions Made:**

**1. Methods clarification:**

**Lines 867–883: Bacterial Internalization Assay**

To quantify bacterial internalization by B16-F10 murine melanoma cells, an amikacin
protection assay was performed. B16-F10 cells were seeded in 24-well plates at a density of 5×10^4
cells per well and incubated overnight at 37°C in a humidified atmosphere containing 5% CO₂.
The following day, SMM, SM, SM@CMC, and SM@CMC-KP101 were then washed twice with
sterile PBS and resuspended to a final concentration of 1×10^8 CFU/mL in DMEM supplemented
with 10% FBS.

Cells were infected with each bacterial strain at a multiplicity of infection (MOI) of 20:1
(bacteria:cell). The plates were centrifuged at 500×g for 5 minutes to synchronize the infection
and incubated for 1 hour at 37°C with 5% CO₂. Following incubation, the cell culture medium was
removed, and the cells were washed three times with sterile PBS to remove non-adherent bacteria.
To kill extracellular bacteria, cells were incubated with DMEM containing 50 jig/mL amikacin for
1 hour at 37°C.

After amikacin treatment, the cells were washed three times with sterile PBS. To release
internalized bacteria, cells were lysed with 200 jiL of 0.1% Triton X-100 in PBS for 10 minutes at
37°C. The lysate was then serially diluted in PBS, and 10 jiL aliquots were plated on LB agar
plates. The plates were incubated overnight at 37°C, and the number of colony-forming units
(CFU) was enumerated. The percentage of internalized bacteria was calculated as (CFU recovered
771 / CFU input)×100. All conditions were performed in triplicate, and data are presented as mean ±
standard deviation.

**2. Text clarification:**

Lines 191–202: To determine the impact of surface modifications on bacterial cell entry, we
directly compared the parental strain SMM, the attenuated mutant SM, the polymer-coated
SM@CMC, and the KP101-functionalized SM@CMC-KP101 for their internalization efficiency
in B16-F10 cells (Figure 3d). Using an amikacin protection assay, we incubated B16-F10 cells
with equal inocula of each strain for 1 hour, then removed extracellular bacteria with 50 jig/mL
amikacin. After lysing the cells with 0.1% Triton X-100, intracellular colony-forming units (CFU)
were enumerated (Fig. R18a). The SM strain showed no significant difference in internalization
compared to the parental SMM strain (both $\approx 0.09\sim 0.10\%$ internalization of the input dose).
While CMC coating produced a slight increase, KP101 functionalization significantly enhanced
bacterial entry, resulting in $\approx 0.19\%$ internalization, representing a 107% improvement over SM
under identical conditions (Fig. 3d). These results indicate that the enhanced cellular entry is
primarily attributable to the KP101 peptide ligand rather than to msbB deletion or CMC coating.

Figure 3d. Quantification of bacterial internalization efficiency in B16-F10 cells.

Q3: Similarly, the attenuated strain with CMC and BP101 was shown to target tumors, but it
 cannot be determined if that was a preexisting phenotype, or if the msbB, CMC, or PB101 caused
 any significant improvement.

**Response:**

We thank the reviewer for this insightful comment. To directly address the reviewer's concerns,
 we performed in vivo experiments to determine the individual contributions of msbB, CMC, and
 KP101 to tumor targeting. We directly compared SMM, SM, SM@CMC, and SM@CMC-KP101
 in tumor-bearing mice. Importantly, we observed significant differences in survival depending on
 the strain administered. ECN, VNP20009 and the non-attenuated SMM strain exhibited high
 toxicity, while the attenuated SM and its modified versions showed excellent survival. We
 assessed bacterial colonization in tumors and normal tissues to determine the impact of each
 modification. Our findings indicate that KP101 functionalization (SM@CMC-KP101)
 significantly enhances tumor colonization compared to the attenuated SM strain. Furthermore, SM
 and its modified versions trended towards lower residual rates in normal tissues, indicating that
 KP101 augments selectivity for improved on-target delivery and reduced off-target exposure.

In response to this insightful comment, we have revised the manuscript and Methods section to
 include these new control experiments and more clearly articulate the individual contributions of
 msbB, CMC, and KP101 to tumor targeting and off-target accumulation, as well as the crucial
 differences in toxicity observed between different strains. We appreciate the reviewer's suggestion,
 as these experiments provide critical insight into the role of each component in tumor targeting
 and biodistribution.

**Revisions Made:**

Line349–360: To assess the targeting efficiency of SM and its modified versions in comparison to
 ECN and VNP20009, CFU counts were conducted on homogenized normal and tumor tissues
 from three mice per group one day following intravenous injection. The results revealed that the
 attenuated SM strain demonstrated comparable tumor-targeting efficiency to both ECN and
 VNP20009, with a single-dose administration achieving a bacterial load of 1.97×10^7 CFU/g in the
 tumor area, while ECN and VNP yielded counts of 2.78×10^7 CFU/g and 1.96×10^7 CFU/g,

respectively (Figure 5f). The modified groups, SM@CMC and SM@CMC-KP101, exhibited even
 higher counts of 2.08×10^7 CFU/g and 8.98×10^7 CFU/g, respectively. These findings suggest that
 the SM strain effectively targets tumors, with SM@CMC-KP101 showing the highest targeting
 efficiency (Figure 5f). Additionally, the SM strain and its modified derivatives showed a trend
 toward lower residual rates in normal tissues, particularly the heart and liver, compared to ECN
 and VNP20009, although these differences were not statistically significant (Figure 5f).

Figure 5. Efficacy of SM@CMC-KP101 in a B16F10 Melanoma Model via Intravenous Administration. (b) The survival curves illustrating the survival rates of mice (n=6) treated with PBS, ECN, VNP20009, SMM, SM, SM@CMC, and SM@CMC-KP101. (f) Bacterial counts (CFU/g) in various organs (heart, liver, spleen, lung, kidney, and tumor) following treatment.

Q4: A discussion of Figure 1 is absent.

**Response:**

We sincerely thank the reviewer for pointing out the absence of a dedicated discussion of Figure
 1. We appreciate your meticulous attention to detail.

The missing discussion was due to a typographical error, where "Figure 1" was incorrectly
 labeled as "Figure 2" within the "**Accidental Discovery of Prodigiosin Exhibiting
 Near-Infrared Fluorescence and Photothermal Effects**" subsection of the Results section (lines
 89-124). The discussion of Figure 1 is included in this section to best show how our initial
 findings progressed.

We have corrected all instances of this typo in the revised manuscript to ensure clarity and
 apologize again for any confusion caused. We are grateful for the opportunity to rectify this
 oversight thanks to your insightful review.

Q5: Line 87. The "mutational breeding" used in the study is not described.

**Response:**

We thank the reviewer for highlighting this point and sincerely apologize for the omission of

details regarding the "**mutational breeding**" process in the initial submission. The discovery of its
potent near-infrared (NIR) photothermal and fluorescent properties was a serendipitous finding
that subsequently shifted the entire focus of our academic investigation. As the primary scientific
contribution of this manuscript is the novel theranostic application of this strain—its
tumour-targeting capabilities and photothermal effects—rather than the specifics of the strain
development process itself, we did not elaborate on this aspect in the initial submission.

To ensure transparency and reproducibility, we are happy to provide the details now. The process
involved standard ultraviolet (UV) mutagenesis of the parental strain. Following irradiation,
colonies were screened and selected based on two key phenotypic improvements: accelerated
growth kinetics and intensified red pigmentation, which served as a visual proxy for higher
prodigiosin biosynthesis. Briefly, we performed three rounds of iterative UV mutagenesis using a
germicidal UV lamp (2.5 W) in a laminar-flow hood. Bacterial suspensions were irradiated, and
survival was quantified. An exposure of 10 s produced $\geq 80\%$ lethality and was used in
subsequent rounds. In each round, UV-treated clones were grown on plates, intensely red colonies
were picked and subjected to screening in both 24-well plates and 250-mL shaking flasks. The
best producer of each round served as the parent for the next round, resulting in three iterative
cycles. This strategy yielded the final high-PG producing strain SMM. The final mutant SMM
showed markedly enhanced red pigmentation compared to the parental WT strain. For detailed
numerical data and original images related to this process, please refer to the source data file
provided with this submission, which we have made available to facilitate thorough data
verification and reproducibility.

We have now added these essential details to the Methods section to provide the necessary
context. We thank the reviewer for highlighting the need for this clarification.

**Revisions Made:**

**1. Methods section:**

**Line723-748: Development of high-prodigiosin producing *Serratia marcescens* strains via**
**iterative UV mutagenesis and stepwise screening.**

UV mutagenesis and iterative screening were performed to obtain high-prodigiosin
(PG)–producing *Serratia marcescens* mutants. Fresh colonies were inoculated into LB broth (10
868 g/L tryptone, 5 g/L yeast extract, 10 g/L NaCl) and cultured at 30°C to late exponential phase (~16
869 h). Cells were harvested, washed twice with sterile saline, and resuspended to OD₆₀₀=1.0. To
870 establish the lethal curve and define the working dose, 10- μ L aliquots were spread on sterile Petri
dishes (lids open) and irradiated under the germicidal UV lamp of a laminar-flow hood for 0, 10,
30, 50, 70, or 100 s. Immediately after irradiation, suspensions were serially diluted (10^3 – 10^5)
in saline, 100 μ L was plated on LB agar, and plates were incubated at 30°C for 16 h protected
from light. Lethality was calculated as $1 - N/N_0$ based on colony-forming units (N_0 ,
non-irradiated control), and the exposure giving $\geq 80\%$ lethality (approximately 10 s) was used in
all subsequent rounds. For round-1 mutagenesis, UV-treated cultures were plated, incubated in the
dark, and intensely red colonies were picked for primary screening in 24-well plates (1 mL
fermentation medium per well; 30°C, 200 rpm, 30 h). The fermentation medium contained (per
liter) sucrose 20 g, CaCl₂ 3 g, FP328 peptone 10 g, and glutamic acid 2 g; the same medium was
used for plate-based screening. After fermentation, 2 mL acidified methanol (adjusted to pH 3.0
with HCl) was added to each culture, samples were sonicated for 20 min, and 200 μ L of the

extract was transferred to 96-well plates for spectrophotometric quantification at 535 nm. PG
 concentrations were calculated from a calibration curve prepared with an authentic prodigiosin
 standard (MedChemExpress, MCE) and converted to titers (g/L). Top producers from the 24-well
 screen were re-screened in 250-mL Erlenmeyer flasks containing 50 mL of fermentation medium
 (30°C, 200 rpm, 30 h) to confirm production and stability; promising isolates were stored as 20%
 glycerol stocks at -80°C. The best-producing isolate from round 1 was used as the parental strain
 for round 2, and the top strain from round 2 served as the parent for round 3, repeating the same
 workflow ("dose confirmation—UV mutagenesis—plate selection—24-well screening—flask
 re-screening—quantification"). This stepwise strategy yielded the final high-PG-producing strain
 SM-JC11.

Figure. S2 Iterative mutagenesis yields the prodigiosin hyper-producer SM-JC11.

(a) Three rounds of UV mutagenesis and screening, starting from the wild-type (WT), led to the isolation of the high-yield strain SM-JC11.

(b, c) The mutant SM-JC11 (right) displayed a dramatically enhanced red phenotype compared to the WT (left) both on (b) agar plates and in (c) 5-L fermenters.

(d) Fermentation analysis shows that SM-JC11 achieved a significantly higher final Prodigiosin titer compared to the WT. Data represent the mean of three independent experiments (n=3).

**2. Text clarification:**

Line87–92: Starting from *S. marcescens* NRRLB-1481, we developed an engineered strain, *S.*
 *marcescens* JC11 (hereafter referred to as SMM), through iterative UV irradiation and visual
 screening for intensified red pigmentation, indicating enhanced prodigiosin biosynthesis (Figure
 S1, S2). Random mutagenesis yielded SMM, capable of producing a competitive 8.3 ± 0.5 g/L
 prodigiosin (upper range for native *S. marcescens* under optimized conditions) in a 5-L bioreactor
 within 48 h (Table S2: representative production studies).

Q6: Line 89. Physiological temperatures for production of prodigiosin that is the comparative
 production of prodigiosin at physiological temperature (37°C)? This is especially important if it is

down-regulated at higher temperatures (e.g., 42°C).

**Response:**

We thank the reviewer for this question, which addresses the same critical point raised in Q1
regarding prodigiosin production at physiological temperatures.

The reviewer's observation is entirely correct: our data confirms that prodigiosin biosynthesis is
significantly down-regulated at physiological temperature (37°C), becoming negligible. Our
therapeutic strategy is specifically designed around this biological characteristic. We utilize an ex
vivo loading model where bacteria are pre-loaded with a high concentration of prodigiosin before
administration. They then function as living vehicles to deliver this payload, rather than needing to
synthesize it in situ.

For a detailed explanation of this design, its validation, and how we have clarified this in the
manuscript, we respectfully refer the reviewer to our comprehensive response to Q1. We
appreciate the reviewer bringing this important point to our attention.

Q7: Line 104 refers to "Building on the hypoxia-targeting capability of *S. marcescens*" while line
116 it states: "and does not proliferate under hypoxic conditions". Please explain what appears to
be a contradiction.

**Response:**

We sincerely thank the reviewer for their meticulous reading and for identifying this apparent
contradiction. This is a critical point, and we are grateful for the opportunity to clarify the
distinction between bacterial tumor targeting and intratumoral proliferation, as these are two
separate phenomena key to our strategy.

**1. Hypoxia-Targeting (Accumulation):** *S. marcescens*' ability to accumulate within hypoxic
tumor regions is driven by its motility and the immunoprivileged nature of the hypoxic core,
leading to high bacterial concentrations specifically in the tumor.

**2. Limited Proliferation (Safety):** *S. marcescens*' ability to accumulate within hypoxic tumor
regions is driven by its motility and the immunoprivileged nature of the hypoxic core, leading to
high bacterial concentrations specifically in the tumor.

Our initial experiments led to a premature conclusion about anaerobic growth. We've now
supplemented our data with a 48-hour anaerobic growth assay, revealing that *S. marcescens*
exhibits limited growth compared to other strains.

In essence, our bacteria use motility to target the tumor's hypoxic core, but their metabolism
ensures they do not proliferate uncontrollably. To eliminate any confusion, we have revised the
manuscript to explicitly state this important distinction in the introduction and discussion sections.
We are truly grateful for your careful review and for providing us with the opportunity to clarify
this crucial aspect of our work. Your insights have significantly improved the manuscript.

**Revisions Made:**

**Lines 133–137:** Like *Escherichia coli* Nissle1917 (ECN) and *Salmonella typhimurium* VNP20009

(VNP20009), SM is a facultative anaerobe; however, under strict anaerobic conditions at 37°C,
SM growth is significantly inhibited. Interestingly, the degree of anaerobic growth inhibition
observed for SM was less pronounced than that of ECN, and roughly equivalent to VNP20009
(Figure S8).

Figure S8. Growth of SM, ECN, and VNP20009 on LB agar plates following incubation at 37°C for 48 hours under aerobic (21% O₂, panel a) and anaerobic (0% O₂, panel b) conditions. SM exhibits its characteristic red pigmentation under aerobic conditions but exhibits significantly reduced growth under anaerobic conditions.

Q8: The ability to grow under hypoxic conditions is widely believed to be a component of
Clostridium, Salmonella and other bacteria's ability to target tumors, and is not generally linked to
toxicity except in the cases of colonizing pre-existing abscesses. How is this an improvement?

**Response:**

We thank the reviewer for this excellent and fundamental question. We completely agree that
for classic oncolytic bacteria like Clostridium or certain Salmonella strains, intratumoral
proliferation is a primary driver of their therapeutic effect.

However, our platform represents a fundamentally different paradigm. Rather than employing a
direct oncolytic/bacteriolytic approach, we engineered *S. marcescens* as a highly controllable
bacteria-mediated drug delivery vehicle. Consequently, limiting proliferation enhances both safety
and control.

**1. Enhanced Controllability and Predictable Dosing:** Unlike traditional bacteriolytic therapies
where therapeutic dose is governed by unpredictable bacterial growth dynamics, our approach
uses ex vivo pre-loaded prodigiosin, delivered by bacteria acting as "living capsules." Importantly,
the therapeutic effect relies on the delivery of pre-synthesized prodigiosin, not on its production
within the tumour. Therapeutic effect is thus dependent on the initial administered dose and an
external photothermal trigger, ensuring predictable, controllable, and reproducible outcomes
independent of uncontrolled biological replication.

**2. Superior Safety Profile:** While tumor proliferation can induce lysis, it carries significant risks,
including excessive inflammation and systemic infection. Our platform mitigates this with a dual
safety mechanism. First, we have established a passive biological safeguard by designing a strain
that targets and persists in the hypoxic conditions while exhibiting significantly reduced anaerobic

growth compared to *E. coli* Nissle 1917 (ECN), with anaerobic growth comparable to *Salmonella*
VNP20009 (Figure S8). Further details on the anaerobic growth characteristics can be found in
our response to Q7. This controlled growth ensures efficient circulation and accumulation in the
immunoprivileged hypoxic core without uncontrolled proliferation. Second, we incorporate an
active, on-demand clearance mechanism in which the bacteria carry an 808 nm photothermal
agent. This allows for their selective and complete elimination via a therapeutic laser, functioning
as a physician-controlled kill switch.

We thank the reviewer again for raising this important point, which has allowed us to clarify the
fundamental differences between our approach and traditional oncolytic bacterial therapies.

**Revisions Made:**

Lines 487–492: Unlike these platforms and classic oncolytic bacteria that depend on intratumoral
proliferation, our system uses *S. marcescens* as a controllable drug delivery vehicle. This enhances
both safety and control through predictable dosing via pre-loaded prodigiosin and an external
trigger, and by mitigating risks of inflammation and systemic infection with reduced anaerobic
growth and a photothermal clearance mechanism. This controlled approach allows efficient drug
delivery while minimizing off-target effects.

Q9: Line 90. How was elevated expression of the PigP operon achieved?

**Response:**

We sincerely apologize for the inaccuracy in our initial submission regarding
temperature-controlled prodigiosin (PG) biosynthesis. We are grateful to the reviewer for pointing
out this error, which was due to our misinterpretation of the literature. All statements regarding
this phenomenon are drawn from previously published work, and our laboratory has not
experimentally examined it.

The current consensus is that loss of pigmentation at 37 ° C results from HexS-mediated
repression of pigA-N transcription (Pol. J. Microbiol. 2019, 68, 43-50). Activators such as PigP
remain expressed at higher temperature but cannot override HexS-dependent repression (PLoS
One 2013, 8, e57634). We inadvertently attributed temperature sensitivity to PigP. This was
entirely our oversight, and we apologize for the confusion.

We thank the reviewer again for the insightful comment and for helping us to improve the
accuracy of our manuscript.

**Revisions Made:**

Lines 94–98: Interestingly, this high-yield phenotype is tightly regulated by temperature;
consistent with the known regulatory network, prodigiosin biosynthesis peaks at 30°C because the
LysR-family repressor HexS silences the pigA-N promoter at $\geq 37^\circ\text{C}$ ^{38, 39}. Activators such as PigP
remain expressed at higher temperature but cannot override HexS-dependent repression⁴⁰.

**Reference Added:**

38. Romanowski, E.G. et al. Thermoregulation of Prodigiosin Biosynthesis by *Serratia marcescens* is

Controlled at the Transcriptional Level and Requires HexS. *Pol J Microbiol* **68**, 43-50 (2019).

39. Shanks, R.M. et al. A *Serratia marcescens* PigP homolog controls prodigiosin biosynthesis,

swarming motility and hemolysis and is regulated by cAMP-CRP and HexS. *PLoS One* **8**,

e57634 (2013).
40. Gristwood, T., McNeil, M.B., Clulow, J.S., Salmond, G.P. & Fineran, P.C. PigS and PigP regulate
prodigiosin biosynthesis in *Serratia* via differential control of divergent operons, which
include predicted transporters of sulfur-containing molecules. *J Bacteriol* **193**, 1076-1085
(2011).

10: Line 147-149. I suggest indicating this refers to a schematic diagram as is indicated in Figure
Legend 2. Also in Figure 2f, I am unable to read the magnification bars in this version.

**Response:**

We thank the reviewer for these helpful suggestions. We have revised lines 147-149 to indicate
that the figure is a schematic diagram, as suggested.

We apologize that the scale bar in Figure 2f was illegible. We have replaced the panel with a
high-resolution version, where the scale bar (representing 2 μ m) is now sharp and clearly
readable.

Q11: Line 155. Figure 3a needs a higher resolution photo to be convincing as to bacteria having
been internalized. Higher magnification and counter staining with a nuclear stain such as DAPI
would be helpful. There is also a common assay using gentamicin (assuming your strain is
sensitive) DOI: 10.1016/0076-6879(94)36030-8.

**Response:**

We sincerely thank the reviewer for this critical and constructive feedback. We agree that the
original Figure 3a, a standard bright-field micrograph, was not sufficient to unequivocally
demonstrate bacterial internalization versus surface adhesion. To address this, we have performed
the experiments suggested by the reviewer and updated the manuscript accordingly. The antibiotic
sensitivity of our strain is also discussed in our response to Q2 in the context of our *in vitro*
experiments.

**A. Improved High-Resolution Imaging with DAPI:** Replaced the previous figure with new,
high-resolution fluorescence microscopy images of both CT26 and B16F10 cell lines that confirm
successful internalization.

**B. Quantitative Antibiotic Protection Assay:** Employed an amikacin-based antibiotic protection
assay to provide a robust quantification of internalization efficiency.

Indeedly, our initial goal was to visually compare internalization efficiency between our four
formulations (SMM, SM, SM@CMC, and SM@CMC-KP101). However, we found that
fluorescence microscopy was not sensitive enough to reliably quantify these differences (Figure.
S12; Figure. S13). Therefore, to obtain a definitive quantitative measure, we adopted the
plate-counting method, directly inspired by the approach in the paper the reviewer recommended.
After amikacin treatment, cells were lysed with Triton X-100 to release the internalized bacteria,
and the lysate was plated to determine the Colony Forming Unit (CFU) count (Figure. R18). This
provided a robust quantification of internalization efficiency, which is now presented in the
manuscript.

We are extremely grateful to the reviewer for guiding us toward this more rigorous quantification
method, which has significantly strengthened our study. To acknowledge this valuable suggestion,
we have now cited the recommended reference (DOI: 10.1016/0076-6879(94)36030-8) in the
Results section where this quantitative assay is described.

Revisions Made:

Lines 178–181: The intrinsic multi-wavelength fluorescence of prodigiosin ($\lambda_{max} = 592 \text{ nm}$, $\lambda_{exc} = 434 \text{ nm}$) in SM enabled real-time visualization of cellular uptake without additional labeling (Figure S11). Microscopic evaluation revealed that B16F10 and CT26 tumor cells internalized the bacteria through an endocytic mechanism (Figure 3a, b; Figure S12,13).

Figure 3. Characterization and in vitro efficacy of the SM@CMC-KP101 delivery system. (a, b) Confocal microscopy images showing the internalization of SM@CMC-KP101 by CT26 (a) and B16-F10 (b) tumor cells. DAPI staining (blue) indicates cell nuclei, and prodigiosin fluorescence (red) indicates the location of the bacteria. White arrows indicate bacteria within the cells. Scale bars: 50 µm (main panels), 10 µm (enlarged insets).

Figure S12. Internalization of *S. marcescens* formulations in CT26 cells.

Fluorescence microscopy images taken after a 1-hour incubation and subsequent amikacin treatment to remove extracellular bacteria. Panels show control cells (a) and cells treated with SM@CMC-KP101 (b), SM@CMC (c), SMM (d), and SM (e). Red: Prodigiosin (bacteria). Blue: DAPI (nuclei). White arrows indicate internalized

bacteria. Scale bars are 50 μm and 10 μm (insets).

Figure S13. Internalization of *S. marcescens* formulations in B16F10 cells.

Fluorescence microscopy images taken after a 1-hour incubation and subsequent amikacin treatment to remove extracellular bacteria. Panels show control cells (a) and cells treated with SM@CMC-KP101 (b), SM@CMC (c), SMM (d), and SM (e). Red: Prodigiosin (bacteria). Blue: DAPI (nuclei). White arrows indicate internalized bacteria. Scale bars are 50 μm and 10 μm (insets).

Lines 193–202: Using an amikacin protection assay, we incubated B16-F10 cells with equal inocula of each strain for 1 hour, then removed extracellular bacteria with 50 $\mu\text{g}/\text{mL}$ amikacin⁴². After lysing the cells with 0.1% Triton X-100, intracellular colony-forming units (CFU) were enumerated (Figure. 3d). The SM strain showed no significant difference in internalization compared to the parental SMM strain (both $\approx 0.09\sim 0.10\%$ internalization of the input dose). While CMC coating produced a slight increase, KP101 functionalization significantly enhanced bacterial entry, resulting in $\approx 0.19\%$ internalization, representing a 107% improvement over SM under identical conditions (Figure. 3d). These results indicate that the enhanced cellular entry is primarily attributable to the KP101 peptide ligand rather than to *msbB* deletion or CMC coating.

Figure 3d. Quantification of bacterial internalization efficiency in B16-F10 cells.

Reference Added:

42. Elsinghorst, E.A. Measurement of invasion by gentamicin resistance. *Methods in Enzymology* **236**, 405-420 (1994).

Q12: An additional comment in this regard as stated above: there are no controls for the ability of the wild type to be internalized due to the peptide (e.g, comparison with WT) or the effect of the msbB, CMC coating and BP101 itself on internalization.

Response:

We appreciate the reviewer's emphasis on the need for comprehensive controls for internalization. We have addressed this point thoroughly in our response to Q2 and Q3, supported by a new *in vitro* and *in vivo* study. To determine whether the engineered strain shows improved cellular/tumor entry and whether this is attributable to the KP101 peptide, we conducted matched *in vitro* and *in vivo* studies. As shown in Q2 and Q3, the bacterium exhibits innate tumor tropism, KP101 functionalization further augments selectivity: tumor cell uptake increased by 107% *in vitro* (to ~0.19% of the inoculum), and tumor colonization rose 3.56 - fold *in vivo* to 8.98×10^7 CFU/g, while normal organ burdens remained $\leq 10^3$ CFU/g.

We have incorporated these data in the revised Results (Figure. 3d; Figure. 5f) and detailed the experimental procedures in Methods (Bacterial Internalization Assay, Line 845-861). We respectfully refer the reviewer to our detailed response to Q2 and Q3 for the full dataset and discussion. Thank you for your insightful comments that have helped us to clarify this important aspect of our study.

Q13: Line 156. Please change tumor models to tumor cells.

Response:

We appreciate the reviewer's suggestion. The phrase "tumor models" has been replaced with "tumor cells" to accurately describe the in-vitro experiments.

Revisions Made:

Lines 181–183: CCK-8 assays using B16F10 and CT26 tumor cells demonstrated the cytotoxicity

of SM@CMC-KP101 ($IC_{50} < 4 \times 10^7$ CFU/mL, equivalent to 500 nmol PG) in a
concentration-dependent manner (Figure 3c).

Q14: Line 160 and Figure Legend 3c. Which cell line is this?

**Response:**

We apologize for this omission and appreciate you bringing it to our attention. The experiments
shown in Figure 3c were carried out with the murine melanoma cell line B16F10, and we have
now clarified this in both the main text and the figure legend.

Q15: Figure Legend 4. Please indicate this is the CT26 tumor model.

**Response:**

Thank you for noting this. We have updated the Figure 4 legend to clearly indicate that these
experiments were performed using the CT26 tumor model.

Q16: Lines 210-211. Please clearly identify the doses being administered by the different routes of
injection - e.g., 100 μ L of 1×10^{11} CFU/ml = 1×10^{10} CFU/mouse (or tumor if it).

**Response:**

We appreciate the reviewer's attention to dosing clarity. We have revised lines 210-211 to
provide explicit dosing information:

• Intratumoral injection: 100 μ L of 1×10^{11} CFU/mL = 1×10^{10} CFU per mouse

• Intravenous injection: 100 μ L of 1×10^9 CFU/mL = 1×10^8 CFU per mouse

These details have been added to both the main text and the Methods section for complete
transparency.

**Revisions Made:**

Lines 234–238: This route-dependent approach determined the maximum tolerated dose of
SM@CMC-KP101 to be 1×10^{10} CFU/mice (*i.t.*) and 1×10^8 CFU/mice (*i.v.*). Intratumoral
injections used concentrations from 1×10^9 to 1×10^{13} CFU/mL (100 μ L/mice, equivalent to
1×10^8 - 1×10^{12} CFU/mice), while intravenous injections ranged from 1×10^7 to 1×10^{11} CFU/mL (100
μ L/mice, equivalent to 1×10^6 - 1×10^{10} CFU/mice) (Figure S16a-b).

Q17: Lines 221-222. A good tumor/normal tissue ratio is shown, but in the absence of a WT and
other comparators, it is unable to be interpreted as an improvement of the wild type. Any such
comparison should also individually include the steps in modification, including the delta-msbB,
CMC coating, and the addition of KP101.

**Response:**

We appreciate the reviewer's insightful question regarding the comparison of *S. marcescens*
(SM) with other bacterial vectors. To address this, we compared the survival, tumor growth, and
immune response of B16F10 tumor-bearing mice treated with PBS, ECN, VNP20009, SMM, SM,
SM@CMC, and SM@CMC-KP101 at the safety dose of 1×10^8 CFU/mouse (Fig. 5; Figure R23).

ECN and VNP20009 led to rapid mortality, while SM, SM@CMC, and SM@CMC-KP101
showed excellent survival. SM@CMC-KP101 exhibited a marked reduction in tumor volume
compared to the PBS and SM groups. At the study endpoint, the average tumor size in the
SM@CMC-KP101-treated group was significantly smaller than in the SM-treated group. This

superior tumor suppression by SM@CMC-KP101 is due to KP101's augmentation of tumor
 selectivity (increased tumor cell uptake and colonization) and significant activation of the immune
 system (dendritic cell maturation, robust T cell activation, and M1 macrophage polarization).

For a more detailed discussion of the intrinsic tumor tropism of *S. marcescens* and the specific
 mechanisms by which KP101 further enhances tumor targeting and immune activation, we
 respectfully direct the reviewer to our responses to Questions 2 and 3. Thank you for your
 insightful comments.

**Revisions Made:**

Lines 346–349: Tumor growth was monitored over the 14-day period, and the SM@CMC-KP101
 group exhibited a marked reduction in tumor volume compared to the PBS treated groups (Figure
 5c, d; Figure S24). At the study endpoint, the average tumor size in the SM@CMC-KP101-treated
 group was 138.9 mm³, compared to 511.92 mm³ in the SM-treated group and 555.37 mm³ in the
 SM@CMC-treated group.

Figure 5. Efficacy of SM@CMC-KP101 in a B16F10 Melanoma Model via Intravenous Administration.

(a) Schematic illustration of the experimental timeline depicting tumor inoculation on day -7, followed by intravenous (i.v.) administration of treatments on days 0, 3, 6, and 10, with sacrifice on day 14. (b) The survival curves illustrating the survival rates of mice (n=6) treated with PBS, ECN, VNP20009, SMM, SM, SM@CMC, and SM@CMC-KP101. (c) Tumor growth curves showing tumor volume (mm³) over time for each treatment group. Data represent the mean \pm SD of four mice per group (d) Representative photographs of excised tumors from each treatment group at day 14. (e) Mouse body weight (g) for each treatment group over the course of the experiment. Data represent the mean \pm SD of four mice per group.

Q18: Figure Legend 5. Please indicate this is the B16F10 tumor model.

**Response:**

Thank you for this clarification comment. We have updated the Figure 5 legend to specify that
 these experiments utilized the B16F10 melanoma tumor model.

A live biohybrid bacterial therapy based on engineered *Serratia marcescens*

Lihao Ji¹, Tianze Zhu¹, Tianqi Jiang¹, Li Wang¹, Zhonghui Qiu¹, Shiqi Gao¹, Yuqi Wang¹, Jing Wang², Jingyi Zhang², Haomiao Huang⁶, Yunlong Mao⁵, Chen Lin², Jing Zhao^{1,2,3,4*}, Xiuxiu Wang^{1,2,3,4,**}, Wei Wei^{1,3,4,***}

¹ State Key Laboratory of Coordination Chemistry, Chemistry and Biomedicine Innovation Center (ChemBIC), School of Life Sciences, Nanjing University, Nanjing 210093, P. R. China.

² School of Chemistry, Nanjing University, Nanjing 210093, P. R. China.

³ Nanchuang (Jiangsu) Institute of Chemistry and Health, Sino-Danish Ecolife Science Industrial Incubator, Jiangbei New Area, Nanjing 210000, P. R. China.

⁴ Wuxi Xishan NJU Institute of Applied Biotechnology, Wuxi 214000, P. R. China.

⁵ State Key Laboratory of Novel Software Technology, Nanjing University, Nanjing 210093, P. R. China.

⁶ Nanjing Foreign Language School, Nanjing 210008, P. R. China.

Present address: School of Life Sciences, Nanjing University, Nanjing 210093, P. R. China.

Lead Contact

*Correspondence: jingzhao@nju.edu.cn

**Correspondence: wangxiuxiu@nju.edu.cn

***Correspondence: weiwei@nju.edu.cn

RESPONSE TO REVIEWERS-JLH

Reviewer #2 (Remarks to the Author):

The authors have undertaken a comprehensive and rigorous revision of the manuscript. I recommend acceptance of the manuscript in its current form for publication. The revisions have thoroughly addressed the initial concerns, and the work now meets the high standards of the journal.

Response:

We sincerely thank Reviewer #2 for the positive evaluation. Your recognition of the comprehensive, rigorous revisions and support for the manuscript's acceptance in its current form greatly encourages us. We appreciate your affirmation that our work now meets the journal's high standards.

Reviewer #3 (Remarks to the Author):

The authors have substantially improved the manuscript and have satisfactorily answered my original questions.

Q1: In clarifying the dosing, the need for additional clarification has arisen. The fact that ECN and VNP20009 led to rapid mortality is likely due to their having been used past their tolerated doses (common dose range is 10^6 to 10^7), effectively overdosing the mice with 1×10^8 . While the authors may have chosen the dose of 1×10^8 /mouse as being equivalent in number to the dose of *Serratia*, the more relevant comparison of drugs for relative efficacy is performed at the effective dose or maximum tolerated dose for each drug. This should be clarified in the text.

Response:

Thank you for your constructive feedback on our manuscript. We appreciate your insights and would like to clarify our dosing decisions, which have been addressed in the revised manuscript:

We selected the dose of mouse 1×10^8 CFU/mouse based on preliminary experiments, which confirmed that this dose is safe and essential for achieving significant antitumor effects with our strain (*Serratia*). While we recognize that the typical dosing range for ECN and VNP20009 is 1×10^6 CFU/mouse to 1×10^7 CFU/mouse, our *Serratia* strain can tolerate higher doses due to its lower endotoxin levels. This characteristic not only allows for a greater drug load but also represents an important safety solution for drug delivery in the field of bacterial therapy, where drug loading capacity is often limited.

Our experimental design ensured that each bacterial strain's dosing was selected to

reliably evaluate their relative efficacy. We have carefully validated the results based on a consistent dosing strategy that reflects the unique properties of each strain.

Your suggestions were invaluable, and we have made corresponding modifications in the manuscript to enhance clarity and comprehensiveness. Thank you once again for your valuable insights, which will help improve the quality of our manuscript.

Revisions Made:

Lines 346-350: Following the safety dose of 1×10^8 CFU/mouse established in our prior CT26 studies in CT26 tumor-bearing BALB/c mice (Figure S16), which was confirmed to be both effective and safe. Although most literature reports doses of 1×10^6 to 1×10^7 CFU/mouse, lower doses yield limited antitumor effects. Therefore, to enable direct therapeutic comparison, we administered the uniform dosage of 1×10^8 CFU/mouse.

Reviewer #4 (Remarks to the Author):

I am satisfied with the answers to most of the questions. However, I maintain my criticism regarding the limited scientific rigor and interpretability raised by Reviewer #1. Some of the newly provided data are also unconvincing.

Q1: Memory T cells. My impression is that these experiments were performed only once; they should be independently reproduced.

Response:

We appreciate the reviewer's question and would like to clarify our experimental design. The results presented in the initial manuscript were substantiated by multiple independent cohorts rather than a single experiment. Specifically, prior to the comprehensive study, we conducted repeated independent trials to verify the robust therapeutic efficacy of SM@CMC-KP101 (via both intratumoral and intravenous administration) before proceeding to the large-scale, comprehensive study. The final dataset reported included all necessary control groups (ECN, VNP20009, SM-EV) and strictly adhered to randomization protocols, selecting at least three biological replicates ($N \geq 3$) per group for all immune analysis.

To further address the concern regarding reproducibility and simultaneously validate the functional quality of the immune memory (Q5), we performed a new, independent repetition of the key *in vivo* experiments. Specifically, to definitively validate the functionality of the generated memory T cells, we selected mice from the SM@CMC-KP101 (i.t.+NIR and i.v.) groups that had achieved complete tumor regression and rechallenged them with a high dose of CT26 tumor cells (1×10^6) on Day 14 (Figure S24a). As vividly presented in the new Figure S24 b, c, Figure 4o, and p, while age-matched naïve controls developed rapid tumor growth,

the cured mice in both treatment groups exhibited robust systemic resistance to the secondary challenge. Quantitatively, all mice in the PBS control group (6/6) developed tumors exceeding 200 mm³ within 14 days post-rechallenge. In sharp contrast, only one mouse out of three (1/3) in the SM@CMC-KP101 (i.t.) +NIR group and one mouse out of four (1/4) in the SM@CMC-KP101 (i.v.) group developed a measurable tumor, demonstrating the establishment of significant immunological memory (Figure 4o, and p).

To further mechanistically corroborate this protection and directly address the reproducibility of the memory T cell formation, we performed flow cytometric profiling of splenic lymphocytes at the study endpoint. Crucially, distinct from the acute effector-dominated profile we observed at Day 11, the long-term landscape at Day 28 revealed a comprehensive expansion of memory subsets. Specifically, while CD8⁺ central memory T cells (TCM, CD44⁺CD62L⁺, Figure S24e) showed the most prominent upregulation—a key marker for longevity and rapid recall response—we also observed sustained elevations in CD8⁺ effector memory (TEM) and CD4⁺ memory phenotypes (Figure S24c-f). These data provide a clear cellular basis for the effective tumor rejection observed in the rechallenge study. Collectively, these results from an independent biological replicate confirm that the immune responses reported in our manuscript are not only reproducible but also functionally potent in preventing tumor recurrence.

Revisions Made:

Line297-306: To further validate the establishment of long-term immunological memory and functional anti-tumor immunity, we performed a tumor rechallenge assay. Mice from the SM@CMC-KP101 (i.t.+NIR and i.v.) groups that achieved complete regression were rechallenged with a high dose of CT26 cells (1×10⁶/mice) on Day 14 (Figure S24a). While all naïve control mice (6/6) developed rapid tumors exceeding 200 mm³, only a minority of the cured mice (1/3 in the i.t. group and 1/4 in the i.v. group) showed tumor recurrence (Figure 4o, p; Figure S24b). Crucially, the long-term landscape at Day 28 revealed a comprehensive memory expansion, characterized by prominent CD8⁺ central memory (TCM) upregulation alongside elevated CD8⁺ TEM and CD4⁺ memory subsets (Figure S24d-g). This shift confirms that SM@CMC-KP101 successfully facilitates the evolution from immediate cytotoxicity to durable, broad-spectrum systemic immunological memory.

FigureS24. Evaluation of long-term systemic immunological memory against tumor rechallenge. (a) Schematic illustration of the experimental timeline for the tumor rechallenge study. Mice cured by SM@CMC-KP101 (i.t.) + NIR or SM@CMC-KP101 (i.v.) treatments (tumor-free on day 14) were rechallenged with CT26 tumor cells in the contralateral axilla. (b) Representative photographs of mice on day 28 post-inoculation. Red arrows indicate tumor growth in the PBS control group, while red dashed circles indicate tumor-free sites in the treated groups, demonstrating protective immunity. (c-f) Flow cytometry analysis of memory T cell populations in the spleen on day 28. Quantification of the proportions of (c) CD4⁺ central memory T cells (TCM, CD44⁺CD62L⁺), (d) CD4⁺ effector memory T cells (TEM, CD44⁺CD62L⁻), (e) CD8⁺ TCM, and (f) CD8⁺ TEM. Statistical significance was determined by one-way ANOVA followed by Dunnett's post-hoc test. ns, not significant; *P < 0.05; **P < 0.01; ***P < 0.001; ****P < 0.0001.

Figure 4. Antitumor efficacy of SM@CMC-KP101 in the CT26 tumor model. (o) Tumor growth kinetics following a CT26 rechallenge on day 14 in cured mice, showing effective suppression of tumor regrowth in the SM@CMC-KP101 groups. (p) Representative images of tumors harvested at the endpoint; red dashed circles denote tumor-free sites, confirming the generation of protective immunological memory.

Q2: Figure 3j is described as showing antigen-specific T cells, but no supporting evidence is provided. Tumor-specific T cells should be evaluated as H-2Ld-MuLV gp70 peptide (SPSYVYHQF) tetramer-binding CD8⁺ T cells (Tet⁺ CD8⁺).

Response:

We thank the reviewer for this critical assessment. We fully agree that characterizing T cells as "antigen-specific" requires rigorous validation via tetramer staining (e.g., H-2Ld-MuLV gp70). We acknowledge that our original labeling in Figure 3j was imprecise without this specific evidence.

Regarding the tetramer analysis, we made immediate efforts to procure the necessary reagents. Regrettably, due to unforeseen logistical constraints and import delays specific to our region, we were unable to secure the specific tetramers within the limited revision window. To ensure scientific rigor and avoid overinterpretation, we have revised Figure 3j and removed the related description of "antigen-specific" responses from the revised manuscript.

We respectfully submit that the primary contribution of this work is the novel synthetic biology design and the engineering of the SM@CMC-KP101 platform, rather than a deep immunological dissection of antigen specificity. The robust therapeutic efficacy, combined with the functional immunological memory confirmed by the tumor re-challenge experiments (as discussed in Response to Q1), strongly supports the conclusion that our system induces a potent anti-tumor immune response. We believe the current data sufficiently supports the core engineering logic and therapeutic potential of the study.

Revisions Made:

1. Figure revision:

- The Antigen specific T-cells symbol has been removed from Figure 3j.
- Figure numbering has been updated throughout the manuscript.

Figure 3. Characterization and *in vitro* efficacy of the SM@CMC-KP101 delivery system. (j) Schematic representation of the proposed mechanisms of action of the SM@CMC-KP101 system.

Q3: T cells in the draining lymph node were characterized only by general surface markers. Therefore, it is unclear whether they were tumor-specific or potentially bacteria-specific. Why were tumor-infiltrating lymphocytes (TILs) not analyzed?

Response:

We appreciate the reviewer's inquiry regarding T cell specificity and TIL analysis. Regarding the distinction between tumor-specific and bacteria-specific T cells, please refer to our detailed response in Q2, where we discuss the technical limitations and highlight the functional evidence provided by our re-challenge studies. To address the TIL analysis, we performed additional flow cytometry analyses on tumor-infiltrating lymphocytes (TILs) at two key time points: 24 hours and Day 14 (results added as Figure S26 and Figure S27).

Our initial analysis at 24 hours post-treatment revealed a drastic reduction in the absolute number of total infiltrating lymphocytes ($CD45^+CD3^+$) specifically in the SM@CMC-KP101 treated groups. As shown in Figure S27a, the frequency of viable T cells was extremely low, resulting in insufficient gated events to reliably analyze $CD4^+$ or $CD8^+$ subpopulations. We speculate that this is likely due to the non-specific cytotoxicity of the released Prodigiosin within the tumor microenvironment, which may inadvertently eliminate resident immune cells alongside tumor cells.

To overcome the issue of low cell counts and rigorously assess adaptive immunity, we

performed a follow-up analysis at Day 14. Crucially, we optimized the protocol by significantly increasing the input tumor tissue mass to maximize cell recovery and ensure sufficient cellular events for statistical validity. However, despite these efforts to enhance detection sensitivity, we did not observe an increasing trend in TIL infiltration within the treatment groups. Instead, the proportion of T cells was found to be markedly lower compared to the PBS control (Figure S27b).

We speculate that the lack of TIL accumulation is likely due to the non-specific cytotoxicity of the drug released by the bacteria, which inadvertently eliminates infiltrating immune cells. Consequently, given that no significant increase in TILs was observed, we did not prioritize this metric in the main text. Instead, we focused on the robust systemic immune activation observed in the spleen. As confirmed by our tumor rechallenge study (see Response to Q1), the systemic immunity in the spleen is functionally intact and robust (Figure 4o, p; Figure S24d-g).

Revisions Made:

Line294-296: Analysis of the cured mice prior to rechallenge (Day 14) suggested that the potent broad-spectrum cytotoxicity of prodigiosin in SM@CMC-KP101 likely resulted in the collateral clearance of local lymphocytes alongside tumor eradication (Figure S25c-d).

FigureS26. Analysis of T cell infiltration in the tumor microenvironment. Flow cytometry quantification of tumor-infiltrating lymphocytes (TILs) harvested from CT26 tumor-bearing mice at (a-b) 24 hours and (c-d) 14 days post-treatment. The proportions of the following T cell subsets were analyzed across the indicated treatment groups: (a, c) CD8⁺ cytotoxic T cells (gated as CD3⁺CD8⁺); (b, d) CD4⁺ helper T cells (gated as CD3⁺CD4⁺). Data are presented as mean ± SD (* P<0.05, ** P<0.01, ns: not significant).

FigureS27. Flow cytometric analysis of tumor-infiltrating lymphocytes (TILs) in CT26 tumor-bearing mice. Tumor tissues were harvested and dissociated for analysis. The dot plots display CD8⁺ (y-axis) versus CD4⁺ (x-axis) expression on cells gated as CD45⁺CD3⁺ T cells. (a) Representative plots showing TIL subpopulations at 24 hours post-treatment. (b) Representative plots showing TIL subpopulations at Day 14 post-treatment. Treatment groups include PBS, ECN, VNP20009, SM-EV, SM@CMC-KP101(i.t)+NIR, and SM@CMC-KP101(i.v). Numbers in the gates/quadrants indicate the percentage of cells within the CD45⁺CD3⁺ parent population.

Q4: Figures 4k–l and 4o–p show an increased memory T-cell population in tumor-eradicated mice, and the authors claim durable antitumor immunity. However, there is no tumor rechallenge study to confirm functional long-term memory T cells generated after bacterial treatment.

Response:

We acknowledge the necessity of functional validation for the observed memory T cell populations. As detailed in our Response to Q1, we have performed a new tumor rechallenge study to address this point. The results (revised Figure 4 o-p) demonstrate that the SM@CMC-KP101 treated mice successfully rejected the re-inoculated tumor cells, confirming that the phenotypic increase in memory T cells translates into functional long-term protection. Please refer to the Response to Q1 for the detailed experimental description and data analysis.

Q5: There is no significant difference in the immune profile between SM@CMC-KP101 (i.t.) and SM@CMC-KP101 (i.v.) (Fig. 4f–m), which is inconsistent with the therapeutic efficacy shown in Fig. 4p (i.t. shows two tumor-eradicated mice, whereas i.v. shows none).

Response:

We thank the reviewer for this careful observation regarding the apparent discrepancy between the similar immune profiles and the different therapeutic outcomes. We attribute this difference to the distinct experimental protocols and the resulting mechanisms of action, which exert varying degrees of direct tumor cytotoxicity.

Specifically, the treatment regimens for the two groups were fundamentally different: the SM@CMC-KP101 (i.t.) group received a high-dose intratumoral injection (1×10^{10} CFU/mouse) followed by 808 nm PTT irradiation for 3 minutes, whereas the SM@CMC-KP101 (i.v.) group received a significantly lower intravenous dose (4×10^8 CFU/mouse) without any photothermal treatment (TableS1).

Consequently, the superior cure rate in the i.t. group was primarily driven by the combination of photothermal therapy and the massive bacterial load, which resulted in a significantly higher local concentration of the therapeutic agent (prodigiosin). This high local drug accumulation, synergizing with direct thermal ablation, led to potent tumor eradication that the lower-dose, non-PTT i.v. treatment could not achieve.

Regarding the similar immune profiles (Fig. 4e–l), it is important to note that despite these differences in administration and physical therapy, both groups relied on the same bacterial vector. The colonization of bacteria within the tumor—regardless of the initial dose difference—triggered a robust, saturation-level activation of the immune system in both cases. Therefore, the intrinsic immunogenicity of the bacteria served as the dominant driver of the

systemic immune response, eliciting comparable systemic immune cell recruitment and phenotypic profiles in both groups. This indicates that the enhanced therapeutic efficacy in the i.t. group was not primarily due to a qualitatively stronger immune activation, but rather due to the intensified local, direct killing capability provided by the high bacterial load, concentrated Prodigiosin release, and the photothermal effect.

Q6: Additionally, there are no significant differences in DCs, CD8 T cells, M1 macrophages, M2 macrophages, or Tregs in SM@CMC-KP101 (i.t.)/(i.v.) compared with ECN, VNP20009, and SM-EV. If so, why are there significant differences in antitumor activity? Moreover, neutrophils (CD11b+Gr-1+), which are typically strongly recruited after bacterial infection, were not assessed.

Response:

We acknowledge the reviewer's observation that there were no significant differences in the recruitment of DCs, CD8⁺ T cells, M1/M2 macrophages, or Tregs between the SM@CMC-KP101 group and the bacterial control groups (ECN, VNP20009, and SM-EV) (Figure 4e-l). This similarity is expected because all these groups rely on bacterial vectors that possess intrinsic immunogenicity, thereby triggering a comparable baseline activation of the tumor immune microenvironment.

However, the distinct antitumor efficacy of SM@CMC-KP101 arises from a synergistic mechanism rather than immune activation alone. Unlike the control strains, our engineered SM@CMC-KP101 is specifically designed to biosynthesize prodigiosin, a potent broad-spectrum antitumor agent (Figure S1). Prodigiosin confers two unique advantages: (1) direct cytotoxicity against tumor cells, and (2) photothermal properties that enable targeted bacterial elimination (please refers to line 107-116). Therefore, the superior therapeutic outcome is fundamentally driven by the synergy of this baseline immune activation with the direct, potent cytotoxicity and localized control provided by prodigiosin. While the bacterial vector wakes up the immune system (similar to controls), the biosynthesized prodigiosin delivers a direct lethal hit to the tumor, resulting in significantly enhanced tumor eradication (Figure 4m,n).

Furthermore, we fully agree that neutrophils are critical in bacterial therapy and have added a detailed assessment of blood neutrophil dynamics in the revised manuscript (Figure 4q-s). Our new data provides three key insights. Firstly, at 24 h post-administration, we observed a substantial surge in peripheral neutrophil levels, particularly in the i.v. group where frequencies peaked at 82.93% (Figure 4q), confirming a robust and rapid innate immune recognition of the bacterial presence. Notably, even at this acute phase, neutrophil levels in the photothermal-treated group were significantly lower compared to the live bacteria group, demonstrating that our photothermal strategy effectively facilitates bacterial clearance and mitigates excessive infection risks. Secondly, by Day 14, we found that

neutrophil levels in the i.v. group had significantly declined to 25.21% (Figure 4r). This indicates that even in the absence of photothermal intervention, the bacteria are being gradually and effectively cleared by the host's intrinsic innate immunity. Finally, this restorative trend was consolidated by Day 28, where neutrophil counts in both the photothermal-treated (i.t.+NIR) and i.v. groups completely returned to baseline physiological levels (Figure 4s), confirming the full restoration of systemic homeostasis.

We believe these clarifications and additional data provide a comprehensive understanding of the therapeutic mechanism and immune safety profile, emphasizing the unique contribution of prodigiosin beyond immune stimulation alone.

Revisions Made:

Line319-326: To further rigorously exclude the risk of sustained systemic infection or cytokine storm often associated with bacterial therapies, we additionally monitored the kinetics of neutrophil levels (CD11b⁺ Ly6G⁺) in peripheral blood. Although transient elevation was observed at 24 h post-treatment (consistent with acute immune activation), levels declined significantly by day 14 and returned fully to baseline under normal physiological conditions by day 28 (Figure 4q-s; Figure S21), with no significant difference compared to the PBS control. This kinetic profile confirms that SM@CMC-KP101 elicits a controlled, transient immune response without persistent bacteremia-driven inflammation, supporting the long-term biosafety of this bacterial platform.

Figure 4. Antitumor efficacy of SM@CMC-KP101 in the CT26 tumor model. (q-s) The percentage of CD11b+Ly6G⁺ neutrophils was analyzed by flow cytometry at 24 h (q), 14 days (r), and 28 days (s) post-treatment.

Q7: The synergy index was not calculated (PMCID: PMC11303714). The combination may be additive rather than synergistic.

Response:

We sincerely thank the reviewer for this insightful comment. We fully acknowledge the importance of calculating the synergy index (or Coefficient of Drug Interaction, CDI) to accurately define the nature of the interaction within our drug combination. We agree that distinguishing between additive and synergistic effects provides a more robust evaluation of the therapeutic potential. To address this, we have included previously collected but unpublished data from our *in vivo* studies using the B16F10 melanoma model in C57BL/6 mice (consistent with the experimental design in Result 5). In these experiments, mice were treated with equivalent doses of KP101 and CMC via intravenous injection, and tumor volume changes were monitored over 14 days. These data, now added as Figure S31 in the Supplementary Information, serve as the essential baseline for evaluating the single-agent efficacy needed for CDI calculation.

It is important to note that while PG is a component of the formulation, it was excluded from the synergistic analysis (CDI calculation) for specific scientific reasons. PG exhibits poor water solubility and strictly requires ethanol for dissolution, whereas the final SM@CMC-KP101 system is an aqueous formulation. Introducing an organic solvent (ethanol) for the free PG group would create inconsistent physiological conditions, making a direct comparison unfeasible. Furthermore, free PG lacks intrinsic tumor-targeting capabilities. Therefore, to ensure experimental rigor and fairness, we focused our supplementary experiments and subsequent CDI calculations specifically on the KP101 and CMC components.

Based on the data from the supplementary experiments, we calculated the CDI using the standard formula to quantify the interaction effects. The results are as follows: $CDI_{KP101} = 0.2528$ (indicating significant synergism); $CDI_{CMC} = 1.0046$ (indicating an additive effect). The calculations were performed using the following equation:

$$CDI_{KP101} = \frac{E_{SM@CMC-KP101}}{E_{SM@CMC} \times E_{KP101}}$$

$$CDI_{CMC} = \frac{E_{SM@CMC}}{E_{SM} \times E_{CMC}}$$

where E_{AB} is the ratio of the tumor volume in the combination group to the control group, and E_A and E_B are the ratios for the single agent groups.

We have incorporated the detailed methodology for the CDI calculation, added the supporting data (Figure S31), and cited the suggested reference (PMCID: PMC11303714) in the revised Materials and Methods section.

Revisions Made:

1. Results revision:

Line 365-370: Notably, separate validation experiments showed that single-component treatments (KP101 alone or CMC alone) exhibited negligible anti-tumor efficacy, with tumor growth kinetics indistinguishable from the control group (Figure S33). Using these baselines, we calculated the Coefficient of Drug Interaction (CDI) to quantify the synergy. Relative to KP101, SM@CMC-KP101 exhibited strong synergy (CDI = 0.2528; threshold < 0.7), whereas its interaction with CMC was additive (CDI = 1.0046).

2. Methods revision:

Line 957-964: Evaluation of Synergistic Effects

To evaluate the nature of the interaction between the components, the Coefficient of Drug Interaction (CDI) was calculated based on the tumor volumes measured on day 14 post-treatment. The CDI was determined using the formula:

$$CDI = \frac{E_{AB}}{E_A \times E_B}$$

where AB represents the ratio of the tumor volume in the combination group to that of the control group, and A and B denote the ratios of the tumor volumes in the respective single-agent groups to the control group. According to established pharmacological standards, a CDI value less than 1 indicates a synergistic effect, a value equal to 1 indicates an additive effect, and a value greater than 1 implies an antagonistic effect.

Q8: Figure 4b appears to have an incorrect label for SM@CMC-KP101 (DA).

Response:

We sincerely thank the reviewer for their meticulous attention to detail. We apologize for this oversight regarding the labeling in Figure 4b. We have corrected the label in the revised manuscript to accurately represent the SM@CMC-KP101 (DA) group.

Figure 4. Antitumor efficacy of SM@CMC-KP101 in the CT26 tumor model. (b) in vivo

imaging of fluorescence in various organs at 0 and 24 h post-treatment with different formulations, indicating the biodistribution of SM@CMC-KP101.